# Depressive Symptoms in Expecting Fathers: Is Paternal Perinatal Depression a Valid Concept? A Systematic Review of Evidence

**DOI:** 10.3390/jpm12101598

**Published:** 2022-09-28

**Authors:** Marianna Mazza, Georgios D. Kotzalidis, Carla Avallone, Marta Balocchi, Ilenia Sessa, Ilaria De Luca, Daniele Hirsch, Alessio Simonetti, Delfina Janiri, Emanuela Loi, Giuseppe Marano, Gabriella Albano, Vittorio Fasulo, Stefania Borghi, Angela Gonsalez del Castillo, Anna Maria Serio, Laura Monti, Daniela Chieffo, Gloria Angeletti, Luigi Janiri, Gabriele Sani

**Affiliations:** 1Institute of Psychiatry and Psychology, Department of Geriatrics, Neuroscience and Orthopedics, Fondazione Policlinico Universitario A. Gemelli IRCCS, 00168 Rome, Italy; 2Department of Psychiatry, Università Cattolica del Sacro Cuore, 00168 Rome, Italy; 3Department of Neurosciences, Mental Health, and Sensory Organs (NESMOS), Sapienza University of Rome, Sant’Andrea Hospital, Via di Grottarossa 1035-1039, 00189 Rome, Italy; 4Unit of Clinical Psychology, Fondazione Policlinico Universitario A. Gemelli IRCCS, 00168 Rome, Italy

**Keywords:** peripartum depression, paternal, pregnancy, expecting fathers, couvade syndrome, gender, personalized approach to the patient

## Abstract

Background. Since the identification of Couvade syndrome in the late 1950s, little attention has been dedicated to the issue of depression in expecting fathers. Objective. To quantify the extent of depression in expecting fathers and find out if they match their pregnant partners’ depression. Methods. We conducted a PubMed and ClinicalTrials.gov search using paternal depression and all its variants as terms. We used the *P*referred *R*eporting *I*tems for *S*ystematic Reviews and *M*eta-*A*nalyses (*PRISMA*) 2020 statement to include eligible studies. Results. We identified a grand total of 1443 articles, of which 204 were eligible. The total number of fathers/expecting fathers involved was 849,913. Longitudinal studies represented more than half of the included studies; more than three-quarters of the studies used the Edinburgh Postnatal Depression Scale (EPDS). The average occurrence of paternal depression was around 5%, which confers the entity some clinical dignity. Depression tends to occur more in expecting women and new mothers than in expecting partners or new fathers, while the co-occurrence in the same couple is quite low. Limitations. The methodological heterogeneity of the included studies prevents us from meta-analyzing the obtained data. The validity of the instruments used is another issue. Conclusions. Paternal depression is distinct from maternal depression and occurs at lower rates (about half). The very existence of a paternal depression clinical entity is beyond any doubt. Future research should address methodological heterogeneity.

## 1. Introduction

The fact that a woman’s pregnancy could affect her husband’s mental and physical health has long been known; the English naturalist Robert Plot (1640–1696) observed in 1677 that “In the birth of man it is equally strange, that the pangs of the woman in the exclusion of the child have somtimes affected the *Abdomen* of the *husband*, which yet to such as have experimented the secrecy of *sympathies*,…” [1]. The term used was Couvade syndrome, after a Breton word meaning hatching or brooding [2]. These authors defined Couvade syndrome as “as a state in which physical symptoms of various kinds occur in the husbands of pregnant women, and are of psychogenic origin and connected in some way with pregnancy”. The syndrome is also called sympathetic pregnancy, emphasizing the empathetic and sympathetic nature between wife and husband, which David Hume (1711–1776) envisaged as sympathy [3].

The pathways leading to the sharing of symptoms and distress during a woman’s pregnancy are entangled in the intracouple relationships, which may change during pregnancy and involve increased anxiety and a loss of self-confidence in both members of the couple. For example, one of the two or both may have a sense of inadequacy regarding his/her role as a parent and may thus influence the other, increasing his/her anxiety or helplessness about the new parental role [4]. The pregnancy period is one marked by the continuous and rapid evolution of intracouple dynamics and the resetting of aims and scope in life for each couple member and for the couple as a whole. We may assume that the brain is reoriented towards increased maturation, but sometimes this process is far from complete. Younger and more inexpert parents may act in immature ways and hurt one another’s feelings; in this way, depression may result.

The presence of depression in women during the perinatal period is well recognized. It can be of diverse nature, according to how pregnancy interacts with the basic characteristics of the childbearing woman. The depression could be framed within a major or other depressive disorder or develop within the context of a bipolar diathesis, in which case it would need a different approach and treatment. Differently from women’s perinatal depression, which has been known since the time of Hippocrates (4th Century BC) [5], depressive reactions in the father were recognised only in the 20th century [6]. However, since the late 1950s, the literature on fathers’ depressive reactions during their wives’ pregnancies has not kept pace with that on women’s perinatal depression (1733 records on PubMed on 13 August 2022 vs. 11,602 results on the same database on the same day). This disparity is regretted by investigators of fathers’ perinatal depression, because its consequence is that fewer resources are available for dedicated programs directed to counteracting paternal depression [7,8]. The fact that there are no diagnostic criteria for paternal depression in commonly accepted diagnostic systems [9] renders this disparity a natural consequence.

The presence of depression in a person during the perinatal period could be a factor similar to the synchronization of menses in women living together [10], meaning that it could be easily transmitted to the partner, or the presence of depression could be disjoint in couples, with each member developing his/her own depression during the perinatal period. To respond to this question, we collected all papers providing data on depression in expecting fathers and attempted to compare the presence of depression in men with that in women to see whether they are coordinated. Additionally, we aimed at evaluating the extent to which fathers develop depressive symptoms during their partners’ pregnancy by collecting epidemiological data from studies that evaluated depression using adequate assessment tools.

To this end, we carried out thorough PubMed and ClinicalTrials.gov database searches and reviewed each eligible article systematically.

## 2. Methods

We carried out a PubMed search with the following strategy: (paternal[ti] OR father*[ti] OR marital[ti] OR gender[ti] OR male*[ti] OR man[ti] OR men[ti] OR partner*[ti]) AND (delivery OR post-delivery OR predelivery OR prebirth OR postbirth OR postpartum OR prepartum OR pre-partum OR pregnancy OR post-partum OR perinatal OR postnatal OR trimester*) AND (depression OR depressive OR depressed). The resulting documents were searched for eligibility based on the following criteria: experimental studies containing data on fathers’ depressive symptoms that were adequately assessed and considered the period from 3 months into their wives’ pregnancy to 12 months post-delivery, which had to be normal. Cases of pregnancies considered at risk, stillbirths, where there were no data on depressive symptoms, where there were no data on fathers, no pregnancy, no depression, unrelated to the subject matter, unfocused or inadequate study design, nonhuman (animal) studies, reviews or meta-analyses, opinion papers (editorials, letters to the editor), evaluations after the above specified time range, case reports or series, protocols, overlapping data (reporting data of the same patient population or partially overlapping populations, in which case we included only the best quality article, or when studies were of the same quality, the one including the highest number of subjects), duplicate articles, in vitro studies, and post mortem studies were excluded. We also investigated the ClinicalTrials.gov site for paternal depression (condition) and included studies with data and added those studies that were not duplicates of the articles resulting from our PubMed search.

As adequate means to probe depression in the male population of spouses, we accepted the following instruments: Beck Depression Inventory (BDI); Center for Epidemiologic Studies Depression Scale (CES-D); Depression Inventory (DI) and its short form (DI-SF); Edinburgh Postnatal Depression Scale (EPDS) and its partner version (EPDS-P); Gotland Male Depression Scale (GMDS); Hospital Anxiety and Depression Scale; (HADS); Hamilton Rating Scale for Depression (HDRS or HAM-D); Inventory to Diagnose Depression (IDD); Montgomery-Åsberg Depression Rating Scale (MADRS); Major Depression Index (MDI); Profile of Mood States (POMS); and 21-item Depression, Anxiety, Stress Scales (DASS-21).

We adopted the *P*referred *R*eporting *I*tems for *S*ystematic Reviews and *M*eta-*A*nalyses (*PRISMA*) 2020 statement [11]. The selection process is shown in Figure 1, where reasons for exclusion are detailed. All authors participated in article search (independently) and decision processes to establish eligibility. All eligible articles were downloaded and discussed using Delphi rounds. No more than three were necessary.

## 3. Results

The searches were conducted on 7 February 2022. The PubMed output was 1439 records and the ClinicalTrials.gov was 4 out of 6 (one was recruiting and another not yet recruiting). The total number of records amounted to 1443. A total of 204 were considered eligible (Table 1). The search spanned from January/February 1970 to 3 February 2022, while eligible papers ranged from July/August 1983 to 1 February 2022 (actually published in advance in electronic form on 21 October 2021). The per year distribution is shown in Figure 2.

**Table 1 jpm-12-01598-t001:** Summary of included studies.

Study	Population Studied	Design/Instruments	Results	Conclusions/Observations
[12]	50 first-time fathers, age, $\bar{x}$, mean 27.1 yr, range 18–45	Cross-sectional. Prenatal questionnaire: SMAT, DI, BCAI, FAIOF, FAIOC. Postnatal: DI, BCAI LCI. Subjects completed questionnaires within 3 wks of their infant’s birth and again 3–5 wks postpartum	3 items on the DI, both prenatally and postnatally, reflecting some degree of disturbance of the fathers were irritability, sleep disturbance, and fatigability. The SMAT resulted in a total sample mean of 124.6, which indicated that the sample was maritally well-adjusted. Prenatal BCAI was 2.57 out of 3.0 compared to 2.1 on the postnatal BCAI. The results of the FAIOF showed subjects reporting a mean of 10.8. In contrast, prospective fathers reported on the FAIOC that they planned to be involved in 12.7 out of 13 of the activities with their infant. In LCI new fathers reported lifestyle changes in an average of 38% of the lifestyle items. Thirty-seven subjects reported not having a sex preference for the expected infant. Of the 16 who did have a preference, 12 preferred a boy and four preferred a girl. Prior to the infant’s birth, 36 husbands reported they were equal in importance to the mother, 16 felt less important, and one felt more important. Postnatally, 29 fathers felt equal to the mother, 19 felt less important, and two fathers felt more important in the care of their baby. For 44/53 fathers to be, pregnancies were planned. Expectant fathers reported in the last three weeks of their wife’s pregnancy that the method of infant feeding would be totally breastfed (37), partially breastfed (11), and formula fed (5). 3–5 wks after baby’s birth, fathers reported that 21 infants were totally breastfed, 10 were partially breastfed, and 19 were formula fed	Very few symptoms of depression were noted in the sample, so the correlations between variables, though statistically significant, require further investigation. Greater age, higher income, more years of education, and good marital adjustment were associated with fewer signs of depression. Fathers who reported more lifestyle changes and greater involvement in baby care also reported more signs of depression especially irritability, sleep disturbance, and fatigability. Age and months married did seem to have an impact on the marital adjustment of the couples. Older fathers seemed to project fewer baby care activities while younger fathers planned to do everything. No significant relationship between The FAIOF and the prenatal or postnatal BCAI suggests that many other factors impact collectively to predict the father’s involvement. Unplanned pregnancies (17%) related to greater lifestyle changes
[13]	T0 (Pregnancy) = 147 fathers; T1 (Early post-partum) = 117 fathers; T2 (1 mo.) = 106 fathers; T3 (4 mo.) = 85 fathers; T4 (8 mo.) = 82 fathers; 2/3 of sample Anglo-White, 10% Asian, 14% Hispanic, 3% Black, and 6% other. Age $\bar{x}$ = 31.30	Cross-sectional. GHI, HPQ, STAI, CES-D, SOMS, PSOC, FFFS, MAT, ISSB and questionnaire for perceived support, RSE, LES and a questionnaire for mate’s pregnancy risks	Physical symptoms: no linear increase over time. Psychological Symptoms: At T4 anxiety scores increase, although not significantly. Depression results 8.94, ±7.96; 9.07,± 7.62; 9.21,± 7.18; 8.98, ±7.41; 9.47, ±1.36. General health status indicated a decrease over time	At T4 psychological symptoms of anxiety and depression did not change significantly. At T0, 52% of the variance in depression was explained. A T1, 42%, and T2, 45% of the variance in anxiety was explained. At T3, 43%, and T5, 50% of the variance in depression was explained. A consistent predictor of anxiety and depression was parental competence
[14]	T0 (5 wks pre delivery) = 192 individuals; T1 (8 wks post-delivery) = 177 individuals	Cross-sectional. SSNI, CES-D.	Woman have significantly higher levels of depressive symptomatology at T0. Over time woman’s levels of depression decrease at the trend level, whereas men’s levels increased slightly. Women also scored higher than men on perceived practical and emotional support and reassurance at each timepoint. Overall, social support did not correlate with depressive symptomatology for either gender at T0, whereas overall support was inversely correlated with ♀ symptoms at T1 but not with ♂ symptoms at T1. Each dimension of support was inversely correlated to depressive symptoms in ♀ at T1, whereas only practical support and reassurance were linked to T1 depressive symptomatology in ♂.	♀ and ♂ did not differ in depressive symptoms at T1. The characterization of depressive mood in the postpartum period as a predominantly “female problem” may constitute a social construction of reality inadequately subjected to empirical testing by means of gender comparison
[15]	200 postnatal couples Mothers, Age $\bar{x}$ = 28.8 yr. Fathers, Age $\bar{x}$ = 27.7 yr vs. 155 control couples	Longitudinal. EPDS 13-item version; T0 (6 wks postpartum) T1 (6 mo. postpartum)	Prevalence of depression: 27.5% in mothers at T0 25.7% in mothers at T1 9.0% in fathers at T0 5.4% in fathers at T1 39 of all participants who scored 13 or more on the EPDS at six months postpartum, and their partners agreed on PAS interview. Comparing the results, the sensitivity of the EPDS, using a cut-off score of 13, was 95.7% for cases of depression in mother and 85.7% in fathers. Specificity, 71% for mothers and 75% in fathers.	The prevalence did not differ significantly in either mothers or fathers from a control group. As expected, mothers had a significantly higher prevalence of psychiatric “caseness” both T0 and T1 than fathers. Fathers were significantly more likely to be cases if their partners were also cases
[16]	54 first-time mothers and 42 of their husbands from Oporto (Portugal) ♀$\bar{x}$ = 25 yr range 17–38 ♂$\bar{x}$ = 26.2 yr range 20–37	Longitudinal. EPDS and SADS at T0 = 6 mo. antenatally. T1 = 3 mo. postnatally (subsample:24 ♀ 12 ♂) T2 = 12 mo. Postnatally	Prior pregnancy: 46.3% ♀ had at least one episode of depression. 21.4% ♂ ♀ had a significantly greater lifetime history of depression than did the ♂. during pregnancy: the difference was not statistically significant 16.7%♀ 4.8% ♂ after childbirth: 49.0% ♀ 23.8% ♂ Comparing the ♀ to the ♂ cumulative incidence of depression at the periods 0–3 and 4–12 months after birth, a statistically significant difference was found in the first period, but not in the last	Giving birth and being more directly involved in child rearing renders mothers more vulnerable than fathers to depression in the first few months postnatally. postnatal depression in the fathers seems to follow on from the early occurrence of depression in their wives
[17]	55 married couples. yr range 18–45; expected first child. 97% White, 4.5% Hispanic, 2.7% Asian. ♀ $\bar{x}$ = 30 yr ± 4.6 ♂ $\bar{x}$ = 32 yr ± 4.7	Longitudinal. T0 = ♀ second trimester T1 = 6 mo. Post-delivery. Questionnaire for the perception of work, SSNI, CES-D	$\bar{x}$ CES-D score for ♀ at T0 was 14.9 and 13.6 at T1; for ♂ at T0 was 10.08 and 10 at T1. At T0 and T1 ♀ scores were significantly higher than ♂. At T1 31% of ♀ and 18% of ♂ scores 16 or higher indicating probable depressive symptomatology	When ♂ and ♀ were analyzed together, perceptions of low levels of emotional support from partner and low control and social gratification at work and in the parenting, role were significant predictors of higher depressive symptomatology. Perception of boss supportiveness and social gratification at work were more important for ♀ who were working than for ♂
[18]	370 Irish mothers and their partners. T0 = 289 ♀, 181 ♂ T1 = 224 ♀, 175 ♂	Longitudinal. EPDS, Highs Scale. T0 = 3 d postpartum T1 = 6 wks postpartum	At T0, 11.4% of ♀ scores ≥ 13 on EPDS $\bar{x}$ = 6.8 ± 4.8 and 18.3% scores ≥ 8 on Highs Scale $\bar{x}$ = 4.4 ± 3.1 3% of ♂ scores ≥ 13 on EPDS $\bar{x}$ = 4.1 ± 3.2 at T1 11% of ♀ scores ≥ 13 on EPDS $\bar{x}$ = 6.9 ± 4.7 and 9% scores ≥ 8 on Highs Scale $\bar{x}$ = 3.3 ± 2.8. 1.2% of ♂ scores ≥ 13 on EPDS $\bar{x}$ = 3.1 ± 2.9 and 11% scores ≥ 8 on Highs Scale $\bar{x}$ = 4 ± 3.4	Mothers’ mood state at 3 days postpartum was the best predictor of psychopathology at 6 weeks suggesting that it is possible to identify mothers at risk of postnatal mood disturbance. Mood disturbance in partners was not prominent and when present, it was elation rather than depression, possibly consistent with the effect of a supposed positive life event such as childbirth. Only two partners displayed symptoms of depression 6 weeks postnatally, suggesting that the mood disturbance experienced by mothers after childbirth may be more related to the biological and hormonal aspects of the event rather than the life event
[19]	Longitudinal study. 7018 English partners of female participants. 6667 completed all the tests	Longitudinal. EPDS, a questionnaire investigating family structure, an inventory for stressful life events, a questionnaire for social support, a questionnaire for quality of partnership and a relationships history. T0 = 18 wks of gestation T1 = 8 wks posr birth	At T0 3.5% of ♂ scores ≥ 12 at EPDS $\bar{x}$ = 4.06 ± 3.80. at T1 3.3% of ♂ scores ≥ 12 at EPDS $\bar{x}$ = 3.70 ± 3.76	These rates varied significantly by family type: men in stepfamilies had more than twice the rates found for men in traditional families before the birth and after the birth. Men’s depressive symptoms were correlated with their partners’ depressive symptoms before and following delivery. The correlations between mothers’ and partners’ depressive symptoms in the stepfather families before the birth and after the birth were higher than for the men in other family types before the birth and following the birth.
[20]	51 couples, 26 expecting the first child, 25 the second. ♂$\bar{x}$ = 32.79 yr ± 5.52 ♀$\bar{x}$ = 30.43 yr ± 4.57	Longitudinal. CES-D, PANAS, DAS-S, PSI-SF, COPE T0 = recruitment T1 = 2 wks postpartum	For ♂ only statistically significant correlation was between number of children and coping. For ♀ was between number of children and parenting stress. At T1 ♀PA and NA were significantly correlated. ♂PA and NA were significantly correlated as were PA and depression and depression and NA. At T1 39.2% of ♀ were depressed. 35% mild depression, 50% moderate depression, 15% severe depression. for ♂ 25.5% were depressed. 69.2% mild depression, 30.8% moderately depressed and none severely depressed.	A notable proportion (25–39%) of parents in this study reported elevated depressive symptoms. In nearly half couples (47%) at least one parent reported elevated depressive symptoms, and in nearly 20% both parents did so.
[21]	Longitudinal study. 124 couples with unintended first pregnancy. 58.90/, white, 25% African American, 7.7% Hispanic, and 7.7% Asian ♂$\bar{x}$ = 31.5 yr ± 5 ♀$\bar{x}$ = 30 yr ± 4.1 34% lived in Chicago and 66% in surrounding suburbs. All the couples were living together, and 94.8% were married.	Longitudinal. A questionnaire to investigate pregnancy intent, CES-D, a questionnaire to investigate relationship distress and a questionnaire for social support. T0 = 2–3 mo. antenatal T1 = 4 mo. postpartum	In this sample, pregnancy had been unintentional for 33.9% of ♂ and 29% of ♀. ♀ CES-D T0 $\bar{x}$ = 12.37 ± 9.53 T1 $\bar{x}$ = 6.94 ± 6.22 ♂ CES-D T0 $\bar{x}$ = 6.12 ± 6.43 T1 $\bar{x}$ = 5.21 ± 5.95	A strong association between men’s perception that a pregnancy was unintended and maternal depressive symptomatology was detected. Women’s reports of unintended pregnancy showed a tendency for decreased postpartum depressive symptoms. The findings for men were at the trend level.
[22]	327 couples with a first-time pregnancy, from Melbourne T0 = 251 T1 = 204 T2 = 166 T3 = 151 ♂$\bar{x}$ = 32 yr ♀$\bar{x}$ = 30 yr	Longitudinal. EPDS, PANAS, DI-SF, State anger and anxiety scale, DAS, intimate bond questionnaire, SSQ, the masculine and feminine gender role stress scale. T0 = 24–26 wks antenatally T1 = 36 wks antenatally T2 = 1 wk postnatally T3 = 4 mo. postnatally	Scores ≥ 10 at EPDS for ♂ are: T0 = 12% T1 = 8.7% T2 = 6% T3 = 5.8 For ♀: T0 = 19.5% T1 = 21.1% T2 = 21.6% T3 = 13.9%	The incidence of distress/dysphoria in women and men during mid-and late pregnancy is of concern. The patterns of incidence and onset of distress differed between ♀ and ♂. This difference is characterized by a gradual increase in distress in vulnerable ♀ from mid-pregnancy until after the arrival of the baby, while ♂ who report distress in mid pregnancy appear gradually to resolve this over time leaving a smaller residual group of troubled ♂ by the time the pregnancy is over
[23]	Longitudinal study. 157 first time couples	Longitudinal. SADS, DSQ, IBM, BDI, GHQ, EPDS, PBI, IPSM. T0 = 20–24 wks antenatally T1 = 6 wks postnatally T2 = 12 wks postnatally T3 = 52 wks postnatally	Cumulative incidence of depression by 12 mo. postpartum ♂ = 10.1%, ♀ = 27.3%. Using the BDI cut off > 16: ♀: T0 = 0%; T1 = 7.7%; T2 = 3.7%; T3 = 4.4% ♂: T0 and T1 = 0.7%; T2 = 0.8%; T3 = 3.1%. On GHQ ♀ scores higher than ♂ antenatally and postnatally but not at T3. For ♀ and ♂ there was a positive significant correlation between their level of neuroticism and their depression scores at each of the four assessment points	The prevalence of depression was measured by calculating the percentage of ♂and ♀ who scored above the EPDS (♀, 6 wks postpartum), the BDI, or the GHQ (♀and ♂, remaining timepoints) clinical cut-off points each assessment period. The incidence of self-reported depression in ♂ was consistently lower than in ♀ Theories for this gender difference include an under-reporting of depressed affect by men, either due to a real difference in the experience of depression poorer recall of symptoms by men expressing disturbed affect through different symptoms than those assessed on diagnostic interviews or self-report measures
[24]	251 couples from Sydney. the sample sizes for the data analyses vary from 200 to 218 for the ♂, 230 to 238 for the ♀, and 212 to 218 for couples. Numbers vary depending upon whether the analyses inspect complete self-report data, caseness data, or a combination of both. ♂$\bar{x}$ = 29.1 yr ± 4.6 ♀$\bar{x}$ = 27.2 yr ± 4.2	Cross-sectional. EPDS, CES-D, DIS 6–7 wks postpartum.	11 ♂ met criteria for distress, 3 for depression only, 3 for comorbid depression and anxiety, 5 for anxiety disorder. Of the 208 ♂ providing data for both depression and anxiety modules of the DIS, 2.9% met criteria for depression and 5.3% for distress caseness. 16 ♀ met criteria for depression only, 6 for panic, and 5 for anxiety. Of the 230 ♀, 10.4% met criteria for depression and 16.1% for distress. EPDS scores: Distressed ♂$\bar{x}$ = 9.4 ± 5; Non-distressed ♂$\bar{x}$ = 4.1 ± 3.5. ♂ and ♀ differed significantly on $\bar{x}$ scores (♂$\bar{x}$ = 4.35 ± 3.72 vs. ♀$\bar{x}$ = 6.34 ± 4.33; *p* < 0.01)	The EPDS is both reliable and valid for fathers. It discriminates between distressed and non-distressed fathers, using caseness of either just depression or depression and anxiety. Rate of depression in ♂ may at first seem low. However, when compared to large scale community studies it appears to agree with the general finding of lower rates of depression in husbands than wives
[25]	A 6-mo follow- up of a community sample of women who were evaluated for psychiatric disorders at 2 mo postpartum. 48 couples and 50 controls. Couple $\bar{x}$ age: ♂$\bar{x}$ = 32.0 yr ♀$\bar{x}$ = 29.9 yr Control $\bar{x}$ age: ♂$\bar{x}$ = 33.8 yr ♀$\bar{x}$ = 30.3 yr	Cross-sectional. EPDS, SCID-NP, LIFE, SCL-90-R, a life stress scale of the parenting stress index and a questionnaire for treatment history.	In the original sample, 17 ♂ (24%) were diagnosed with a psychiatric disorder at 2 mo postpartum. 2 of these ♂were lost to follow-up. Of the 15 ♂, 9 (60%) remained symptomatic at 6 mo postpartum. 3 new cases were diagnosed at 6 mo. Anxiety and depression scores were elevated among ♂in the index group, whether or not the ♀were in remission at 6 months. Life stress was found to be correlated with ♀depressive symptoms on the SCL90-R, with ♂ depressive symptoms, and with ♂somatic symptoms.	The results of this study of a community sample of postpartum women and their partners indicates that mental health problems tended to persist for several mo. after the birth of the infant. The results of this study confirm other research showing that the partners of women with postpartum psychiatric disorders often exhibit mental health problems. Many of the fathers appeared to be suffering from chronic mental health problems, which continued to affect them after the birth of their children. However, even among fathers with no psychiatric diagnosis, those whose partners had a postpartum psychiatric disorder continued to report relatively high levels of psychological symptoms at 6 mo. postpartum
[26]	Longitudinal study 127♀ and 122♂ ♂$\bar{x}$ = 31.2 yr (range 17–49) ♀$\bar{x}$ = 28.2 yr (range 18–39)	Longitudinal. GHQ-28, STAI, IES. T0 = 0–4 d after birth T1 = 6 wks postpartum T2 = 6 mo postpartum	Depression: T0♂$\bar{x}$ = 0.101 T0♀$\bar{x}$ = 0.137 T1♂$\bar{x}$ = 0.137 T1♀$\bar{x}$ = 0.055 T2♂$\bar{x}$ = 0.079 T2♀$\bar{x}$ = 0.108 Psychological distress: T0♂$\bar{x}$ = 16.4 T0♀$\bar{x}$ = 22.0 T1♂$\bar{x}$ = 17.9 T1♀$\bar{x}$ = 17.2 T2♂$\bar{x}$ = 15.9 T2♀$\bar{x}$ = 16.7 State Anxiety: T0♂$\bar{x}$ = 29.5 T0♀$\bar{x}$ = 30.5 T1♂$\bar{x}$ = 30.2 T1♀$\bar{x}$ = 29.2 T2♂$\bar{x}$ = 15.9 T2♀$\bar{x}$ = 30.1	The present study clearly demonstrates that within a six-month perspective, the birth of a healthy child only provokes long term psychological distress in 19% of mothers and 11% of fathers
[27]	2 samples of first-time parents from Sydney. First sample: 216 ♀ 196 ♂ ♂$\bar{x}$ = 29.1 yr ± 4.6 (range 20–44) ♀$\bar{x}$ = 27.2 yr ± 4.2 (18–41) Second sample: 192 ♀ 160 ♂ ♂$\bar{x}$ = 29.4 yr ± 4.7 (range 20–47) ♀$\bar{x}$ = 27.5 yr ± 3.5 (19–38)	EPDS, CES-D, POMS 6–8 wks postpartum. At the 6-week home interview the mother and father were separately administered the Diagnostic Interview Schedule: Depression and Anxiety modules	♀ meeting criteria for both depression and anxiety at 6 wks postpartum had a significantly higher antenatal EPDS score ($\bar{x}$ 11.7 ± 5.8) than those with just anxiety ($\bar{x}$ 8.3 ± 4.4) depression ($\bar{x}$ 7.0 ± 3.2). ♂ with just depression scored significantly higher on their antenatal CES-D ($\bar{x}$ 16.5 ± 18.2) than those with either no disorder postpartum ($\bar{x}$ 7.4 ± 6.2) or a mix of anxiety and depression ($\bar{x}$ 6.0 ± 6.8)	Do not appear to be a clear pathway for the differential development of an anxiety or depressive disorder postpartum for ♀. A history of an anxiety disorder appears to be a greater risk factor for the development of postpartum mood disorder than a history of a depressive disorder. No Such finding was evident for the development of disorders in ♂
[28]	472 pregnant ♀and 308 expectant fathers ♂ from Lübeck, Germany, divided in 3 groups: Group 1: 88 ♀ 61 ♂ Group 2: 344 ♀ 223 ♂ Group 3: 40 ♀ 24 ♂	Cross-sectional. ADS-L for depressive reaction. Short questionnaire of actual stress by Müller and Basler for perceived stress.	ADS-L scores: Group 1: Normal: 90% ♀ 98% ♂ Depressive: 9.5% ♀ 2% ♂ Group 2: Normal: 90% ♀ 97.5% ♂ Depressive: 10% ♀ 2.5% ♂ Group 3: Normal: 94% ♀ 96% ♂ Depressive: 6% ♀ 4% ♂ Stress reaction before prenatal testing Group 1: ♀ $\bar{x}$ = 3.4 ± 0.9 ♂ $\bar{x}$ = 3.1 ± 0.8 Group 2: ♀ $\bar{x}$ = 3.4 ± 1.5 ♂ $\bar{x}$ = 3.1 ± 0.8 Group 3: ♀ $\bar{x}$ = 3.9 ± 1.3 ♂ $\bar{x}$ = 3.6 ± 0.5 Stress reaction after prenatal testing: Group 1: ♀ $\bar{x}$ = 3.1 ± 1.4 ♂ $\bar{x}$ = 2.9 ± 1.0 Group 2: ♀ $\bar{x}$ = 2.5 ± 1.3 ♂ $\bar{x}$ = 2.6 ± 1.0 Group 3: ♀ $\bar{x}$ = 3.3 ± 1.2 ♂ $\bar{x}$ = 3.1 ± 0.5	ADS scores ≤ 23 normal ≥23 depressive. Questionnaire for stress total scores range from 1 (minimum stress) to 6 (maximal stress). Comparable analysis of depressive reactions before prenatal diagnosis showed no significant difference between the prenatal test groups neither for the pregnant women nor for their partners
[29]	284 ♂. 216 ♂ their partner having had a ‘normal’ unassisted delivery and 68 ♂ their partner miscarried < 24th week of gestation $\bar{x}$ = 32.1 yr ± 7.3 (18–56)	Longitudinal. IES, BDI, STAI-state, CRI. T0 = pregnancy (beginning of second trimester) T1 = Birth/Miscarriage T2 = 1 yr after birth/miscarriage	BDI scores: T0 $\bar{x}$ = 5.15 ± 2.42 T1 $\bar{x}$ = 8.61 ± 4.44 T2 $\bar{x}$ = 5.76 ± 3.11 STAI-Y1 scores: T0 $\bar{x}$ = 37.91 ± 9.06 T1 $\bar{x}$ = 38.73 ± 9.96 T2 $\bar{x}$ = 34.51 ± 9.47	Simple effects revealed that between T0 and T1 there was a significant increase on all of the outcome measures with the exception of anxiety, and a change in the coping repertoire used. Between T0 and T2, there was a significant reduction in all outcome measures. In the present sample of men, pregnancy was associated with high levels of stress and anxiety, above that which would be expected for a non-psychiatric population. At pregnancy outcome, levels of stress, anxiety, and depression all increased irrespective of whether the outcome was a live birth or miscarriage. One year following outcome, anxiety and stress levels had fallen significantly below pregnancy levels, whereas depression levels, whilst showing a decrease compared to at the time of pregnancy outcome, remained significantly high compared to pregnancy
[30]	48 ♂. 28 face to face survey 20 postal survey	Cross-sectional. HADS, EPDS, Brief PHQ. Brief PHQ cut-off > 10 EPDS cut-off > 12 HADS cut-off ≥ 8	HADS: 8% ♂ mild depressed mood. EPDS > 12: 8% ♂ 2/4 ♂ > cut-off both HADS and EPDS	The prevalence of depressed paternal mood in the postnatal period was notable at 8%, however, depressed men did not participate in the follow-up study. There was a strong association between higher paternal depressed mood and fussier/more difficult infant temperament.
[31]	106 couples ♀$\bar{x}$ = 32.13 yr ± 4.39 (22–44) ♂$\bar{x}$ = 34.18 yr ± 5.5 (24–52) 52% primiparous	Longitudinal. Blues questionnaire, EPDS (cut-off 9/10), PBQ, ICQT1 1 wk postpartumT2 2 mo postpartum	Blues questionnaire ♀$\bar{x}$ = 29.77 ± 16.57 ♂$\bar{x}$ = 18.32 ± 10.11 EPDS scores T1♀$\bar{x}$ = 6.09 ± 5.05 ♂$\bar{x}$ = 4.28 ± 2.64 T2♀$\bar{x}$ = 4.38 ± 3.79 ♂$\bar{x}$ = 2.5 ± 2.37	The mean EPDS score was significantly higher in mothers than in fathers both at one week and at two months. New mothers experienced more blues symptoms, had higher overall mean percentage scores on the Blues Questionnaire than new fathers. The mothers’ mean percentage score of blues for each day peaked on day4 after the delivery, while fathers’ peaks on day 1
[32]	Cross-sectional study at $\bar{x}$ 22 wks ♀ gestation 50 teenage couples 50 non-teenage couple (control group) Teenage ♂$\bar{x}$ = 20.7 yr Non-teenage ♂$\bar{x}$ = 29.6 yr	Cross-sectional. HADS (HADS-A/HADS-D), GHQ-28	Teenage ♂ scores: HADS median: 11.5 HADS-A median: 7.5 HADS-D median: 4 GHD-28 median: 9 Non-teenage ♂ scores: HADS median: 3 HADS-A median: 4 HADS-D median: 1 GHD-28 median: 0	The younger age of onset of fatherhood was independently associated with higher HADS and GHQ-28 scores in a multivariate analysis. Symptoms in anxiety domains had a stronger association compared to those in depressive domains in both the HADS and GHQ-28
[33]	367 couples with an ART pregnancy. ♀$\bar{x}$ = 33.2 yr ± 4.4 ♂$\bar{x}$ = 35.2 yr ± 5.8; 379 spontaneous pregnancy couples (control). ♀$\bar{x}$ = 33.3 yr ± 3.0 ♂$\bar{x}$ = 34.1 yr ± 5.4; T0 = 18–20 wks of gestation T1 = 2 mo post-partum T2 = 12 mo post-partum	Cross-sectional. GHQ-36, STAI, checklist of nine changes and stressors for stressful life events.	♂in the control group reported higher levels of depressive symptoms than ♂ in the ART group. Among ART ♂in both groups, depressive symptoms ↑ after the baby was born. ♂’s anxiety symptoms first ↓from pregnancy to post-delivery, and then ↑ at T2 in both ART and control groups	ART is not a risk factor for the development of psychological symptoms in mothers and fathers to be
[34]	11833 ♀ 8431 ♂	Longitudinal. EPDS cut-off ≥ 12 8 wks post-partum. ♂ reassessed at 21 mo. post-partum.	EPDS score ≥ 12: 10% ♀ 4% ♂	♂ depression might have a direct effect on the way ♂ interact with their children, as has been reported for postnatal depression and other psychiatric conditions in ♀.
[35]	156 ♀ and their partner at 20 wks antenatally. ♀$\bar{x}$ = 28.4 yr ♂$\bar{x}$ = 33.5 yr First assessed with CES-D and divided in 2 groups: 36% depressed ♀ 32% depressed ♂	Cross-sectional. CES-D, STAI, STAXI, Daily Hassles Scale.	Depressed ♂ score: CES-D: $\bar{x}$ = 8.52 ± 6.0 STAI: $\bar{x}$ = 34.9 ± 7.7 STAXI: $\bar{x}$ = 18.3 ± 5.9 Daily hassles scale: $\bar{x}$ = 20.6 ± 5.2 Non depressed ♂ score: CES-D: $\bar{x}$ = 21.4 ± 9.3 STAI: $\bar{x}$ = 44.2 ± 8.7 STAXI: $\bar{x}$ = 21.8 ± 3.6 Daily hassles scale: $\bar{x}$ = 28.3 ± 6.5	Depressed ♂ as compared to non-depressed ♂ experience depression and anxiety symptoms during pregnancy, not unlike the depression and anxiety symptoms reported for depressed ♀ versus non-depressed ♀ during pregnancy. The level of depression and anxiety symptoms are not significantly different for depressed ♀ and depressed ♂ during pregnancy.
[36]	5089 couples	Cross-sectional. Short form of CES-D. Total score between 10 and 14 moderate depression and ≥15 severe depression.	14% ♀ and 10% ♂ had moderate or severe depressive symptoms. CES-D score: ♀$\bar{x}$ = 4.58 ± 4.96 ♂$\bar{x}$ = 3.69 ± 4.67	Postpartum depression in ♂ was strikingly high (10%) and more than twice as common than in the general adult male population in the US
[37]	58 ♂ with partner with PPD (index group) ♂$\bar{x}$ = 34.1 yr ± 6.5 ♀$\bar{x}$ = 32.2 yr ± 5.2 116 ♂without partner with PPD (comparison group) ♂$\bar{x}$ = 34.3 yr ± 5.2 ♀$\bar{x}$ = 32.3 yr ± 4.5	Cross-sectional. BDI-II cut-off > 13, BAI cut-off > 7, GHQ-28 cut-off > 5, SPHERE, AUDIT, AQ	Index group score: AQ $\bar{x}$ = 58.9 ± 16.7 AUDIT $\bar{x}$ = 6.5 ± 4.5 BAI $\bar{x}$ = 3.8 ± 4.5 BDI-II $\bar{x}$ = 8.0 ± 6.4 GHQ-28 $\bar{x}$ = 4.1 ± 4.3 SPHERE $\bar{x}$ = 5.1 ± 7.4 Comparison group: AQ $\bar{x}$ = 52.2 ± 11.9 AUDIT $\bar{x}$ = 5.9 ± 4.7 BAI $\bar{x}$ = 2.5 ± 3.5 BDI-II $\bar{x}$ = 4.8 ± 4.7 GHQ-28 $\bar{x}$ = 2.4 ± 3.3 SPHERE $\bar{x}$ = 1.9 ± 3.6	Index group reported poorer psychological health than comparison group, on depression, nonspecific psychological impairment and aggression.
[38]	386 couples. ♂$\bar{x}$ = 30.3 yr ± 8.8	Cross-sectional. BDI (cut-off ≥ 10 presence of depression; ≥19 mild/severe depression) 12th weeks postpartum	BDI >9: ♀ 23.6%; ♂ 11.9% BDI >18: ♀ 7.8%; ♂ 4.1%	As severity of ♀ depression increases, prevalence of ♂PPD increase. On the contrary, we had no information on paternal depression before the delivery, and, therefore, we are not able to state that maternal depression caused paternal depression.
[39]	13 partners of ♀ with PND; ♂ $\bar{x}$ = 29.8 yr ± 5.4	Cross-sectional. BDI-II, SSNI, DAS, PSI	♂ scores: BDI-II $\bar{x}$ = 14.76 ± 6.82; SSNI $\bar{x}$ = 2.89 ± 0.92; DAS $\bar{x}$ = 99.69 ± 16.25; PSI $\bar{x}$ = 84.46 ± 15.57	Partners of women with PND showed none-to-mild depression levels
[40]	194 ♂ with different methods of recruitment.124 via general practice postal5 via general practice attenders18 via child health surveillance clinic attenders28 via postnatal ward face-to-face19 via postnatal ward postal	Cross-sectional. HADS cut off ≥ 8 (8–10 = mild, 11–14 = moderate and 15–21 = severe)	♂ HADS scores >8: Depressive symptoms 12%; Anxiety symptoms 30%	17 of 23 ♂ scoring above the cut-off for depressive symptoms also scored above the cut-off for anxiety symptoms.
[41]	101 couples ♂$\bar{x}$ = 32 yr ± 5.0 ♀$\bar{x}$ = 30 yr ± 5.1 T0 1–2 d postpartumT1 2 wks postpartumT2 6 wks postpartum	Longitudinal. EPDS, EPDS-P, BDI, HDRS. ♂ EPDS-P at T1 and T2♀ EPDS, BDI T0, T1, T2 HDRS over the phone at T2	T1 EPDS $\bar{x}$ = 6.8 ± 4.9 EPDS-P $\bar{x}$ = 8.2 ± 4.3 BDI $\bar{x}$ = 9.2 ± 6.8T2 EPDS $\bar{x}$ = 5.0 ± 4.8 EPDS-P $\bar{x}$ = 7.4 ± 5.0 BDI $\bar{x}$ = 6.4 ± 6.4 HDRS $\bar{x}$ = 5.4 ± 6.0	The EPDS-P was found to be moderately related to both a clinician rating of depression and the women’s self-reported depression ratings. As expected, due to reliability of self-report data, the EPDS showed a significantly stronger correlation with the BDI and the HDRS than did the EPDS-P.
[42]	367 couples with IVF/ICSI379 controls with spontaneous pregnancyT1 18–20 wks of gestation T2 2 mo postpartumT3 12 mo postpartum	Longitudinal. DAS at T2 and T3, BDI 13 item version at T1, Stressful Life Events questionnaire	Stressful life events ART group: ♂ 24.5% ♀ 23.4% Control group ♂ 29.0% ♀ 29.3% Depressive symptoms ART group None ♀ 18.3%, ♂ 35.7% Mild ♀ 31.2%, ♂ 28.0% Moderate ♀ 46.7%, ♂ 33.3% Severe ♀ 3.7%, ♂ 2.9% Control group None ♀ 18.4%, ♂ 33.5% Mild ♀ 36.9%, ♂ 31.8% Moderate ♀ 38.5%, ♂ 31.2% Severe ♀ 6.1%, ♂ 3.4%	Successful ART bears no risk for marital adjustment
[43]	128 mother–father–infant triads	Cross-sectional. EPDS (cut-off: ≥10), PSI-SF, DAS, NCATS 2–3 wks postpartum	28% ♀ and 13.3% ♂ had depression at 2–3 wks postpartum. EPDS, ♂$\bar{x}$ = 5.25; ♀$\bar{x}$ = 7.64	Men whose partners had depression had significantly higher depression scores than did men whose partners had no depression. Maternal PPD affected study fathers in negative ways, as shown by higher levels of depression and parenting stress among men whose partners had depression. Paternal marital satisfaction was not associated with maternal depression at 2 to 3 months postpartum
[44]	171 ♀ $\bar{x}$ = 31.9 yr ± 4.7 133 ♂ $\bar{x}$ = 33.8 yr ± 5.4	Longitudinal. Blues Questionnaire, EPDS, VAS. VAS mood asked: happiness, depressed, tiredness, anxiety. EPDS cut-off ≥ 10. Assessments at 1 week postdelivery and 2 months postdelivery	1 wk postpartum: Blues Questionnaire: ♂ $\bar{x}$ = 18.61 ± 9.93 ♀ $\bar{x}$ = 30.56 ± 16.05. VAS: Depressed: ♂ $\bar{x}$ = 7.8 ± 10.75 ♀ $\bar{x}$ = 19.01 ± 18.4 Happiness: ♂ $\bar{x}$ = 84.35 ± 12.7 ♀ $\bar{x}$ = 81.18 ± 14.6 Tiredness: ♂ $\bar{x}$ = 52.06 ± 24.16 ♀ $\bar{x}$ = 64.96 ± 24.16 Anxiety: ♂ $\bar{x}$ = 26.01 ± 20.58 ♀ $\bar{x}$ = 35.83 ± 25.11. EPDS: ♂ $\bar{x}$ = 4.47 ± 2.76 ♀ $\bar{x}$ = 6.52 ± 5.08. Only 1 scored above 10 (0.75%). At 2 mo. 12% of ♀ scores 10 or higher on EPDS $\bar{x}$ = 4.76 ± 3.91 ♂$\bar{x}$ = 2.8 ± 2.44 significantly lower than the score at 1 wk	It was expected that ♀ showed more blues and depressive symptoms than ♂. This has also been found antenatally and in the first months postpartum. Two subscales of Blues Questionnaire’s ‘primary blues’ and ‘hypersensitivity’ was most predictive for high EPDS scores at 2 months in new ♂, while in ♀ the subscale ‘depression’ was most predicted for depressive symptoms later on.
[45]	Survey 687 ♀ (28/31 wks pregnant) 669 ♂	Cross-sectional. EPDS (♀ cut off ≥ 12/13, ♂ cut-off > 11), Marital Satisfaction Scale, The Duke-UNC functional social support questionnaire.	Depressed ♀: 10.3% depressed ♂: 6.5%. Depression of ♂ when ♀ experience depression in pregnancy: 14.5% of ♀ when ♂ experience depression during their partner’s pregnancy: 23.3%	Low marital satisfaction increased the probability of depression during pregnancy in women and men. Among men low affective social support was associated with depression during pregnancy. The prevalence of depression during pregnancy was higher among ♀ and although most psychosocial and personal factors associated with depression during pregnancy were similar for both sexes, low affective social support and partner’s depression were only related to ♂ depression.
[46]	9846 (76.4%) ♂ at T0 8332 (64.7%) ♂ at T1, 7090 (55.0%) ♂ at T2, 6101 (47.4%) ♂ at T3, 4792 (37.1%) ♂ from all four time points. Data with at least one time point: 10,975 ♂ T0: 18 wks prenatally; T1: 8 wks postnatally; T2: 8 mo. Postnatally; T3: 21 mo. postnatally	Longitudinal. EPDS cut off >12	Depression score > 12 for sample with data at all four time point: T0: 2.88% (CI 2.41–3.35); T1: 2.55% (CI 2.10–3.00); T2: 2.32% (CI 1.89–2.75); T3: 3.05% (CI 2.56–3.54); Depression score >12 using imputed data for those with data on at least one time point: T0: 3.88% (CI 3.52–4.24); T1: 3.64% (CI 3.29–3.99); T2: 3.44% (CI 3.10–3.78); T3: 3.87% (CI 3.51–4.23)	Depression in men is relatively common. There is a suggestion from these findings that depressive symptoms are even more common in the prenatal period than postnatally, although the proportion of fathers with high scores changes little across this period.
[47]	2 groups:53 ♂ (IVF) $\bar{x}$ = 34.1 yr ± 4.236 ♂ control $\bar{x}$ = 33.3 yr ± 2.7	Cross-sectional. PFA, FIAI, STAI, KSP, EPDS	Results at 2 mo. postpartum: EPDS: IVF ♂$\bar{x}$ = 3.3 ± 3.2. Control ♂$\bar{x}$ = 3.4 ± 2.6. STAI: IVF ♂$\bar{x}$ = 28.3 ± 5.8. Control ♂$\bar{x}$ = 26.9 ± 5.2	None of the control ♂ but 7.5% of the IVF men scored 10 or more on the EPDS.
[48]	199 ♂T0: antenatally T1: 6 wks postnatally	Longitudinal. Trait section of STAI, PLDS, PCS antenatally. PTSD-Q, IES, state section of STAI, EPDS at T1	EPDS score: 8% $\bar{x}$ = 3.7 ± 4.1 State anxiety STAI-Y1: 6.6% $\bar{x}$ = 32.1 ± 9.9; Clinically significant PTSD in at least one dimension (intrusions, avoidance, and hyperarousal) in 12% of participants	No evidence of PTSD in expecting fathers; rates of depression and anxiety quite low.
[49]	270 ♀ and 213 ♂; ♀age ≤ 19: 13.4%; 20–29: 44.5%; 30–39: 40%; ≥40: 2.1%. ♂age ≤ 19: 4.4%; 20–29: 39%; 30–39: 48.2%; ≥40: 8.4%	Longitudinal. T0: 1st trimester; T1: 2nd trimester; T2: 3rd trimester. STAI, EPDS (cutoff >10)	STAI ♀ score: T0: $\bar{x}$ = 36.16 ± 9.00; T1: $\bar{x}$ = 34.69 ± 9.40; T2: $\bar{x}$ = 36.33 ± 9.12. STAI ♂ score: T0: $\bar{x}$ = 32.66 ± 8.43; T1: $\bar{x}$ = 31.13 ± 7.98; T2: $\bar{x}$ = 32.77 ± 7.94. EPDS ♀ score: T0: $\bar{x}$ = 6.55 ± 4.22; T1: $\bar{x}$ = 6.17 ± 4.39; T2: $\bar{x}$ = 5.63 ± 4.27. EPDS ♂ score: T0: $\bar{x}$ = 5.11 ± 3.97; T1: $\bar{x}$ = 4.22 ± 3.38; T2: $\bar{x}$ = 4.09 ± 3.32.	Symptoms of anxiety were higher in T0, ↓ in T1 and ↑ in T2. Symptoms of depression ↓ throughout pregnancy, with a significant ↓ occurring from the T0 to the T1 and again from the T1 to the T2. it is noteworthy that the T2 seems to be a period of relative calm in terms of psychological morbidity, and significant ↓ both in anxiety and depression symptoms were observed.
[50]	ART parents. T1: 2nd trimester 460 ART couples, 400 controls. T2: 2 mo postpartum 396 ART (324 singletons, 91 twins’ pairs), 319 controls (304 singletons, 20 twin pairs). T3: 1 yr postpartum 325 ART (270 singletons, 55 twin pairs), 262 controls (251 singletons, 11 twin pairs). ART group $\bar{x}$ age: 90 ♀ of twins $\bar{x}$ = 31.7 yr ± 4.0; 364 ♀ of singletons $\bar{x}$ = 33.0 yr ± 4.2. 85 ♂ of twins $\bar{x}$ = 33.7 yr ± 4.5; 357 ♂ of singletons $\bar{x}$ = 35.0 yr ± 5.8.Control group $\bar{x}$ age: 20 ♀ of twins $\bar{x}$ = 33.1 yr ± 5.3; 379 ♀ of singletons $\bar{x}$ = 33.3 yr ± 3.0. 20 ♂ of twins $\bar{x}$ = 32.9 yr ± 5.4; 379 ♂ of singletons $\bar{x}$ = 34.2 yr ± 5.4	Longitudinal. GHQ-36	ART ♂ of twins group score: T1 Depression $\bar{x}$ 1.24, Anxiety $\bar{x}$ 1.44, Sleeping difficulties $\bar{x}$ 1.72; T2 Depression $\bar{x}$ 1.26, Anxiety $\bar{x}$ 1.38, Sleeping difficulties $\bar{x}$ 1.66; T3 Depression $\bar{x}$ 1.36, Anxiety $\bar{x}$ 1.50, sleeping difficulties $\bar{x}$ 1.76.ART ♂ of singletons group score: T1 Depression $\bar{x}$ 1.22, Anxiety $\bar{x}$ 1.38, Sleeping difficulties $\bar{x}$ 1.65; T2 Depression $\bar{x}$ 1.22, Anxiety $\bar{x}$ 1.34, Sleeping difficulties $\bar{x}$ 1.87; T3 Depression $\bar{x}$ 1.28, Anxiety $\bar{x}$ 1.36, sleeping difficulties $\bar{x}$ 1.60.CONTROL ♂ of twins group score: T1 Depression $\bar{x}$ 1.27, Anxiety $\bar{x}$ 1.46, Sleeping difficulties $\bar{x}$ 1.72; T2 Depression $\bar{x}$ 1.37, Anxiety $\bar{x}$ 1.53, Sleeping difficulties $\bar{x}$ 1.87; T3 Depression $\bar{x}$ 1.41, Anxiety $\bar{x}$ 1.59, sleeping difficulties $\bar{x}$ 2.00.CONTROL ♂ of singletons group score: T1 Depression $\bar{x}$ 1.29, Anxiety $\bar{x}$ 1.44, Sleeping difficulties $\bar{x}$ 1.61; T2 Depression $\bar{x}$ 1.26, Anxiety $\bar{x}$ 1.39, Sleeping difficulties $\bar{x}$ 1.62; T3 Depression $\bar{x}$ 1.28, Anxiety $\bar{x}$ 1.36, sleeping difficulties $\bar{x}$ 1.59.	ART mothers of twins reported fewer symptoms of depression than control mothers of twins during pregnancy. The higher levels of depression and anxiety after delivery in both ART and control mothers of twins than in mothers of singletons confirm previous findings of an increased risk of post-partum depression associated with spontaneously conceived multiple births. Twin birth, but not ART, had a negative impact on the mental health of fathers.
[51]	130 ♀ and 130 ♂. ♀$\bar{x}$ = 29.34 yr ± 2.96; ♂$\bar{x}$ = 31.92 yr ± 3.15	Cross-sectional. EPDS (cut-off ≥ 13), PSS, SSRS,	EPDS score ≥13: 13.8% ♀ $\bar{x}$ 8.44 ± 3.88 (range 3–23), 10.8%♂ $\bar{x}$ 7.95 ± 3.60 (range 3–19). SSRS ♀ $\bar{x}$ 35.32 ± 3.96; ♂ $\bar{x}$ 8.48 ± 6.34. PSS ♀ $\bar{x}$ 16.82 ± 4.33; ♂ $\bar{x}$ 16.91 ± 3.67.	Results show no significant differences between paternal and maternal EPDS scores which indicates that depression is a problem for both women and men in the postpartum period. no significant differences between paternal and maternal perceived stress as indicated by the PSS scores.
[52]	87 couples. ♀$\bar{x}$ = 28.95 yr (range 20–37); ♂$\bar{x}$ = 30.98 yr (range 21 to 43)	Cross-sectional. PBI, IRS, STAXI, CES-D, ways of coping-R, infant crying questionnaire.	♀ scores: Emotional minimization: $\bar{x}$ 1.82 ± 0.53; global empathy: $\bar{x}$ 3.80 ± 0.46; trait anger: $\bar{x}$ 1.88 ± 0.55; depression: $\bar{x}$ 1.51 ± 0.32. ♂ scores: Emotional minimization: $\bar{x}$ 1.89 ± 0.52; global empathy: $\bar{x}$ 3.66 ± 0.53; trait anger: $\bar{x}$ 1.81 ± 0.53; depression: $\bar{x}$ 1.43 ± 0.31	Depressive symptoms were linked with more parent-oriented beliefs about crying for fathers, supporting the notion that the pattern of affect and cognition affiliated with depression undermines an orientation toward others’ needs.
[53]	178 mothers, 146 fathers (Japan)	Cross-sectional. 4 wk after birth: EPDS ≥ 8, CES-D ≥ 16	14% depression; no association between maternal and paternal depression. Paternal depression associated with employment status, history of psychiatric trtm, and unintended pregnancy. 8 fathers with unstable employment (7 temporary employees, 1 unemployed)	Perinatal care providers should independently screen for depression in fathers and mothers and focus attention on paternal employment status, especially temporary employment
[54]	Prospective cohort. 551 ♂ $\bar{x}$ 33.4 yr ± 5.9 (18–59). 57♂ who dropped out at 8 wks post-partum $\bar{x}$ 32.4 yr ± 5.5	Cross-sectional. EPDS, BDI, PHQ-9. All participants who scored above the BDI cut-off score of 10.5 or the EPDS cut-off score of 9.5 were invited to return for a psychiatric diagnostic interview (SCID-NP)	Score of groups who completed the study: EPDS $\bar{x}$ 4.9 ± 4.3. BDI: $\bar{x}$ 3.8 ± 5.0; Score of groups who dropped: EPDS $\bar{x}$ 5.4 ± 5.2. BDI: $\bar{x}$ 5.2 ± 7.1	Evidence suggested that postnatal depression in ♂ may have a later onset, probably following the occurrence of depressive symptoms in their partners. In contrast to ♀ who reported relatively high rates of depression during pregnancy and immediately following childbirth, depression in ♂ tends to ↑ over the first postnatal yr and reaches its peak at 12 months postpartum.
[55]	Recruited from 2 different hospitals. Hospital A: 469 ♀$\bar{x}$ 30.7 yr ± 5.0; 307♂$\bar{x}$ 32.0 yr ± 5.8. Hospital B: 394 ♀$\bar{x}$ 29.8 yr ± 5.2; 218♂$\bar{x}$ 31.8 yr ± 5.3	Cross-sectional. EPDS, WPBL-R	EPDS scores: hospital A ♀: $\bar{x}$ 6.6 ± 4.2 (≥13: 9%) ♂ $\bar{x}$ 3.3 ± 2.9 (≥13: 1.3%). hospital B ♀: $\bar{x}$ 6.9 ± 4.5 (≥13: 12.2%) ♂ $\bar{x}$ 3.3 ± 3.4 (≥13: 2.8%)	Self-concept, depressive symptoms, infant centrality and parent’s state of mind on discharge emerged as the most significant parent attributes affecting mothers’ and fathers’ parenting satisfaction.
[56]	189 ♂$\bar{x}$ 35.0 yr ± 5.86 and 184 ♀33.3 yr ± 4.84	Cross-sectional. EPDS, SCID, Fathers depression was diagnosed by clinical interview.	Depressed ♂ EPDS score: $\bar{x}$ 14.79 ± 3.41. non depressed: $\bar{x}$ 6.64 ± 4.40	Depression in ♂ after the birth of the child is associated with an adverse impact on child development, independently of the ♀’sMood.
[57]	1562 ♂ partners of severely peripartum depression women	Cross-sectional. SCL-90	35% of partners scored high on depression items of the SCL-90	About 35% of expecting fathers share their partner’s depressive mood
[58]	Cohort study. 687 ♀ 669 ♂ Subjects who participatedat all of three phases were included, 420 (61.13%) ♀ and409 (61.14%) ♂. T1: 3rd trimester of pregnancy; T2: 3 mo post partum; T3: 12 mo postpartum	Longitudinal. EPDS (cut off ≥ 11), ENRICH Marital Satisfaction Scale, The Duke-UNC Functional Social Support Questionnaire	77.9% ♀ and 87.8% ♂ report no increase of depression from 28 and 31 weeks of pregnancy to 12 months post-partum. The incidence of depression during 12 mo post-partum was higher among ♀ and ♂ with low marital satisfaction (14.7% and 8.8%, respectively), low affective social support (17.8% and 10.3%, respectively), low confidant social support (22.5% and 10.9%, respectively), and with pregnancy depression (32.7% and 33.3%, respectively). Also, this incidence was higher when the partner of either sex was depressed (27.8% in ♀ and 16.3% in ♂)	The incidence of depression was higher among ♀ than ♂at T2, but was similar among ♀ and ♂at T3; the incidence of depression ↓ during the first postpartum year for ♀, but the association is at the limit of statistical significance; psychosocial and personal factors associated with depression 1 year after childbirth were similar for both ♀ ♂ (marital satisfaction, partner’s depression, pregnancy depression), negative life events were only related to ♀’s depression. Lack of affective and confidant social support did not increase the risk of depression in either ♀ ♂.
[59]	739 fathers	Longitudinal. PPD and BD prevalence in fathers; MINI: T1(28–34 wk), T2(30–60 postpartum), T3(12 mo postpartum)	Depression: 5%, 4.5%. 4.3%; BD: 3%. 1.7%. 0.9%. Depressive episodes in fathers significantly associated with manic/hypomanic episodes during pregnancy and postpartum period; not 12 mo after birth. Delivery may act as a specific event whose effects decrease over time	Bipolar episodes were common in men with depressive symptoms during their partner’s pregnancy in the postpartum period and, to a lesser extent, 12 mo after birth. This population should be carefully investigated for manic and hypomanic symptoms. The high prevalence of bipolar related episodes may depend on the instrument used and on this specific period of life which might act as a stressor in association with genetic vulnerability, increasing the risk of highly heritable forms of affective disorders
[60]	1562 ♂ 7 weeks postpartum	Cross-sectional. EPDS, SCID	EPDS score: 335 (85.6%) scored ≤ 10 and 199 (12.74%) scored ≥ 10. A random selection of the EPDS low scorers (n = 266) and all the EPDS high scorers (n = 199) were contacted to obtain permission to be assessed using the SCID diagnostic tool for depression. Thirty-one (16%) ♂ met the DSM-IV criteria for depression while 94 (49%) ♂ failed to meet these criteria. Sixty- seven (35%) ♂ were not currently depressed but were deemed at ↑ risk of developing depression.	The main finding of this study suggests that depression in ♂ in the postnatal period is associated with increased healthcare costs. In terms of incremental costs for each category of care, depression was associated with significantly ↑ community care costs
[61]	99 fathers with diagnosed depressive disorders, 54 w/o	Cross-sectional. To investigate association between paternal depressive disorder and family and child functioning in the first 3 mo of a child life (families seen at home 3 mo after birth). SCID, EPDS ≥ 10, AUDIT, DAS, IBQ, antisocial personality problems scale, perceived criticism in the couple relationship self-report.	Depression in fathers is associated with an increased risk of disharmony in partner relationship, reported by both fathers and their partners, controlling for maternal depression. Few differences in infant’s reported temperament found in the early postnatal period	Depressive disorder affecting fathers is associated with an increased risk of inter-parental conflict in the postnatal period. Paternal depression. Paternal depression seems to be associated with somewhat higher levels of difficulties in infant temperament
[62]	260 couples. ♀ age ≤ 19: 11.8%; 20–29: 46.7%; 30–39: 39, 5%; ≥40: 2.0%. ♂ age ≤ 19: 4.4%; 20–29: 39.0%; 30–39: 48, 2%; ≥40: 8.4%.	Longitudinal. STAI-S, EPDS ≥ 10. T0: 1st trimester; T1: 2nd trimester; T2: 3rd trimester; T3: 1–3d afterbirth; T4: 3 mo postpartum	STAI-S ♀ scores: T0: 13.1%; T1: 12.2%; T2: 18.2%; T3: 18.6%; T4: 4.7%. ♂ scores: T0: 10.1%; T1: 8.0%; T2: 7.8%; T3: 8.5%; T4: 4.4%. EPDS ♀ scores: T0: 20.0%; T1: 19.6%; T2: 17.4%; T3: 17.6%; T4: 11.1%. ♂ scores: T0: 11.3%; T1: 6.6%; T2: 5.5%; T3: 7.5%; T4: 7.2%.	Results generally show ↑ rates of depression compared to anxiety, in ♀ compared to ♂, and during pregnancy compared to T4. Depression was more prevalent than high-anxiety in ♀ either early in pregnancy or at T4. ♀ were more likely than men to show high-anxiety at T2 and at T3, but not during early pregnancy or T4.
[63]	185 ♀ $\bar{x}$ 30.9 yr ± 5.5 (16.4–43.8) 140 ♂ $\bar{x}$ 33.0 yr ± 6.4 (19.7–52.5)	Cross-sectional. M-PHI or F-PHI, SF-12 (MCS, PCS), EPDS, WEMWBS	♀ scores: EPDS: $\bar{x}$ 6.9 ± 5.2 (0.0–23.0); WEMWBS: $\bar{x}$ 50.6 ± 8.6 (27–70); SF-12 MCS: $\bar{x}$ 48.5 ± 10.0 (17.1–64.1); SF-12 PCS: $\bar{x}$ 53.0 ± 6.5 (29.5–62.9).♂ scores: EPDS: $\bar{x}$ 4.3 ± 4.0 (0.0–18.0); WEMWBS: $\bar{x}$ 53.5 ± 9.0 (29–70); SF-12 MCS: $\bar{x}$ 52.0 ± 7.1 (30.6–62.0); SF-12 PCS: $\bar{x}$ 53.3 ± 5. (28.8–62.4)	This was a validation study; M-PHI and F-PHI showed fair convergence with other instruments, especially the EPDS
[64]	376 parents. ♀ $\bar{x}$ 26.64 yr ± 3.38 (21–36); ♂ $\bar{x}$ 27.09 yr ± 4.46 (22–39)	Cross-sectional. PSS, SSRS, EPDS ≥ 13.	PSS scores ♀$\bar{x}$ 17.16 ± 3.75; ♂ $\bar{x}$ 16.03 ± 3.67. SSRS scores ♀$\bar{x}$ 38.53 ± 5.56; ♂ $\bar{x}$ 35.43 ± 4.87. EPDS scores ♀$\bar{x}$ 8.62 ± 3.88; ♂ $\bar{x}$ 8.03 ± 3.60	There were no significant differences in the prevalence of PND or EPDS scores between the new ♀ and ♂. The results showed that there was no significant difference between ♀ and ♂ perceived stress, indicating that the ♂ experienced similar levels of stress as ♀ at postnatal period. Perceived stress and social support were key predictors of EPDS scores for both ♀ and ♂ during the postnatal period.
[65]	Prospective cohort study. 551 couples. ♂$\bar{x}$ 33.4 yr ± 5.9	Cross-sectional. EPDS, BDI, SCID	Prevalence of PND 4.9% (2.4% major and 2.5% minor)	Postnatal depression in one partner correlated with postnatal depression in the other. This association has been consistently demonstrated but is not likely to be related to common risk factors. ♂ postnatal depression was predicted by life events and stress.
[66]	650 ♂	Longitudinal. MINI	2.6% of ♂ were depressed	Suicide risk in ♂ ↑ from prepartum to postpartum
[67]	303 couples at T1: within the first seven days of the birth. 227 couples T2: 6 wks post-partum. 231 couples T3: 3 mo. 220 couples completed all 3 time points. ♀ $\bar{x}$ 31.7 yr (19–44) ♂ $\bar{x}$ 34.3 yr (21–56)	Longitudinal. IES, PTSD-Q, EPDS, STAI, SOS, ECR	♀ EPDS score: T2: $\bar{x}$ 7.14 ± 4.87. T3: $\bar{x}$ 7.14 ± 4.87. ♂ EPDS score: T2: $\bar{x}$ 3.94 ± 3.96. T3: $\bar{x}$ 3.41 ± 4.02; ♀ STAI score: T1: $\bar{x}$ 35.55 ± 8.34. ♂ STAI score: T2: $\bar{x}$ 34.45 ± 7.64.	♀ appeared to be differentially adversely affected by ♂ early symptoms; ↑ levels of acute symptoms in ♂ appeared to be particularly linked to ♀ later distress while the reverse was not true for ♂. Symptoms of post-traumatic stress and postpartum depression were positively related within couples.
[68]	RCT study. Intervention consisting in education sessions. 1574 individuals in a couple relationship (862 ♀ 712 ♂). 315♂ control $\bar{x}$ 29.4 yr (17–54); 365♂ intervention $\bar{x}$ 29.5 yr (16–51). At 6 wks 556 ♂ (253 control, 303 intervention)	Longitudinal. HADS-A, HADS-D. As part of that trial a repeated measures cohort study was conducted to identify changes in self reported levels of anxiety and depression between the ♂ in the intervention group and the ♂ in the control group from bl to six wks	↓ anxiety levels from bl to 6 wks were significant in the intervention group but were not significant in the control group. In the intervention group 12.4% of the ♂ had less anxiety from bl to six weeks postnatal compared to 11.4% of ♂ in the control group. Both the intervention and control groups reported ↓ antenatal anxiety at 6 wks postpartum the antenatal and postnatal depression scores remained unchanged at 4% in both groups.	The intervention may have contributed to the ↓ anxiety scores in the intervention group by providing timely, relevant information to assist the fathers. Most of the fathers in both groups did not register any depression either ante or postnatal with no real changes from bl 6 wks post test, suggesting that depression levels remained constant.
[69]	231♂ $\bar{x}$ = 31 yr ± 6.3 (range 20 to 49)	Cross-sectional. SCID, CAGE Questionnaire	SCID results: Prevalence of any mental disorders 17.7%; Depressive episode 5.2%; Panic disorder 0.9%; Generalised anxiety disorder 4.3%; Co-morbid anxiety disorder and depression 5.2%.	The prevalence of any depressive disorder in ♂ in this study (12.6%) is ↑ than the pooled prevalence in high-income countries (9%) and much ↑ than in the studies which used the same diagnostic assessment in well-resourced Asian countries: Singapore (1.8%) and Hong Kong (3.1%). Panic and generalised anxiety disorders were apparent in 10.4%, but there are no data from other resource-constrained settings with which these can be compared.
[70]	Extensive cohort study. Bl fist visit to child health center, T1: 3 mo postpartum. 305 couples. Depressed ♀ $\bar{x}$ 30.6 yr ± 4.46; Non depressed ♀ $\bar{x}$ 29.9 yr ± 5.03. Depressed ♂ $\bar{x}$ 32.5 yr ± 5.15; Non depressed ♂ $\bar{x}$ 33.0 yr ± 5.65	Longitudinal. At bl: DCS, DAS. T1: EPDS	260 ♀ and 252 ♂ had answered all EPDS questions. 16.5% of ♀ and 8.7% of ♂ sufferedfrom postpartum depressive symptoms according to the EPDS cut-off of >9. ♀ and ♂ with depressive symptoms scored ↑ levels of discord compared to parents without depressive symptoms.	16.5% of the ♀ and 8.7% of the ♂ self-reported depressive symptoms. Results indicated that nearly a quarter (23%) of the children had at least one parent with postpartum depressive symptoms.
[71]	199 couples. ♂ $\bar{x}$ 36.44 yr ± 7.22; ♀ $\bar{x}$ 34.53 yr ± 4.43	Cross-sectional. EPDS (9/10 cutoff), EPDS-P (partner report), IDD	EPDS depression ♂: 12.6%; IDD depression ♂: 4.5%	The EPDS-P was found to be a reliable and valid measure of paternal depression based on the measure’s internal consistency and the measure’s relation with other well-validated measures of depression.
[72]	1403 ♂. Age 18–20: 11.26% (158); 21–25: 29.01 (407); 26+: 59.73% (838). Interviewed at the birth of a child and followed up when the child was 1, 3, and 5 years old.	Longitudinal. CIDI-SF, paternal involvment, PSI, marital relationship.	Age 18–20 score ♂ depression case 1st year 8% (67); 3rd year 10.50% (88); 5th year 9.67% (81). Depressive symptoms score 1st year: none: 91.41% (766); score 1–2: 0.60% (5); score 3–8: 8% (67). 3rd year: none: 91.41% (766); score 1–2: 0.60% (5); score 3–8: 8% (67). 5th year none: 90.10% (755); score 1–2: 0.24% (2); score 3–8: 9.67% (81). Age 21–25 score ♂ depression case 1st year 10.07% (41); 3rd year 13.27% (54); 5th year 11.79% (48). Depressive symptoms score 1st year: none: 88.21% (359); score 1–2: 1.72% (7); score 3–8: 10.07% (41). 3rd year: none: 85.15% (344); score 1–2: 1.49% (6); score 3–8: 13.37% (54). 5th year none: 87.22% (355); score 1–2: 0.98% (4); score 3–8: 11.79% (48). Age 26+ score ♂ depression case 1st year 9.49% (15); 3rd year 16.46% (26); 5th year 12.03% (19). Depressive symptoms score 1st year: none: 87.97% (139); score 1–2: 2.53% (4); score 3–8: 9.49% (15). 3rd year: none: 81.41% (127); score 1–2: 1.92% (3); score 3–8: 16.67% (26). 5th year none: 87.34% (138); score 1–2: 0.63% (1); score 3–8: 12.03% (19)	Emerging adult fatherhood was not related to depression. Rates roughly ranged from 8–16%, depending on the age status of fathers and in which year the depressive symptoms were measured. Late adolescent fatherhood, was a significant predictor of third-year depression. Late adolescent fathers have a 2-fold risk for depression than adult fathers when their children were about 3 years-old
[73]	231 ♂ $\bar{x}$ 31 yr ± 6.3	Cross-sectional. SCID	SCID major depression or dysthymia: 7.4% of ♂; generalized anxiety: 5.2% ♂; co-morbid depression and anxiety: 5.2% of ♂	This study is mean to establish the validity of a group of psychometric instruments. The selected cut-off points for EPDS in ♂ was 4/5, one point higher than that in ♀. The optimal cut-off for each scale was selected as the score that produced the maximum combination of Se and Sp
[74]	Cohort study. T1: 18 wk gestation; T2: 8 mo. postnatally. 11,954 ♀ and 9846 ♂	Longitudinal. EPDS cut off ≥ 12	EPDS score ≥ 12 T1: 13.8% of ♀, 4% of ♂. T2: 8.8% of ♀, 2.9% of ♂. 360 ♀ and 59 ♂ were depressed at both T1 and T2	Parental depression in the postnatal period was associated with adverse child outcomes. Postnatal ♂ depression was found to predict later total problems and conduct difficulties in children
[75]	296 young couple♀$\bar{x}$ 18.7 yr ± 1.63 (14–21), ♂$\bar{x}$ 21.3 yr ± 4.06; Couples completed interviews via separate audio computer-assisted self interviews	Cross-sectional. LES, 15 of 20 items of CES-D	Depression: ♀$\bar{x}$ 10.55 ± 7.39; ♂$\bar{x}$ 8.88 ± 6.62	Among both young ♀ ♂, the relationship between stressful life events and depression disappeared among those with the highest amount of social support. For ♂ with the most family functioning, stressful life events had the strongest association with increased depression, although the trend across levels of family functioning was slight and less dramatic compared to ♀.
[76]	This study uses data collected in a randomized controlled trial. 812 ♂ $\bar{x}$ 31 yr ± 2 yr. The sample was divided in groups by the age: ≤28: 240; 29–33: 354; ≥34: 218	Cross-sectional. EPDS at 3 mo after birth. (cut-off ≥ 11)	Elevated score on the EPDS: ♂ ≤ 28 yr of age: 16.3%; ♂ 29–33 yr: 7.1%; ♂ ≥ 34 yr: 8.7%. Total depression 10.25%	♂ aged 28 years and younger had a more than twofold ↑ risk of depressive symptoms 3 mo. after the birth compared with ♂ 29–33 years of age, whereas ♂ aged 34 years and older had no ↑ risk. Factors associated with an ↑ risk for depressive symptoms included having a low income, low level of education, poor partner relationship quality, and worries about economy and employment
[77]	199 couples completed BL interviews at 4 wk postpartum, 172 couples at 6 mo postpartum. All couples completed separate computer-assisted telephone interviews. ♀ $\bar{x}$ 30.6 ± 5.0; ♂ $\bar{x}$ 32.8 ± 5.6	Longitudinal. CIDI, EPDS (cut-off ≥ 9 for ♀; ≥10 for ♂)	9.3% ♀ 4.1% ♂ had at least one anxiety disorder; a total of 57 ♀ (33.1%) and 30 ♂ (17.4%) met criteria for at least one common mental disorder in the first 6 months postpartum. EPDS scores at 6 mo postpartum: ♀$\bar{x}$ 4.3; ♂ $\bar{x}$ 3.4	EPDS scores of women and men did not correlate significantly. ♀ EPDS scores at 6 mo postpartum were significantly ↑ than ♂
[78]	T1 18–20 wks prepartum: 153 couples; T2 32–34 wks prepartum: 153 couples; T3 3 mo postpartum: 127 couples	Longitudinal. EPDS, STAI, PRAQ.	EPDS scores: T1 ♀ $\bar{x}$ 5.3; ♂ $\bar{x}$ 4.0; T2: ♀ $\bar{x}$ 5.1; ♂ $\bar{x}$ 3.3; T3: ♀ $\bar{x}$ 4.6; ♂ $\bar{x}$ 3.6. STAI scores: T1 ♀ $\bar{x}$ 31.3; ♂ $\bar{x}$ 29.8; T2: ♀ $\bar{x}$ 32.9; ♂ $\bar{x}$ 30.0; T3: ♀ $\bar{x}$ 31.2; ♂ $\bar{x}$ 30.0.	The EPDS and STAI scores in this study were slightly lower than in two Australian studies that also included pregnant mothers and fathers.
[79]	262 ♂	Cross-sectional. Prime-MD	47 of 262 ♂ satisfied the criteria for Prime-MD depression or anxiety, 18 depression (2 major and 16 minor), 11 with co-morbid depression and anxiety (6 major, 4 minor and 1 dysthymia), 18 with anxiety only	The study aimed to evaluate how well the EPDS identify depression and anxiety in ♂
[80]	205 ♂; child $\bar{x}$ age = 11 mo (Québec)	Cross-sectional. A descriptive-correlational study. Psychosocial factors associated with PPD: EPDS ≥ 10, DAS, PES, PSI	8.2% ♂ (infants brestfed ≥ 6 mo) positive to EPDS 8–14 mo after birth. Quality of spousal relationship associated with paternal depression 3 mo after birth persists over time. ♂ in depressed group reported ↑ levels of parenting distress and lower sense of parenting efficacy. Depression in fathers of breastfed infants is associated with the experience of perinatal loss in a previous pregnancy, parenting distress, infant temperament, dysfunctional interactions with child, decreased marital adjustment and perceived low parenting efficacy	♂ depression associated with certain psychosocial factors. The depressed ♂ reported lower quality of marital relationship, perception of lower parenting efficacy and greater parenting distress. These finding indicate the importance of screening for postnatal depression among ♂ of breastfed infants, not only in the wks following the birth of the child, but throughout the first-year f the child’s life
[81]	64 couples. ♀ $\bar{x}$ 29.0 yr ± 3.5; ♂$\bar{x}$ 30.0 yr ± 6.0. T1: 1 mo postpartum; T2: 6 mo postpartum; T3: 12 mo postpartum.	Longitudinal. EPDS (cut-off of ≥13 for major depression), SCID	♀ EPDS score: T1 $\bar{x}$ 4.4 ± 0.6; T2 $\bar{x}$ 4.7 ± 0.5; T3 $\bar{x}$ 4.8 ± 0.5. ♂SCID score: Substance abuse 32.3%; Anxiety disorders 9.7%; Mood disorders 29.0%; Psychosis 3.2%; Comorbid-mood /substance abuse 25.8%	While ♀ depressive symptoms were correlated throughout the first twelve months postpartum, ♂’s mental health moderated the course of these symptoms. Separate from ♀ history of depression, a ♂ substance abuse-related Axis I diagnosis was associated with augmented ♀ depressive symptoms, relative to a ♂ mood related diagnosis or no ♂ diagnosis. The psychiatric lifetime history of the ♂ may be a critical piece in the regulation of maternal depression. According to ♂ lifetime psychiatric interviews, 48.4% of ♂ had a lifetime history of mental1 illness
[82]	393 couples; 401 ♀ $\bar{x}$ 30 yr ± 5; 396 ♂$\bar{x}$ 33 yr ± 6; T0: (demographic data) = 393 couples); T1: (3 mo. Post-partum) = 308 couples; T2: (18 mo. Post-partum) = 272 couples	Longitudinal. EPDS >9; SOC; ICQ	Depressive symptoms measured at T1 = 17.7% ♀; ♂ 8.7% and correlated between mothers and fathers within couples. Mothers and fathers with depressive symptoms had a poorer sense of coherence and perceived their child’s temperament as more difficult than mothers and fathers without depressive symptoms at 3 and 18 months. Post-partum depressive symptoms did not differ according to the parents’ age, child’s sex, first or not first child, SES or mothers’ educational level.	Early parenthood has been studied thoroughly in mothers, but few studies have included fathers. Depressive symptoms were more common in fathers with senior high school educational level compared with those with higher education or 9-year compulsory school
[83]	349 ♀ (first-time) 270♀, 122 ♂ (Longitudinal sample)	Longitudinal. EPDS, CBCLT0 = prenatally T1 = 2 mo postnatally T2 = 6 mo postnatally T3 = 8–9 yr postnatally	The differences between ♂ and ♀ views remained significant considering both internalizing and externalizing problems in boys, and nearly significant regarding internalizing problems in girls. Regarding ♂ reports, ♀ depressive symptoms was associated with significantly elevated level of child’s total and externalizing problems. According to ♀ reports, the finding was similar but for total problems, the difference lacks statistical significance. For internalizing problems, the difference between the groups with and without ♀ depressive symptoms was not significant according to either of the parents	This study aimed at examining the specific view of ♂ and agreement between parental views considering the competence and emotional/behavioural symptoms of their child and the impact of ♀ depressive symptoms on parental perceptions. The findings in this study showed that ♀depressive symptoms affect the child, also from the ♂ perspective
[84]	110 couples. ♂ $\bar{x}$ 31.9 yr ± 5.02 ♀ $\bar{x}$ 28.6 yr ± 4.89	Cross-sectional. FIF, MIF, EPDS	The medical problems in pregnancy: 33.6%. 48.2% of ♀ reported anxiety about motherhood. This rate was 32.7% for ♂ because of a new baby. The proportion of PPD was 9.1% ♀. 1.8% ♂. The previous history of the depression was 7.3% ♀. 0.9% ♂. 17.3% ♀ had a history of depression in close relatives, and this proportion 9.2% ♂. EPDS: ♀4.29 ± 5.33; ♂1.12 ± 2.75	PPD is a serious problem that affects ♀ health and well-being marital relationship as well as offspring’s health and well-being
[85]	163 ♂; 163 ♀	Longitudinal. T1 = 3 mo.; T2 12 mo. (DAS; EPDS)	♀♂ were found to display quite similar EE scores. Regression analyses showed that depression and couple relationship significantly predicted EE in ♀, but not ♂. High levels of depressive symptoms in ♀ predicted increased critical comments in ♂ and less warmth with their children	The findings of the present study are of potential clinical importance, as the parenting characteristics identified by the assessment of EE and its constituent components are potentially amenable to clinical intervention. As these findings relate to the first year of a child’s life the possibility is raised of useful early intervention for depressed parents and their children
[86]	205 ♂ $\bar{x}$ 32.63 yr ± 5.00	Cross-sectional. EPDS, PSS, MSPSS.	EPDS score: $\bar{x}$ 5.77 ± 4.50. PSS score: $\bar{x}$ 45.36 ± 8.06. MSPSS score: $\bar{x}$ 12.21 ± 6.55	The frequency of depression in ♂ was 11.7%. Increase in the ♂’s stress score was associated with increased rates of depression in them. The regression analysis suggested that perceived stress as a factor effecting depression is of high importance in predicting ♂’s PPD
[87]	BiT study, ‘Barnhälsovård i Tiden’ [Child Health Care Today]. Longitudinal cohort study. T1: 1 wks post-partum 401 ♀ 396 ♂. T2: 3 mo post-partum 336 ♀ 333 ♂T3: 18 mo post-partum320♀ 314 ♂T4: 6–8 yr after childbirth 391 couples	Longitudinal. EPDS, SOC-3, SPSQ	T2: EPDS score ≥ 10 = 17.7% 59 ♀, 8.7% 28 ♂EPDS score ≥ 12 = 9.9% 33 ♀ and 5.0% 16 ♂. The main finding of the present study is the association between the time to marital separation and ♂ dyadic discord, mothers’ and/or fathers’ depressive symptoms, and ♀ parental stress during early parenthood. ♀♂selfestimated less dyadic consensus, more depressive symptoms, and higher parental stress than those who were not separated.	The data from the present study showed an association between low dyadic consensus, depressive symptoms and parental stress during early parenthood, and an increased risk of marital separation 6–8 years after childbirth. This knowledge is important for health professionals and could be useful in developing interventions to provide parents with adequate support during pregnancy and early parenthood
[88]	622 ♂ T0: 12 wks gestation; 337 ♂ T1: 36 wks gestation; 150 ♂ T2: 6 mo postpartum. A total of 187 participants completedall three time points of the survey. ♂$\bar{x}$ 34.19 yr ± 5.21 (19–55)	Longitudinal. EPDS > 12/13, MSPSS, RSE	EPDS scores: T0: $\bar{x}$ 5.22 ± 3.59; T1: $\bar{x}$ 5.23 ± 3.67; T2: $\bar{x}$ 5.15 ± 4.17.	Using ≥13 as the standardized cut-off for probable case of depression, the prevalence increased as the pregnancy progress and reached a peak at T2, with 3.3% of the participants scoring above cut-off in T0, 4.1% in T1 and 5.2% in T3, respectively. ♂ antenatal depression, especially ♂ depression in late pregnancy, could significantly predict ↑ level of depression among the expectant ♂ in the postpartum period.
[89]	102 ♂ recruited between the second (>24th week) and third trimester of their partner’s pregnancy. ♂$\bar{x}$ 35.82 yr ± 5.95 (23–55)	Longitudinal. EPDS (T0: between the 2nd and 3rd trimester of pregnancy; T1: 4–6 wks postpartum) cut-off > 9, STAI.	EPDS score: T0: $\bar{x}$ 4.17 ± 3.59; T1: $\bar{x}$ 4.04 ± 3.23. STAI scores: $\bar{x}$ 49.61 ± 11.19	9.8% of the expectant ♂ showed signs of elevated depressive symptoms during their partner’s pregnancy by having scores of 9 or more on the EPDS and are therefore at risk of being diagnosed with a minor or major depressive disorder. Considering our results, on the one hand, this could imply that after adjusting to the new life circumstances, fewer fathers tend to suffer from depressive symptoms postnatally compared to the prepartum which is consistent with the findings of previous studies.
[90]	172 couples	Cross-sectional. EPDS cut-off >12, VPSQ, RLCQ	Perceiving the partner as more controlling was significantly associated with more depressive symptoms in the individual but also in his or her partner. Perceiving the partner as more caring was not significantly associated with fewer symptoms in either the individual or the partner, once other relevant variables were controlled for. Higher levels of depressive symptoms were also associated with having a more vulnerable personality, other coincidental stressful life events and having a more unsettled baby	♀♂ vulnerable personality traits, coincidental adverse life events and more infant crying and fussing were also associated with significantly more depressive symptoms. The quality of the intimate partner relationship is significantly associated with postnatal mental health in both women and men, especially in the context of coincidental stressful events including infant crying
[91]	18,552 from the Millenium Cohort Study (MCS)	Longitudinal. T1(9 mo): Rutter’s 9-item Malaise Inventory (shortened version) T2 (3 yr): Child-Parent Relationship ScaleT3 (5 yr)/T4 (7 yr): Fathers’ parenting activity (involvement) was measured using answers to the amount of parenting activities they undertook with their child	Findings suggest that postnatal paternal depressive symptoms are associated with fathers’ negative parenting. Paternal depressive symptoms significantly predicted higher levels of father-child conflict. Paternal depressive symptoms were not associated with paternal involvement, suggesting that the quality of parenting is influenced by depressive symptoms, but the duration of time spent with the child is not altered. Both maternal depressive symptoms and marital conflict moderated the association between paternal depressive symptoms and father-child conflict	Despite reports showing the huge costs of paternal depression, parenting interventions are still primarily targeted towards mothers. Authors advocate a more family-centred approach and provided that appropriate support and services are put in place, they suggest routine screening for postnatal depressive symptoms in fathers
[92]	897 families; the sample was divided into three groups: (Group A) families with ♀ who have experienced physical/sexual abuse (32.8 yr ± 2.2); (Group B) ♀ with experienced emotional abuse/neglect (33.4 yr ± 2.5); (Group C) ♀ no traumatic experiences (31.4 ± 2.2).	Longitudinal. T0: SCL-90-RT1/T2: SCL-90–9, SVIA	The results showed that ♀ with early traumatic experiences (Groups A and B) had significantly more maladaptive interactions during the feeding of their children, both at 3 mo and 6 mo of age, when compared to mothers who had not experienced traumas (Group C).SCL-90-R mothers’ subscale scores showed a main effect of the group with no time-point effect and no interaction effect between time point and group. The ♀ in Group B had significantly higher scores at T2 than Group A on the somatization, depression, and paranoid ideation subscales. The scores of the ♀ in Group C on all SCL-90-R subscales were significantly lower than those of Groups A and B, both at T1 and T2	Some authors have suggested that sexual abuse can have a more severe impact on subjects’ psychopathology and has more frequently been associated with psychiatric diagnoses (e.g., PTSD). It has also been evidenced that ♀ victims of emotional abuse have a higher risk of psychopathological (including depressive) symptoms
[93]	14,541 pregnant ♀ database; compared data of 418 mothers and 114 fathers with depression	Longitudinal. T0: 18 we gestation (the only timepoint at which mother to be and father to be depression levels were recorded) T1: 21 mo postpartum. EPDS cut-off > 13; CCEI cut-off ≥ 8; CIS-R	EPDS: 418 mothers (10.7%) and 114 fathers (3.3%) reported symptoms of antenatal depression; 725 mothers (16.8%) and 238 fathers (6.9%) reported symptoms of antenatal anxiety (CCEI). Children of mothers with high depressive symptoms at 18 weeks gestation had an increased risk of anxiety diagnosis at 18	This study showed an increased risk of offspring anxiety at 18 years of age after exposure to maternal antenatal depression at 18 weeks gestation. This association was not seen following exposure to paternal depression. These findings highlight the differences between antenatal depression exposures in different parents. This adds to support for a fetal programming effect occurring during pregnancy, leading to potentially long-lasting effects on the anxiety state of offspring.
[94]	53 couples ♂$\bar{x}$ = 34.45 yr ± 5.2 ♀$\bar{x}$ = 30.35 yr ± 5.6	Cross-sectional. EPDS; PSI-SF	EPDS 20.8% ♀ 5.7♂ reported high risk of PPD; 18.9% of ♂ reported borderline scores indicating risk of subclinical PPD; ♂ reported very high scores both in global distress level. No significant correlation between parenting distress and the risk of PPD emerged, both in ♀ than in ♂ group while distress ♀ levels are related to paternal one. Additional analysis regarding the association between the desire of pregnancy and the level of PPD suggests that there is a significant difference between ♂who desired a child e ♂ who did not desire a child	During this critical life event, some of couples of parents experience a high vulnerability and refer significant distress levels; mood disturbances and parenting stress in postpartum period represent high risks of parents and children well-being. Dysfunctional parenting has been assumed as an important risk factor in the development of psychological disturbances in adulthood and several studies have reported a significant correlation between maternal PPD and altered cognitive/affective child development.
[95]	T1 2 wks postpartum: 317 ♀ (249 intervention group, 68 control group), 276 ♂ (210 intervention group, 66 control group). T2 6 mo postpartum: 216 ♀ (181 intervention group, 35 control group), 185 ♂ (152 intervention group, 33 control group). Total ♀$\bar{x}$ = 28.83 yr ± 5.22, IG ♀$\bar{x}$ = 28.83 yr ± 5.22, CG ♀$\bar{x}$ = 30.32 yr ± 5.18; Total ♂$\bar{x}$ = 32.18 yr ± 6.21, IG ♂$\bar{x}$ = 31.62 yr ± 6.28, CG ♂$\bar{x}$ = 33.96 yr ± 5.66	Longitudinal. EPDS (≥ 13 for ♀, ≥ 11 for ♂)	T1 EPDS score: Total ♀$\bar{x}$ = 7.22 ± 5.06, IG ♀$\bar{x}$ = 7.14 yr ± 5.04, CG ♀$\bar{x}$ = 7.52 yr ± 5.14; Total ♂$\bar{x}$ = 4.27 yr ± 3.57, IG ♂$\bar{x}$ = 4.20 yr ± 3.41, CG ♂$\bar{x}$ = 4.51 yr ± 4.06. T2 EPDS score: Total ♀$\bar{x}$ = 5.19 ± 4.71, IG ♀$\bar{x}$ = 5.03 yr ± 4.54, CG ♀$\bar{x}$ = 6.07 yr ± 5.49; Total ♂$\bar{x}$ = 3.76 yr ± 3.67, IG ♂$\bar{x}$ = 3.98 yr ± 3.65, CG ♂$\bar{x}$ = 2.76 yr ± 3.67.	A significant number of parents reported high levels of psychological distress shortly after birth and at 6 mo. postpartum.
[96]	1797 ♀, 1658 ♂	Longitudinal. RDAS; SPIN; BDIT1: 20th pregnancy week. T2: 8 mo post-partum. T3: 18 months post-partum	The test-retest reliability correlations between consecutive measurement points were at least moderate indicating that both ♀ and ♂ social and emotional loneliness were stable during the study period. Concerning ♂ social loneliness, similarly to ♀ the largest class consisted of fathers with very low and even continuously decreasing feelings of loneliness. These groupings revealed that the higher the loneliness was, the more the parents experience these other psychosocial problems	Becoming a parent may increase both mothers’ as well as fathers’ feelings of social and emotional loneliness and these phenomena are highly associated with lower levels of marital satisfactions and higher levels of social phobia and depression
[97]	807 couples	Cross-sectional. EPDS (♂ cut-off ≥ 8; ♀ cut-off ≥ 9)	EPDS ♂ scores: 110 (13.6%) scored ≥8. The $\bar{x}$ age of these ♂ 33.4 ± 5.7 yr.	The prevalence of ♂ depression at four months after childbirth was 13.6%. The factors that were significantly correlated with ♂ depression were the presence of partner’s depression, low marital relationship satisfaction, pregnancy with infertility treatment, experience of visiting a medical institution due to a mental health problem, and economic anxiety
[98]	230 families (230 ♀ $\bar{x}$ = 32.6 yr ± 4.66; 173 ♂ $\bar{x}$ = 35.8 yr ± 7.58)	Longitudinal. EPDS cut-off ≥ 13; BDI cut-off ≥ 18; SCID-I	♀ Preterm EPDS T1 $\bar{x}$ = 9.0 ± 5.69. Term EPDS T2 $\bar{x}$ = 4.7 ± 4.14. Preterm BDI T1 $\bar{x}$ = 9.2 ± 6.76) T2 $\bar{x}$ = 5.2 ± 3.68) ♂ Preterm EPDS T1 $\bar{x}$ = 5.7 ± 4.29 T2 $\bar{x}$ = 3.0 ± 2.99. Term BDI T1 $\bar{x}$ = 5.8 ± 4.30 T2 $\bar{x}$ = 4.0 ± 3.59	The risk of PPD was 4 to 18 times higher for mothers and 3–9 times higher for ♂ of VLBW infants compared to mothers and fathers of term infants. Mean scores on the depression scales and prevalences of clinical PPD were higher in parents of VLBW infants than in parents of term infants, with ♀ in both groups at a higher level than ♂. The most important risk factor for PPD was the birth of a VLBW infant itself. Family related factors like SES, primipara, a pregnancy of high risk or with multiples were not as relevant as individual parental factors like sex, lifetime psychiatric diagnoses and social support
[99]	Data were from four waves of the Personality and Total Health (PATH) Through Life Survey, a longitudinal, population-based survey assessing the health and well-being of the residents of Canberra and Queanbeyan (New South Wales) in Australia. The sample has been divided in EF (expectant father): 88 ♂$\bar{x}$ 22.7 yr ± 1.4; NF (new father) 108 ♂$\bar{x}$ 22.7 yr ± 1.5; AF (alredy father) 61 ♂$\bar{x}$ 23.4 yr ± 1.4	Longitudinal. Goldberg Depression and Anxiety Scales, SF-12	Depression score (0–9): EF: $\bar{x}$ 2.0 ± 1.9; NF: $\bar{x}$ 2.2 ± 2.1; AF: $\bar{x}$ 2.7 ± 2.3. Anxiety score (0–9): EF: $\bar{x}$ 2.7 ± 2.6; NF: $\bar{x}$ 2.8 ± 2.5; AF: $\bar{x}$ 3.5 ± 2.6. General mental health (0–100) EF: $\bar{x}$ 50.8 ± 7.3; NF: $\bar{x}$ 50.7 ± 8.9; AF: $\bar{x}$ 49.5 ± 9.7.	The results showed that men experienced no greater depression or anxiety at the time of being a first-time expectant or new father, compared with levels prior to transition to fatherhood. In summary, this study finds first-time expectant and new fatherhood is not typically associated with increased levels of depression and anxiety
[100]	192 ♂ $\bar{x}$ = 35.04 yr ± 5.9	Cross-sectional. EPDS at 7 wk postpartum, SCID at 3 mo postpartum	54 ♂ met the SCID criteria for depression (19 with current depression, and 35 with a history of depression). ♂ with depression scored significantly ↑ on the EPDS ($\bar{x}$ = 14.79 ± 3.41) than non-depressed fathers ($\bar{x}$ = 6.64 ± 4.40)	Fathers with depression may be withdrawn while interacting with their infants and be less stimulating. They may adopt maladaptive patterns of parenting, thus potentially impairing their children’s development
[101]	N = 13,822	Longitudinal. Cohort study EPDS > 12 at 8 wk and 8 mo after birth (both ♀ ♂). Child outcomes Rutter revised preschool: 42 and 81 mo.	Family factors (maternal depression and couple conflict) mediated 2/3 of the overall association between paternal depression and child outcomes at 3.5 yr. Similar findings when children were 7 yr old. In contrast, family factors mediated less than 1/4 of the association between maternal depression and child outcomes. No evidence of moderation effects of either parental education or antisocial traits.	This study suggests that the association between depression in ♂ during postnatal period and subsequent child behavior is explained predominantly by the mediating role of family factors. In contrast, the association between depression in ♀ and child outcomes is only explained to a small degree by these wider family factors and is better explained by other factors, which might include direct effects of depression on mother-infant interaction
[102]	200 couples♀ $\bar{x}$ = 32.3 yr ± 4.2 ♂ $\bar{x}$ = 34.5 yr ± 4.7 T1 during pregnancy T2 6 mo postpartum	Longitudinal. GHQ, FSOC-S, SRRS, MOS-SSS, MOS-FMFM	Depressive symptoms, ♀ T1 15.5% T2 11.5%; ♂ T1 7.0% T2 10.5%; T1 ♀ > ♂. Predictor for depressive symptoms at postpartum, ♀ family sense of coherence, social support, and depressive symptoms during pregnancy changes in family sense of coherence and social support from pregnancy to 6 months postpartum; and partner’s depressive symptoms; ♂ family sense of coherence and depressive symptoms during pregnancy, changes in family sense of coherence from pregnancy to 6 months postpartum, and partner’s depressive symptoms were significant predictors of depressive symptoms at 6 months postpartum	Partner’s depressive symptoms significantly predict mothers’ ♂ depressive symptoms 6 months postpartum, suggesting that an increase in depression in one partner leads to increase in the other. Prenatal depression was a significant risk factor for postnatal depression among both parents. Social support was only found to predict depressive symptoms at postpartum among the mothers, but not the ♂. Mothers had a comparatively higher level of social support than the ♂ across the perinatal period indicating that social support acts as an important external coping resource that can alleviate depressive symptoms at postpartum. Both stress and family and marital functioning may be moderated and mediated by other factors to predict effect on depressive symptoms. Limitation: predominantly middle class sample
[103]	92 couples in prenatal period, 84 couples in postpartum period. ♂$\bar{x}$ = 33.72 yr ± 5.13 (23–52); ♀$\bar{x}$ = 29.95 yr ± 4.86 (20–43)	Cross-sectional. EPDS, MSPSS, RSE, WFCS, MAT.	Prenatal EPDS scores: ♀$\bar{x}$ = 7.54 ± 4.11, ♂$\bar{x}$ = 5.33 ± 3.12; Postpartum EPDS score: ♀$\bar{x}$ = 8.25 ± 3.29, ♂$\bar{x}$ = 6.52 ± 3.02	The paternal depression risk was measured as 4.3% in the prenatal period, and it was measured as 7.1% in the postpartum period. The EPDS average score was ↑ in the postpartum period. The results showed that low marital adjustment and work–family conflicts affect paternal depression risk. Also, ♀ depression risk did not affect ♂ depression risk, but↑ depression risk was found for ♂ that did not want the pregnancy
[104]	885 ♂ 926 ♀. ♂ $\bar{x}$ = 32.6 yr (20–51)	Cross-sectional. EPDS (cut off ≥ 12)	6.3% of ♂ scored 12 or more on EPDS and 9.1% scored zero. 12.0% of ♀ scored 12 or more on EPDS, and 4.8% scored zero. 1.5% of couples reported depressive symptoms in both ♀♂	The prevalence of depressive symptoms in ♀ seems to be approximately twice that in ♂. Depressive symptoms in one parent were associated with a ↑ risk of depressive symptoms.
[105]	711 couples. T1: 1 mo. postpartum; T2: 6 mo. postpartum; T3: 12 mo. postpartum.	Longitudinal. PSQI, EPDS	At T1, 6.8% of ♂ and 13.4% of ♀ had clinically significant PPD. At T2, 9.7% of ♂ and 13.4% of ♀ had clinically significant PPD. At T3, 9% of ♂ and 13.5% of ♀ had clinically significant PPD. Of those ♂ who reported clinically significant PPD at T1, 26% also had clinically significant PPD at T2 and 50% had clinically significant PPD at T3. Of those ♂ who reported clinically significant PPD at T2, 39% also had clinically significant PPD at T3.	For both partnered ♀ and ♂ and for single ♀, depressive symptoms at 1 month after the birth of a child was associated with poorer sleep at 6 months postpartum, which was in turn associated with more depressive symptoms at both 6 and 12 months postpartum.
[106]	181 couples (362 parents)	Longitudinal. T1: 3 mo. T2: 6 mo. EPDS ♀ ≥ 12 ♂ ≥ 8; STAI > 40; PSI-SF	Mothers reported higher scores on postpartum anxiety, depression, and parenting stress. The scores for all measures for both ♀♂ ↓ from T1 to T2. Parents’ own levels of anxiety and parenting stress and presence of depression in partner seem to directly influence the persistence of both ♀♂ postnatal depression	Both ♀♂ postpartum depression were influenced directly and indirectly by parents’ own anxiety levels and parenting stress as well as by the presence of depression in his/her partner. Although the two models are similar, they differ with respect to the role of parenting stress. The latter was shown to have an effect on maternal postpartum depression at 3 mo. postpartum, whereas it only influences paternal postpartum depression 6 mo. after the child’s birth
[107]	Cohort 1797 ♀ 1658 ♂. T1: 20th wk of gestation. T2: 4 mo postpartum	Longitudinal. EPDS (♀ prenatal cut off: ≥15, postnatal cut off: ≥13; ♂ cut off ≥10).	EPDS score T1: ♀$\bar{x}$ = 6.01 ± 4.1, ♂$\bar{x}$ = 4.01 ± 3.4; T2: ♀$\bar{x}$ = 5.41 ± 4.1, ♂$\bar{x}$ = 4.18 ± 3.6	The amount of maternal prenatal depressive symptoms predicted the amount of postnatal depressive symptoms. The depressive symptoms that started during the prenatal time have been reported to be related to especially severe postnatal symptoms. During the postnatal period, depressive symptoms have more direct and negative influence on mothers’ perceptions of themselves as caregivers and thereby also on their breastfeeding decisions
[108]	Japanese study. 270 ♂ during pregnancy(T0), 214 ♂ few days after birth (T1), 110 ♂ 2 wks after birth (T2), 193 ♂ a mo after birth (T3), 197 ♂ 2 mo after birth (T4), 187 ♂ 3 mo after birth (T5). ♂$\bar{x}$ = 32.0 ± 4.9	Longitudinal. EPDS cut off ≥ 8	Of the 215 fathers, 36 (16.7%) scored ≥ 8 on EPDS. T1: 8.7%; T2: 3.7%; T3: 6.3%; T4: 8.7%; T5: 7.0%	17% of fathers experienced postnatal depressive symptoms. 6.5% of ♂ reported depressivesymptoms at multiple assessment points after birth, suggesting that ♂ depressive symptoms are not transient but a continuing health problem over the medium to long term.
[109]	166 couples. ♂ age ≥ 30: 51.8%; ♂ age < 30: 48.2%; ♀ age ≥ 30: 30.7%; ♀ age < 30: 69.3%. T1: 3 d postpartum; T2: 2 wks postpartum; T3: 6 wks postpartum.	Longitudinal. EPDS, PSOC, KMSS	21.1% ♂ at T1, 20.4% at T2, 13.6% at T3 scored above the EPDS clinical cut-off point (9/10). 57.2% ♀ at T1, 52.6% at T2 and 26.2% at T3 suffered from symptoms of depression.	Almost one out of four fathers in our study experienced depressive symptoms. We discovered the occurrence of paternal PPD diminishes over the course of the 6-week post partum period. Fathers who were manifesting symptoms at 3 days resolved issues; perhaps, by adapting to parenthood, whereas first-time fathers who became depressed after 2 weeks and 6 weeks had, possibly, faced new challenges related to marital discord and parenting responsibilities.
[110]	1140 families (548 ♂)	Longitudinal. Using a 10-item version of the Hopkins Symptom Checklist (SCL-10), a subset of the 25-item version, which in turn is an abbreviated version of the original SCL-90. T1: 6 mo, T2: 12 mo, T3: 24 mo, T4: 36 mo, T5: 48 mo	This study compared the associations between ♂♀ depressive symptoms and early child behavior problems with CBCL/1½-5; SCL scores for Paternal depression at T1: $\bar{x}$ = 1.30 ± 0.37, T2: $\bar{x}$ = 1.22 ± 0.37, T3: $\bar{x}$ = 1.24 ± 0.35, T4: $\bar{x}$ = 1.28 ± 0.36, T5: $\bar{x}$ = 1.26 ± 0.39	Change in paternal depressive symptoms from the postpartum period through following years predicts child behavior problems
[111]	STEPS study. 873 families. T1: 20th gestational wk; T2: 4 mo postpartum.	Longitudinal. EPDS (for ♀ prenatal cut-off ≥ 15, postnatal cut-off ≥ 13 and for ♂ both time points cut-off ≥ 10) RDAS	♀ score on EPDS: T1: 829 ♀ $\bar{x}$ = 6.01 ± 4.1; T2: 802 ♀ $\bar{x}$ = 5.41 ± 4.1. ♂ score on EPDS: T1: 766 ♂ $\bar{x}$ = 4.01 ± 3.4; T2: 715 ♂ $\bar{x}$ = 4.18 ± 3.6	Maternal prenatal depression predicts maternal postnatal depression and paternal prenatal depression predicts paternal postnatal depression
[112]	80 cohabiting primipara couples between 28 wk gestation and delivery	Longitudinal. To describe course of depression in both ♀ and ♂ (and interrelation) from the 3rd trimester of pregnancy through 6th mo. CESD ≥ 16 ♀ EPDS ≥ 12 ♂ EPDS ≥ 10. BL 1, 3, 6 mo postpartum	Depression. Prenatal period: ♂ 20.5% ♀ 32.1%; 1 mo ↓ in both parents; 3 mo ♂↑, ♀↓; 6 mo. ♂↔, ♀↓ Average depressive symptoms severity declined linearly and similarly in both ♀ and ♂	For both ♀ ♂ symptoms severity ratings and classification were stable across time, with prenatal depression persisting through 6 mo. 75%♀ 86%♂. Prenatal depression in ♂ predict worsening depressive symptoms severity in ♀ across first 6 postpartum mo. but not viceversa.
[113]	533 couples	Longitudinal. EPDS; STAI at T1: early pregnancy (until 17 gestational we), T2:mid-pregnancy (17–29 gestational we), T3: late pregnancy (29 gestational we), T4:1 mo PP (T4), T5:6 mo PP	2.4% of ♀ and 0.8% of ♂ had a history of depression. Displays changes in ♂♀ depression during the perinatal periods according to the paternal smoking status at T1. At all 5 time-point measurements, smoking in the ♀ presence was found to have the highest level of perinatal depression in both genders, followed by smoking but not in the ♀ presence and nonsmoking. ♀depressive symptoms were relatively stable during pregnancy, increasing at 1 month after childbirth, and then beginning to decline sharply, regardless of the ♂smoking status. Paternal anxiety symptoms were relatively stable across the 5 assessment phases.	These findings imply a necessity to combine strategies for smoking cessation with interventions for affective disturbances in ♂. We also stress the importance of at least restricting the ♂smoking in the presence of the pregnant wife during perinatal periods if smoking cessation is tentatively unattainable.
[114]	71 ♀ 54 ♂. ♂$\bar{x}$ = 34 yr ± 5.73; ♀$\bar{x}$ = 32 yr ± 5.62	Cross-sectional. PSI-SF, MIRI, SSS, FSS, EPDS.	EPDS score: ♀$\bar{x}$ 6.03 ± 0.51; ♂$\bar{x}$ 4.4 ± 3.49.	↓ ♂, relative to ♀, reported postnatal depression symptoms. However, no association between ♂ and ♀ depressive symptoms was found.
[115]	276 ♀ 276 ♂. ♂$\bar{x}$ = 32.18 yr ± 6.21; ♀$\bar{x}$ = 28.94 yr ± 5.22.	Cross-sectional. EPDS, PBQ,	EPDS score: ♀$\bar{x}$ 7.14 ± 5.02; ♂$\bar{x}$ 4.27 ± 3.57. 15.9% of ♀ and 5.4% of ♂ scored > 12 and > 10 cut-off scores.	A significant number of parents reported high levels of depressive symptoms. Having a partner with an EPDS score above the cut-off significantly increased the risk for depressive symptoms in both ♀ and ♂.
[116]	145 couples. ♂$\bar{x}$ = 37.03 yr ± 5.89; ♀$\bar{x}$ = 35.12 yr ± 4.58.	Cross-sectional. K6, K6-FP, K10, K10-FP, PHQ, PHQ-9-FP.	K6 $\bar{x}$ = 1.21 ± 1.86; K6- FP, $\bar{x}$ = 1.55 ± 2.42; K10, $\bar{x}$ = 2.93 ± 4.20; K10-FP, $\bar{x}$ = 2.34 ± 3.25; PHQ, $\bar{x}$ = 2.21 ± 2.80; PHQ-9-FP, $\bar{x}$ = 1.65 ± 1.80.	K6 and K10 highly specific for detecting depression but poorly sensitive when used by female partners to assess partner’s mental status
[117]	19 ♂ $\bar{x}$ = 36 yr (25–52)	Cross-sectional. EPDS (cut-off ≥ 10), GMDS (cut-off ≥ 13).	EPDS score: $\bar{x}$ 12.5 (5–27). GMDS score: $\bar{x}$ 13 (3–32)	♂ experienced depression, reported loss of control and powerlessness, indicating difficulties with handling stress, feelings, and demands. ♂ with depression reported having no time to reflect over their situation, and that they found it difficult to cope with all conflicting demands and feelings of not being good enough, either at work or as a parent. Several ♂ reported that their partners were or had been depressed during pregnancy and/or postpartum
[118]	T1: 8th–14th wk of gestation; T2: 20th–24th wk of gestation; T3: 30th–34th wk of gestation; T4:1–3 d after birth; T5: 10th–14th wk after birth; T6: 24th–36th mo after birth. 258 couples completed all assessment waves until T5. 129 couple completed all assessment waves until T6. ♂ age ≤ 19 = 4.2%, 20–29 = 38.9%, 30–39 = 46.9%, ≥40 = 10%. ♀ age ≤ 19 = 11.5%, 20–29 = 47.0%, 30–39 = 39.2%, ≥40 = 2.3%	Longitudinal. STAI, EPDS.	Anxiety score ♂: T1: $\bar{x}$ 33.10 ± 8.83; T2: $\bar{x}$ 31.17 ± 8.07; T3: $\bar{x}$ 32.61 ± 7.95; T4: $\bar{x}$ 32.72 ± 9.13; T5: $\bar{x}$ 29.37 ± 7.99; T6: $\bar{x}$ 33.77 ± 9.43. Anxiety score ♀: T1: $\bar{x}$ 36.04 ± 8.88; T2: $\bar{x}$ 34.68 ± 9.37; T3: $\bar{x}$ 36.24 ± 9.03; T4: $\bar{x}$ 35.76 ± 10.25; T5: $\bar{x}$ 30.84 ± 8.36; T6: $\bar{x}$ 34.45 ± 9.48. Depression score ♂: T1: $\bar{x}$ 5.24 ± 3.96; T2: $\bar{x}$ 4.18 ± 3.31; T3: $\bar{x}$ 4.08 ± 3.26; T4: $\bar{x}$ 4.06 ± 3.73; T5: $\bar{x}$ 3.51 ± 3.46; T6: $\bar{x}$ 4.80 ± 4.21. Depression score ♀: T1: $\bar{x}$ 6.42 ± 4.19; T2: $\bar{x}$ 6.00 ± 4.32; T3: $\bar{x}$ 5.49 ± 4.25; T4: $\bar{x}$ 5.87 ± 4.67; T5: $\bar{x}$ 4.72 ± 4.21; T6: $\bar{x}$ 6.11 ± 4.36.	From T1 to T5, anxiety and depression symptoms ↓. From T5 to T6, anxiety and depression symptoms ↑. At T1, ♀ had more anxiety anddepression symptoms than did ♂. No other gender effects were found for time regarding both symptoms.
[119]	Data from a parent-child cohort of a prospective longitudinal study. 700 ♀, 646 ♂, and 583 couples. ♂$\bar{x}$ = 34.6 yr; ♀$\bar{x}$ = 32.3 yr.	Longitudinal. EPDS (cut off ≥ 12), SPSQ.	EPDS score 25 mo after birth: 688 ♂$\bar{x}$ = 5.0 ± 3.6; 623 ♀$\bar{x}$ = 6.0 ± 4.1.	> 11% of ♀ and ≈5% of ♂ reported depressive symptoms and that ≈16% of children had at least one parent with depression. ≈8% of both sexes reported feelings of incompetence with rather strong associations with depressive symptoms for both ♀ and ♂. The reason may be that some of the questions in the incompetence subscale identify feelings that can be associated with depression
[120]	438 ♂ $\bar{x}$ = 34.1 yr ± 6.7	Cross-sectional. BDI-II (≥14 mild/major; ≥20 moderate/major; ≥29 severe/major), EPDS (cut off ≥ 12), GMDS	BDI-II score: $\bar{x}$ = 10.20 ± 8.70. 27.85% mild depression, 13.93% moderate depression, 3.10% severe depression. EPDS score: $\bar{x}$ = 7.30 ± 4.90	The items measuring depressive equivalents and those measuring traditional depressive symptoms correlated highly witheach other. This finding highlights the coexistence of traditional depressive symptoms and depressive equivalents in ♂. 28% of ♂ in the present study reported depressive symptoms above the BDI-II cut-off for mild depression, almost 14% above the BDI-II cut-off for moderate depression.
[121]	174 ♂ $\bar{x}$ = 34.6 yr ± 4.5	Cross-sectional. PHQ-2 (depression range > 2), PSS-4 (stress range > 6).	PHQ-2: $\bar{x}$ = 0.61 ± 1.0; 16.7% score > 2. PSS-4: $\bar{x}$ = 4.2 ± 2.6; 26.3% score > 6	Fathers with elevated psychological distress were more likelyto endorse reluctance to talk to others about their moods oranxieties, and reluctance from family or friends to talk aboutemotional aspects of pregnancy and the postpartum period asbarriers to seeking help to improve mental health duringpregnancy and following the baby’s birth.
[122]	22 ♀; 19 ♂ but only 2 ♂ attended the focus group. ♀$\bar{x}$ = 31.3 yr ± 4.70	Longitudinal. Objectives were: explore the needs of ♀ in the year following childbirth; to compare these needs between ♀ who did not have a psychological disorder and who lived a PPD; and to compare the needs expressed by ♀ with the perception of professionals and fathers about the mothers’ needs.EPDS T1: 1 mo PP; T2: 6 mo PP; T3: 1 yr PP	EPDS score: T1: 11.6 ± 6.61; T2: 10.0 ± 5.02; T3: 7.00 ± 3.74 (The results concerning the comparison of the ♀ needs and the ♂ perception about the ♀ needs should therefore be interpreted with cautious, given the low number of ♂ who attended)	♂ play a central role in the psychological wellbeing of the ♀. Some couples had arguments because of the new organization. Many ♀felt supported by their partner and said it was great to chat with them. Other ♀ would have liked them to be more considerate and present. For ♂, postnatal period appeared to be punctuated by the baby’s rhythm; while for ♀, it also seemed to be marked by their own pace. ♂ understood mothers’ need for sharing and for psychosocial support and thus were aware of the importance of continuing to have a social and family life. For them, are in the same boat which strengthens the ♀♂ complicity of the couple; this did not seem to be always true for ♀.
[123]	196 ♂ $\bar{x}$ = 32.0 yr ± 5.0	Longitudinal. ♂ depression in the prenatal and early postnatal period. EPDS: T1: 20 we gestation; T2: the first few days; T3:1 mo PP; T4: 2 mo PP. Also investigated the association of ♂ depression in the prenatal and early postnatal period with child maltreatment tendency at 2 mo PP with CMS	EPDS score: T1:9.7%; T2:8.0%; T3:6.6; T4:8.8. On the CMS, 36♂ (18.4%) met criteria for child maltreatment tendency. The impact of ♂ depression on child maltreatment tendency changed depending on the timing of the depressive episode’s onset	Current ♂ depression in the early postnatal period was associated with child maltreatment tendency at 2 mo PP. To prevent childmaltreatment, early detection of ♂ depression and an appropriate initial response by healthcare providers is exceedingly important
[124]	149 couples	Longitudinal. A study to link ♂ Tes and ♀ and ♂ depressive symptoms and intimate partner aggression following the bith of a child. T1 = 1–3 mo. postpartum, T2 = 6–9 mo., T3 = 12–16 mo. EPDS (T1,T2,T3), DAS (T2), PSI (T3), and salivary Tes levels in ♂, sampled 3 times over 1 day (T2)	♂with lower aggregate Tes reported more depressive symptoms at T1 and T2. ♀ whose partner had higher evening Tes reported more depressive symptoms at T2 and T3. Higher ♂ Tes and ♂ depressive symptoms at T2 each independently predicted greater fathering stress at T3. Higher ♂ Tes predicted more ♀-reported intimate partner aggression at T3.	In addition to linear relationship between Tes and depression, curvilinear relationship emerged such that ♂ with both low and high Tes at T2 reported more subsequent (T3) depressive symptoms and fathering stress. Higher paternal Tes may protect against paternal depression. It seems contribute to maternal distress and suboptimal family outcomes. Intervention that supplement or alter ♂’s Tes may have unintended consequences for family well-being.
[125]	140 couples; ♀ $\bar{x}$ = 33.9 yr ± 4.6; $\bar{x}$ = 36.4 yr ± 5.7	Longitudinal. EPDS; IES-r	EPDS at T1 (3rd trimester): $\bar{x}$ = 2.89 ± 2.7; T2 (2–5 days PP): $\bar{x}$ = 97.97 ± 14.8; T3(4 we PP): $\bar{x}$ = 2.49 ± 2.3; IES-r $\bar{x}$ = 8.41 ± 6.6. The present study indicated no relationship between ♀♂ depressive symptoms during pregnancy; however, their depressive symptoms were moderately related 4 we after childbirth.	Higher occurrence of depressive symptoms in the last trimester of pregnancy is predictive of ♂depressive symptoms and acute stress in the fourth we PP. ♂ Depression during partners’ pregnancy predicts their subjective birth experience. ♂ subjective birth experience, in turn, predicts their depressive symptoms and their acute stress reaction in the fourth we after childbirth
[126]	140 ♂ $\bar{x}$ = 33.40 yr ± 4.45	Cross-sectional. CES-D cut-off score of ≥16; EPDS-P. The survey included measures assessing demographic information, depressive symptoms, relationship satisfaction, and social support. Participants were provided with descriptions of three commonly used evidence-based treatment methods for depression.	31 (22.1%) ♂ met the cut-off score of 16 on the CES-D. 18 (12.9%) ♂ reported a history of seeking or receiving treatment for depression or another psychological disorder, while 21 (15.0%) ♂ indicated previous treatment for another reason (e.g., relationship distress). Participants reported preferring individual and couple psychotherapy to pharmacotherapy for treatment of PPD.	♂ prefer psychological interventions over drug treatment for PPD
[127]	3523 ♂	Longitudinal cohort study to evaluate paternal depression during pregnancy and postpartum. (EPDS, PHQ)	3rd trimester 2% EPDS ≥ 12. 9 mo. after birth 4% PHQ. Association between postnatal ♂ depression	Expectant fathers at risk of depression when perceiving stress or reporting poor health. Depression higher during postpartum period and associated with adverse social and relationship factors
[128]	125 participants randomly divided into 2 groups having 63 participants each. One receiving 2 weekly group lifestyle-based training session (lasting 60–90 min) and a control group.	Longitudinal. EPDS; STAI T1: gestational ages of 24–28 we before the intervention; T2: 8 we after; T3: 6 we after childbirth	Compared with the control group, there was a significant decrease in depression (adjusted difference: −1.6; 95% CI −2.8 to −0.5), state anxiety (−5.7; −8.6 to −2.9) and trait anxiety (−5.0; −7.8 to −2.2) scores at 8 weeks after intervention as well as postnatal depression (−3.3; −5.0 to −1.5); postnatal state anxiety (−7.5; −11.6 to −3.4), and postnatal trait anxiety (−8.3; −12.2 to −4.4) in the intervention group.	Positive impact of training on prenatal and postnatal depression and anxiety in fathers
[129]	126 1st time ♂	Longitudinal. Study to evaluate PPD trajectory in Italy. T1 = 7–8 th mo., T2 = 40 days after delivery, T3 = 5–6 th mo. after delivery, T4 = 12th mo. after delivery.	3 heterogeneous subpopulations of ♂: 1. Low and stable levels of depressive symptoms at all 4 time points (resilient 52%) 2. Moderate depressive symptoms that marginally decrease from pregnancy to 5–6 mo. postpartum and then went back to bl levels after 1 yr postpartum (distress 37%) 3. Relatively consistent high levels of depressive symptoms reaching the threshold for clinical depression 1 yr (emergent depression 11%)	Depression rate: ↓ from prenatal period to the 1st mo. after child birth, ↑ over the course of the 1st yr. For ♂ early pregnancy seems to be the most stressful period during the transition to parenthood. Longitudinal patterns of psychopathological symptoms ♂ ≠ ♀. EPDS scores during pregnancy were predictors of EPDS scores after pregnancy.
[130]	T1 (prenatal): 1043 ♀ 31 yr ± 4.0; 1043 ♂ 33.8 yr ± 4.6; T2 (post-partum): 849 ♀; 747♂	Longitudinal. Identification of predictors of PPD symptoms in both ♀ and ♂. EPDS at T1 (prenatal); T2 (post-partum); EPDS cut-off ≥ 10	EPDS score at T1: 15.2%♀, 13.6♂; T2: 11.8% ♀; 12.1% ♂ 15.2% of mothers scored above the EPDS ≥ 10 cutoff prenatally, while 14% of fathers scored above the EPDS ≥ 9 cutoff	Predictors of high PPD symptoms in ♀ included low household income, high prenatal depressive symptoms and postnatally, low social support (from fathers, presumably) and higher number of stressful life events experienced. For ♂ similar predictors were identified, including low household income, high prenatal depressive symptoms, and low postnatal social support (from mothers, presumably) and postnatal smoking
[131]	306 ♂ $\bar{x}$ = 27.80 yr ± 6.95	Longitudinal. Possibly/Probably Depressed EPDS > 9 and Nonsymptomatic Fathers EPDS ≤ 9. T1: interview at 1 mo. after birth; T3: 12 mo. after the child’s birth.	♂ reported low mean levels of depressive symptoms overall at both T1 $\bar{x}$ = 4.15 ± 4.26 and T3 $\bar{x}$ = 3.88 ± 4.02. At both T1 and T3, 11% of the sample was above cutoffs for probable or possible depression and 89% of the sample was under the clinically suggestive cutoff of 9 on the EPDS. There was substantial range and variability in depressive symptoms, with 11% of the sample possibly or probably depressed at each time point. Very few ♂ screened high at both time points; only one ♂ had possible depression at both T1 and T3, and six ♂ had probable depression at T1 and T3	Fathers display low average depressive symptoms, fairly low perceived stress
[132]	174 ♂ $\bar{x}$ = 34.6 ± 4.5	Cross-sectional. PHQ (was used to assess Depressive symptomatology); PSS (was used to assess perceived stress associated with daily life situations).	PHQ score: $\bar{x}$ = 0.61 ± 1.0 with 16.7% (29/174) of participants scoring in the depressed range (score > 2). PSS score: $\bar{x}$ = 4.2 ± 2.6, with 26.3% (46/174) of men scoring in the top quartile (score > 6) on this stress scale. Self-reported diagnosis of any psychiatric or psychological disorder was 6.9% (12/174). Prior treatment for an emotional problem was reported by 12.6% (22/174) of the sample, with psychotherapy (18/22, 81.8%) and medication (13/22, 59.1%) found as the two most frequently used modalities	During the perinatal period, ♂ claim Internet-delivered information related to parenting, supporting their partner, and optimizing their emotional adjustment during the transition to parenthood. Gender-tailored elements to reduce stigma and overcome barriers to seeking and accepting help are also important
[133]	T1:194 ♀ $\bar{x}$ = 34 yr; 186 ♂ $\bar{x}$ = 35 yr; but only 172 participants also answered the questionnaire at T2. Participants were divided into two groups: Group A: 145 at T1 and 74 at T2; Group B: 225 at T1 and 98 at T2, according to information provision, more or less distinct two-step processes	Longitudinal. HADS to assess psychological distress in expectant parents, before and after receiving information about PND. Comparison of Groups A and B (two different procedures of information giving), psychological distress, and levels of anxiety and depression	HADS: (T1) participants in group A had a higher rate of health-related worry. Differences in the other dimensions of psychological distress were not observed between participants from group A and group B at T1. General anxiety decreased among the participants in group A but was unchanged in group B. The rate of depressive symptoms was unchanged over time in both groups. At T1 ♀ had a higher degree of depressive symptoms than ♂and 2.6% ♀vs. 0.5% ♂ were categorized as clinical cases. These rates were unchanged at T2	More distinct two-stage process (more time and information, 30–45 minutes) reduced anxiety (group A), while less distinct two-stage process (less time and information, 10–15 minutes, group B) left it unchanged. ♀ more worried and depressed than ♂
[134]	150 couples ♀ $\bar{x}$ = 28.01 yr ± 3.16; ♂ $\bar{x}$ = 29.87 ± 3.96	Longitudinal. Couples agreed to take part in a year-long, Longitudinal study in which they would be interviewed at four time points: T1: 3rd trimester of pregnancy; T2: 1 mo PP; T3: 4 mo PP; T4: 9 mo PP. PPD is misured with PDSS	For both ♂♀ PPD scores were significantly lower at 4 mo PP in comparison to their depression scores at 1-month PP. Co-parent depression was also explored as a potential control variable; analyses indicated co parent depression did not significantly predict PPD for ♀or for ♂ at either 1 mo or 4 mo PP	Feeling self-efficacy as a parent (parenting efficacy) is linked to numerous positive outcomes for new parents. Conversely, perceived inability to meet expectations is associated with negative mental health consequences for ♀ and ♂. Parenting efficacy is negatively associated with PPD for both ♀ and ♂ throughout the transition period. ♀ and ♂ whose parenting efficacy experiences were more negative than expected reported higher levels of PPD at 1 mo PP. This effect dissipates for ♀ but not ♂, by 4 mo PP, suggesting differences in the experiences of ♀ and ♂ during this transition
[135]	3202 ♀ $\bar{x}$ = 30.92 ± 4.56 2076 ♂ $\bar{x}$ = 32.60 ± 5.33	Longitudinal. EPDS, SCL-90: T1: 14 gwks; T2: 24 gwks; T3: 34 gwks	EPDS: T1 $\bar{x}$ = 5.17 ± 4.02; T2 $\bar{x}$ = 5.00 ± 4.12; T3 $\bar{x}$ = 4.91 ± 4.10; SCL-90: T1 $\bar{x}$ = 3.31 ± 3.91; T2: $\bar{x}$ = 3.92 ± 4.26; T3 $\bar{x}$ = 3.24 ± 4.00	Approximately 85–90% of the parents in the FinnBrain Cohort reported low levels of depressive or anxiety symptoms at any trimester. This is in line with several previous studies showing that most parents in the general population report only minimal or mild symptoms during pregnancy and the PP. ♀ with low levels of depressive or anxiety symptoms were more often living with the ♂of the child, smoked less frequently, and were expecting their first child more often than ♀ with high or changing levels of depressive or anxiety symptoms.
[136]	T1 Antenatal period: 480 Couples; T2 6 we Post-partum :255 Couples; 6 mo Post-partum:121 Couples	Longitudinal. EPDS cutoff ≥ 12	There were no significant differences on demographic characteristics between participants who completed all three questionnaires and those who dropped out at T2 or between those who completed all three questionnaires and those who dropped out at T3. There was significant difference on prenatal depressive symptoms between ♂who completed all 3 questionnaires $\bar{x}$ = 4.74 ± 2.97 and ♂ who dropped out at T2 $\bar{x}$ = 5.53 ± 3.88. No difference on prenatal depressive symptoms was found for ♂ who completed all 3 questionnaires and those who dropped out at T3 $\bar{x}$ = 5.32 ± 3.70. At T2, ♂ PPD symptoms were associated with ♀ PPD symptoms.	Effective assessment and interventions targeted at preventing or identifying and reducing ♂ PPD and improving father-infant relationship would help to lower the risk of infant disorders and poor development. Strategies improving the ♂ mental health during antenatal period and their partner’s psychosocial well-being may also reduce paternal postpartum depression.
[137]	580♀ $\bar{x}$ = 32.77 yr ± 5.11; 385 ♂ $\bar{x}$ = 35.14 yr ± 5.07	Cross-sectional. PSS; EPDS ≥ 10. The main objective was to analyze the relationship between parental perception of child vulnerability in infants and perception of parental competence, considering the role that PPD and parental stress exert in that relationship.	PPD ♂ $\bar{x}$ = 4.54 ± 3.89; ♀ $\bar{x}$ = 5.86 ± 4.77. Regarding postnatal depression, both globally and differentiating between ♀♂ the averages were below the reference cutoff point for this variable. So this population is predominantly healthy in terms of PPD the results revealed an association between parental perception of their child’s vulnerability and parent’s perception of parental competence through depression and parental stress.	The relationship between parental perception of child vulnerability in infants and perception of parental competence is linked to the association between parental perception of child vulnerability and increased PPD and experienced stress
[138]	T1(prenatal): 420 participants; T2 (8 we postnatal): 403 participants	Longitudinal. STAI-S, EPDS ≥ 10	23.3% (94 persons) of♂ showed a high level of prenatal depressive symptoms.19.6% (79 persons) of ♂ showed a high level of postnatal depressive symptom. The study showed that the rate of depressive symptoms among expectant ♂ decreased across 8 we postpartum compared with the 3 trimesters	A high level of prenatal depressive symptoms is a risk factor for a high level of PDS.
[139]	3425 ♂ $\bar{x}$ = 30.6 yr ± 8.0	Cross-sectional. EPDS ≥ 10	EPDS analyses reveal that 9.1% of these Jamaican ♂ had EPDS scores of 10 or higher (possible depression). Measures of social support were associated with higher EPDS scores. ♂ age was weakly negatively predictive of EPDS scores.	Educational attainment was not related to EPDS scores, though higher indices of material wealth were weakly, negatively related to EPDS scores. Whereas relationship status was unrelated to depressive symptoms, relationship quality negatively predicted depressive symptoms
[140]	♂ 806	Longitudinal. EPDS-3 every 90 days during the child’s first 15 months of life	4.4% of fathers screened positive for depression.	Fathers comprised 11.7% of the proportion of parents who screened positive for depression.
[141]	26 couples who conceived with OD 52 controls who conceived with IVF/ICSI52 controls who conceived naturally (NC)	Longitudinal. GHQ-36T1: wks 18–20; sociodemographic informations, GHQ-36T2: 2 months postpartum; Information on pregnancy and delivery, perinatal data, GHQ-36T3: 12 months postpartum, GHQ-36	T1 = 4.2%♂ clinically significant mental health problemsT2 = 2.6%♂ clinically significant mental health problemsT3 = 5.3%♂ clinically significant mental health problems	Mental health problems of oocyte donation fathers did not differ from those of IVF/ICSI and NC control fathers at T1-T3
[142]	♀ 911 (expectant mothers) $\bar{x}$ age: 29.3. ♂ 587 (expectant fathers) $\bar{x}$ age: 32.3	Longitudinal. T0: 3rd trimester of pregnancy, T1: birth. T2: 2–3 months postpartum T3: 6–8 months postpartum. Sociodemographic information, EPDS, DASS-21	EPDS ≥ 10: 12.2% ♀ (expectant mothers); 8.4% ♂ (expectant fathers), DASS-21 ≥ 8: 28.8% ♀ (expectant mothers) 13.3% ♂ (expectant fathers)	Couples having perceived social/family support were less likely to suffer from depression or anxiety. Intimate partner violence, poor relationship with partner, partners’ depression in earlier pregnancy and/or current pregnancy, living in rented house, wife’s depression in current pregnancy in expectant fathers were associated with a greater likelihood of anxiety.
[143]	129 couples	Longitudinal. T0: 8–14 wks gestation T1: 20–24 wks gestation T2: 30–34 wks gestation T3: 1–3 days postpartum T4: 3 months postpartum T5: 30 months postpartum. Sociodemographic informations, RQ (positive and negative), STAI-S, EPDS	T0 ♂: RQp $\bar{x}$ 3.64, RQn $\bar{x}$ 2.19, STAI-S $\bar{x}$ 33.10, EPDS $\bar{x}$ 5.24 T1 ♂: RQp $\bar{x}$ 3.65, RQn $\bar{x}$ 2.11, STAI-S $\bar{x}$ 31.17, EPDS $\bar{x}$ 4.18 T2 ♂: RQp $\bar{x}$ 3.65, RQn $\bar{x}$ 2.08, STAI-S $\bar{x}$ 32.61, EPDS $\bar{x}$ 4.08 T3 ♂: RQp $\bar{x}$ 3.66, RQn $\bar{x}$ 2.06, STAI-S $\bar{x}$ 32.62, EPDS $\bar{x}$ 4.06 T4 ♂: RQp $\bar{x}$ 3.66, RQn $\bar{x}$ 2.08, STAI-S $\bar{x}$ 29.37, EPDS $\bar{x}$ 3.51 T5 ♂: RQp $\bar{x}$ 3.44, RQn $\bar{x}$ 2.13, STAI-S $\bar{x}$ 33.77, EPDS $\bar{x}$ 4.80	Negative interaction moderated depression from 3- to 30-months postpartum. Couples with high negative interaction scores experienced a steeper increase in depression from 3- to 30-months postpartum. Fathers with low positive interaction scores experienced an increase in anxiety, fathers with high positive interaction scores and mothers with high or low positive interaction scores did not experience changes in anxiety from 3- to 30-months postpartum
[144]	♂ 51 ♀ 51 Sample exposed to antenatal trauma	Cross-sectional. Sociodemographic informations, EPDS, TSC-40, FFMQ	32.9% EPDS ≥ 9; 16.5% EPDS ≥ 12	Mindfulness and trauma symptoms are significant predictors of depressive symptoms in future parents. Those with lower mindfulness exhibited higher levels of depression
[145]	41 couples expecting twins 25 conceived spontaneously (CS); 16 conceived by assisted reproduction techniques (ART)	Longitudinal. T1: 13 wks of pregnancy T2: 21 wks of pregnancy T3: 30 wks of pregnancy T4: 1 wks after birth T5: 4 wks after birth T6: 8 wks after birth Socio-demographic questionnaire, marital relationship, attitudes towards sex, pregnancy and the baby, STAI, EPDS	T1 ♂: EPDS $\bar{x}$ 4.6, STAI-S $\bar{x}$ 34.6 T2 ♂: EPDS $\bar{x}$ 4.1, STAI-S $\bar{x}$ 32.0 T3 ♂: EPDS $\bar{x}$ 4.6, STAI-S $\bar{x}$ 33.6 T4 ♂: EPDS $\bar{x}$ 4.6, STAI-S $\bar{x}$ 35.5 T5♂: EPDS $\bar{x}$ 4.0, STAI-S $\bar{x}$ 34.0 T6 ♂: EPDS $\bar{x}$ 3.6, STAI-S $\bar{x}$ 32.1	ART parents may experience more psychological difficulties during the transition to twin parenthood than SC parents. ART mothers appear to be more at risk of high levels of postpartum depressive symptoms
[146]	♂ 3656; $\bar{x}$ age: 35.4	Cross-sectional. 3–6 months after childbirth: sociodemographic information, EPDS, GMDS. The study compared EPDS score ≥ 10 (mD) and ≥ 12 (MD), in relation to GMDS score ≥13	EPDS ≥ 10: 13.3% EPDS ≥ 12: 8.1% GMDS ≥ 13: 8.6% 5.2% of the fathers are not detected if score 12 or higher is used for the EPDS	Significant association between possible depression and low income, low education, being born in a foreign country, having three or more children. No associations regarding paternal age
[147]	♂ 100 $\bar{x}$ age:	Longitudinal. Sociodemographic informations, EPDS	EPDS ≥ 12: 12% EPDS ≥ 9: 28% Depression was most prevalent in the first 6 months postpartum (EPDS ≥ 12 = 16.4%; EPDS ≥ 9 = 32.7%) compared to the second 6 months (EPDS ≥ 12 = 6.7%; EPDS ≥ 9 22.2%)	Risk factors: having an infant with sleep problems, a previous history of depression, a lack of social support, poor economic circumstances, not having paternity leave and not being married
[148]	Impact of maternal nausea and vomiting (NVP) in pregnancy on expectant fathers 300 ♂ $\bar{x}$ age: 30.5	Cross-sectional. 3rd trimester of pregnancy: sociodemographic informations, HADS	HADS ≥ 8 depression: No NVP 6%; Mild NVP 8%; Moderate NVP 10%; Severe NVP 12% HADS ≥ 8 anxiety: No NVP 17%; Mild NVP 29%; Moderate NVP 34%; Severe NVP 67%	No correlation between paternal depression and maternal NVP, but a significant association was found between moderate and severe maternal NVP and paternal anxiety
[149]	31 fathers of preterm newborns	Cross-sectional. PSI, STAI-S, EPDS at 6–8 wks after birth of their infant	PSI$\bar{x}$ 10.0	Fathers exhibited low life stress, only one father experienced depressive symptoms
[150]	300 Australian ♂ of infants between 1 and 12 mo.	Cross-sectional. PSOC, EPDS, m-PNTQ	m-PNTQ scores: $\bar{x}$ 106.1 ± 20.6. PSOC$\bar{x}$ 70.1 ± 10.4. EPDS$\bar{x}$ 5.6 ± 4.2 PSOC score cutoffs arbitrarily chosen. PSOC 70–96 high, 51–69 moderate, 16–50 low; EPDS cut-off 12 or more to detect probable depression	Adopted cutoffs arbitrary, not commonly accepted. Postnatal negative thoughts are common in fathers during the first-year post-birth
[151]	70 couples	Longitudinal. T0 (3rd trimester): sociodemographic information, EPDS, K10, MSSST1 (12 months follow-up): EPDS, K10, MSSS	T0: ♂ EPDS $\bar{x}$ 3.47, ♀ EPDS $\bar{x}$ 5.17; ♂ K10 ♀ 14.0, ♀ K10 $\bar{x}$ 15.2; ♂ MSSS $\bar{x}$ 27.3, ♀ MSSS $\bar{x}$ 27.3.T1: ♂ EPDS $\bar{x}$ 3.61, ♀ EPDS $\bar{x}$ 4.62; ♂ K10 ♀ 14.2, ♀ K10 $\bar{x}$ 13.9; ♂ MSSS $\bar{x}$ 27.1, ♀ MSSS $\bar{x}$ 27.6	Fathers reported fewer depressive symptoms and psychological distress than mothers. ↑ levels of perceived social support and satisfaction in marital/partner relationship, combined with the correlated high socio-economic factors are protective for the onset of depression and anxiety in peripartum for both ♂ and ♀
[152]	146 primiparous and 105 first time ♂. ♂$\bar{x}$ = 35.59 yr ± 5.97 ♀$\bar{x}$ = 32.76 yr ± 4.97	Cross-sectional. 2nd trimester; Psychosocial Risk Interview, EPDS, MSPSS, STAI-T	EPDS ≥ 9, 30.87% ; ♀ $\bar{x}$ = 12.22 ± 3.02 (range 9–21); 17.1 ♂$\bar{x}$ = 10.94 ± 2.16 (range 9–15)	Depressive symptoms during pregnancy more represented in ♀ due to possible underestimation of paternal depressive condition. ♂ display different depressive symptoms from ♀. Risk factor related to pregnancy, low social support, personal psychiatric risk factors, negative life events would have predicted depressive symptomatology during pregnancy in both ♀ and ♂. Marital dissatisfaction, personal history of depression and personal trait anxiety associated with ↑ depression during pregnancy. No association between SES and depressive symptoms during pregnancy for ♂
[153]	52 couples with SGA fetuses (case group) $\bar{x}$ age: 33.48 ± 4.98 68 parents with AGA, fetuses (comparison group) $\bar{x}$ age: 32.84	Cross-sectional. Last trimester of pregnancy: EPDS, PSS, PRAS, CD-RISC	♀ EPDS $\bar{x}$ = 7.12 ♂EPDS $\bar{x}$ : 4.98 ♀ PSS $\bar{x}$ : 26.83; ♂PSS $\bar{x}$ : 24.02; ♀ PRAS $\bar{x}$ : 23.55; ♂PRAS $\bar{x}$ : 19.11; ♀ CD-RISC $\bar{x}$ : 28.11; ♂CD-RISC $\bar{x}$ : 32.85	Mothers obtained ↑scores than fathers on psychological distress but ↓ for resilience. Mothers of SGA newborns were more distressed than the other groups but there were no differences between fathers of SGA *vs*. AGA newborns
[154]	N = 487 fathers completing the 2 mo postpartum follow-up $\bar{x}$ age: 34.27	Cross-sectional. A prospective cohort study with 622 men who completed sociodemographic and psychosocial questionnaires during their partner’s third trimester of pregnancy. 487/622 (78.3%) and 375/622 (60.3%) completed the questionnaires at 2 and 6 months postpartum, respectively. Measures: EPDS (cutoff of ≥10), PSQI, the Dyadic Adjustment Scale (DAS), the Modified MOS Social Support Survey was used to assess social support; Financial stress was measured with four items; The Parenting Stress Scale (PSS)	The prevalence of depression in fathers was 13.76% at 2 mo and 13.60% at 6 mo postpartum. Men with depression during their partner’s pregnancy were 7 times more likely to have depression 2 mo postpartum. Depression at both antenatal and 2 mo postpartum assessment was associated with increased risk of depression at 6 mo postpartum. Older age, poor sleep quality at study entry, worse couple adjustment, having a partner with antenatal depression, and elevated parental stress were associated with depressive symptoms at 2 mo postpartum. Poor sleep quality, financial stress and a decline in couple adjustment were independently associated to depressive symptoms at 6 mo postpartum	The psychosocial risk factors identified provide opportunities for early screening and targeted prevention strategies for fathers at risk for depression during the transition to parenthood
[155]	290 ♂ $\bar{x}$ = 34.97 yr ± 8.56. 347 ♂ invited to participate. 57 (16.42%) excluded due to incomplete EPDS or had a history of psychiatric illness	Cross-sectional. EPDS cutoff score ≥9	Of the 290 participants, 98 had depression according to the EPDS cutoff (33.79%). 72 (24.8%) ♂ were invited to undergo a structured interview with a psychologist, of whom 57 (79.16%) consented and agreed. 15 (20.83%) who did not agree to participate in the diagnostic interview did not differ from participants	A substantial proportion of expecting fathers have perinatal depression
[156]	532 prospective fathers/partners (529 ♂; 3 ♀); $\bar{x}$ age: 31.5 yr ± 5.47, range 17–54 yr	Cross-sectional. Sense of Coherence (SOC-13), lifestyle variables (BMI and AUDIT), smoking and use of snuff, use of alcohol, physical activity, sleeping difficulties and sexual satisfaction, SF-36, STAI, EPDS	9.8% of fathers bore high risk for depression; unemployment, financial distress, ‘very or fairly bad’ health or sexual satisfaction, low score on the SOC scale were risk factors. Fathers with depression smoked more often, made hazardous use of alcohol, and slept with difficulty. Difficult sleepers 5.7 times more likely to have several depressive symptoms. Hazardous users of alcohol and smokers had respectively 3.1 and 3.0-fold higher risk for depression. The single strongest risk factor was a low score on the SOC scale (10.6 higher risk for depression). High state anxiety in 8.9% and trait anxiety in 7.9%, difficult sleepers reported ‘very or fairly bad’ health	Data collected 5 years prior to publication. Depression in expecting fathers is correlated with maladaptive lifestyle, unsatisfying life, bad sleep, ill health and anxiety; given the cross-sectional nature, it is not possible to make causal relationship conjectures
[157]	70 fathers; $\bar{x}$ age: 37.01	Cross-sectional. 3 mo postpartum, mothers and fathers completed self- report questionnaires to evaluate symptoms of depression (EPDS) and anxiety (STAI-Y), and a form to gather socio-demographic data. At 3 mo, mother-infant couples were video-recorded and evaluated with the Child-Adult Relationship Experimental Index (CARE-INDEX)	2.9% of the fathers had scores exceeding the clinical cutoff (≥12) for depression; 21.7% of fathers had scored high on state anxiety (cutoff ≥ 39), and 11.6% of fathers had scored high on trait anxiety (cutoff ≥ 42). There were also significant associations between paternal depression and anxiety and mother-infant interaction style	Paternal anxiety and depression affect the quality of mother-infant interaction
[158]	531 fathers: 317 first time, 214 experienced. (<30 yr 18.4% and 6.7%; 30–35 yr 47.6% and 41.9%; ≥35 yr 34.0% and 51.4%)	Longitudinal. Mothers and fathers were evaluated at baseline during early pregnancy (T1), the second trimester (T2), and the third trimester (T3) as well as at 1 month (T4), 6 months (T5), and 1 year (T6) postpartum.531 couples (n = 1062) successfully completed all 6 assessments until 1 year postpartum. Parameters assessed: EPDS, STAI, WHOQOL-BREF	Post hoc contrasts for depression scores showed in fathers significant ↑ from T4 to T5 and ↓ from T2 to T3. Significant changes in anxiety symptoms from T1 to T6 in both mothers and fathers. Among men, increasing levels of depression and decreasing scores in the HRQoL social relations domain were observed during the first year after childbirth. ↑ depression and anxiety risks and ↓ HRQoL physical health and social relations domain scores throughout the perinatal period: > experienced than first-time fathers	Transition to parenthood, even in experienced fathers, is fraught with psychological difficulties throughout the perinatal period. Paternal depression is inversely correlated with QoL
[159]	331 ♂; ♂ age $\bar{x}$ > ♀	Longitudinal. 2 stage FU study; T1 within 1 wk from birth; T2 6 wks postpartum; Sociodemographic, obstetrics and infant data questionnaire, SCID-1, WMHCIDI	♂ Prevalence of depression at T1 3% Incidence at T2 5.9% Overall prevalence 8.9%; ♀ a T2 17.%; unemployed ♂ depression > employed	♀ incidence and prevalence of depression > ♂
[160]	♀ N = 1036; ♂ = 878	Longitudinal. Prenatal collection points: T1: 21 wks; T2: 28 wksT3: 32 wks; T4: 36 wks; Postnatal collection points: T5: 6 wks; T6: 6 months; T7: 12 months. Sociodemographic information, EPDS, ECR	Fathers’ depression. T1 $\bar{x}$ = 2.44; T2 $\bar{x}$ = 2.36; T3 $\bar{x}$ = 2.71; T4 $\bar{x}$ = 2.39; T5 $\bar{x}$ = 2.41; T6 $\bar{x}$ = 2.57; T7 $\bar{x}$ = 2.38. $\bar{x}$ ♂ symptom levels lower than the ♀ levels	Mothers’ depressive symptoms late in pregnancy predicted elevated symptom levels in fathers 6 weeks after birth
[161]	♀ N = 419; ♂ N = 228	Longitudinal. T1: 3rd trimester of pregnancy; T2: 1 wk post-partum; T3: 1-month post-partum. T1: HADS-FT2: HADS-F, ASDS-FT3: HADS-F, PDS-F	T1: Prevalence rates for probable PTSD was 24.3% for mothers and 20.3% for fathers T2: ♀ (63.9%) and ♂ (51.7%) manifested symptoms of acute stress disorder. T3: ♀ (20.7%) and ♂ (7.2%) had symptoms of PTSD	Both parents may suffer from acute stress disorder and PTSD symptoms after childbirth; acute stress disorder is a predictor of PTSD after childbirth for both parents
[162]	177 fathers, divided in: 38 ELBW (extremely low birth weight), $\bar{x}$ age 34.54; 56 VLBW (very low birth weight), $\bar{x}$ age 35.14; 83 FT (full-term), $\bar{x}$ age 33.07	Longitudinal. Perinatal depression was evaluated by the Edinburgh Postnatal Depression Scale (EPDS) at 3, 9, and 12 months postpartum	↑Levels of postnatal depression in the 1st mo, but only for parents of ELBW babies. Paternal depression assessed at 3 mo positively correlated with depression at 9 mo for the ELBW group, at 12 months for the VLBW group, and both 9 and 12 mo for the FT group	Prematurity leads to specific reactions in mothers and fathers, which are stronger in ELBW, intermediate in VLBW, and mild in FT
[163]	129 expectant fathers; $\bar{x}$ age: 35.1	Cross-sectional. Assessed during pregnancy with: Cambridge Worry Scale (CWS); The seven-item GAD-7; The 10-item EPDS; hostility subscale of the Brief Symptom Inventory BSI; Berlin Social Support Scale BSSS	Mean scores of the GAD-7 are above the norm values for the general German population, but below those for samples in primary care, with 6.1% reporting moderate and 1.6% reporting severe levels of anxiety. 13.5% showed a positive cutoff for depression. The mean score of pregnancy-related worries was low, with only 0.8% of the sample reaching a total score of 3, indicating less than major worry	↑Parental mental health problems (depression and anxiety) associated with lower levels of parental responsiveness, in turn related to poor parent-infant bonding
[164]	♀ = 57 ♂ = 53 infant aged $\bar{x}$ 6 months	Cross-sectional. 2 wk: sociodemographic informations, sleep diary, SPQ, CES-D	Paternal nocturnal duration: $\bar{x}$ 441.3 min Sleep quality score $\bar{x}$ 7.11 Paternal depression total score: $\bar{x}$ 8.04	Experienced mothers reported more fragmented sleep and perceived having worse sleep quality than first-time mothers, paternal sleep did not differ as a function of parental experience
[165]	1027 men with $\bar{x}$ age = 32.4 years	Longitudinal. EPDS at T1 (during pregnancy) and T2 (8 weeks postpartum). Socio-demographic characteristics, relationship factors, and health behaviors were assessed at T1. Perceived social support was measured with the short version of the Social Support Questionnaire (F-Soz U-14); The short version of the Partnership Questionnaire (PFB-K) was used to assess relationship satisfaction	During pregnancy (T1), the mean EPDS score was 3.9 ± 3.6). According to the standard EDPS cut-off scores, 50 expectant fathers (5%) screened positive for minor depression (≥10) and 40 (4%) for major depression (≥12). At the postpartum (T2) assessment, $\bar{x}$ EPDS score was 3.5 ± 3.3. Of those participating in the T1 assessment, 20 (2%) screened positive for minor depression and 22 (3%) for major depression	Perceived social support and relationship satisfaction appeared to be protective against paternal depression
[166]	♀ = 1945; ♂ = 1722	Longitudinal. T1: ♀ 13 wks postpartum; CES-D, DASS-21. T2: ♀ 5–6 yr follow-up, DASS-21. T3: ♂♀ 11–12 yr follow-up, DASS-21	Maternal anxiety and depressive symptoms were not related to paternal depressive symptoms at child age 11–12 years, while maternal postpartum depressive symptoms, depressive symptoms at 5–6 years and maternal anxiety at 5–6 years were positively related to paternal anxiety at 11–12 years	Fathers and children seem to be affected only to a small extent by maternal postpartum anxiety or depression
[167]	♂ = 881 (low-income)	Longitudinal. T1: 1 month after birth, T2: 6 months after birth, T3: 12 months after birth, EPDS, Time Spent with Infant, FFCWS (Parenting Self-Efficacy), material support provided to the baby	T1: EPDS $\bar{x}$ 3.71 T2: EPDS $\bar{x}$ 4.49 T3: EPDS $\bar{x}$ 4.07. Father’s parenting self-efficacy ($\bar{x}$ : 1–4): 3.71. Father’s provision of material support to the offspring (range: 0–20):	Greater time spent with the infant, parenting self-efficacy, and material support were all significantly associated with lower paternal depressive symptoms during the 1st year
[168]	♂ = 423; $\bar{x}$ age: 32.02	Cross-sectional. Sociodemographic and personal factors; Relationship factors; Infant and environmental factors; Substance use; EPDS	EPDS: 17% ≥ 10, 83% < 10	17% of the participants had paternal postpartum depression. Family income, substance use, family support, marital relation, unplanned pregnancy, and infant sleeping problems significantly associated with paternal postnatal depression
[169]	950 couples ♂ $\bar{x}$ age: 31.7	Cross-sectional. Socio-demographic characteristics; Mother-in-law relationship; Marital Satisfaction; EPDS	♂ EPDS ≥ 10 10.2%♂♀ EPDS ≥ 10 4.4%	4.4% of the couples both ♀♂ showed depressive symptoms. Maternal marital satisfaction showed a significant mediating effect on paternal PPD; direct effect of maternal PPD on paternal PPD
[170]	148 ♂; 34.6 years	Cross-sectional. At their mandatory two-and-a-half-year medical control were assessed with these instruments: The Edinburgh Postnatal Depression Scale (EPDS), The Swedish Parenthood Stress Questionnaire (SPSQ), Relationship Questionnaire (RQ)	Of the fathers, 11.5% had depressive symptoms (EPDS ≥ 12). The association between depressive symptoms and parental stress was moderate for fathers. Thirdly, of the RQ subscales, only the preoccupied attachment style was associated with depression, as shown by the difference in scores between depressed and non-depressed parents, while no differences between the groups were found on the remaining subscales (i.e., secure, avoidant or fearful). For the fathers, parental stress in the areas of social isolation and health problems, as well as the preoccupied attachment style, were the best predictors.	Depressive symptoms in fathers and mothers are not limited to the first-year post-partum. Elevated levels of depressive symptoms in early parenthood beyond the first year of the child’s life for both mothers and fathers
[9]	79 ♂ $\bar{x}$ age: 33.7	Longitudinal. T0: second trimester of pregnancy T1: 2nd month after childbirth. T0: psychiatric interviewand socio-demographic data collection, STAI-Y: PDSS	STAI-Y: Anxiety proneness: moderate in total sample ($\bar{x}$ 58.7 ± 4.2). 1.3% = mild; 70.9% = moderate; 27.8% = severe). PPND: upper normal limit in total sample ($\bar{x}$ 59.2 ± 33.1) 44.3% = normal adjustment; 13.9% = significantsymptoms; 41.8% = positive	Positive correlation among total score of STAI-Y and PDSS subscales, Anxiety/Insecurity (*p* = 0.011), Emotional Liability (*p* = 0.007), Cognitive Impairment (*p* = 0.023), and Loss of Self (*p* = 0.012)
[171]	116 ♂	Longitudinal. T0: 24 wks T1: 36 wks T2: post-birth T3: 6 months after birth. EPDS, Family life satisfaction, Childbirth coaching self-efficacy, Pregnancy anxiety, Labour length expectation pre-birth, Helpful - non-med pain manage, Birth satisfaction, Parenting self-efficacy, Family life satisfaction	EPDS: $\bar{x}$ 5EPDS ≥ 10 12% (EPDS ≥ 15 29%; EPDS ≥ 20 14%)	12% met criteria for PPD. Predictive factors: earlier depression, family life satisfaction, expectations, interventions at birth, pain management for their partner, and pregnancy anxiety
[172]	257 ♂	Longitudinal. T1: 13–14 yr T2: 15–16 yrT3: 17–18 yrT4: 19–20 yrT5: 23–24 yrT6: 27–28 yrT7: 12 months postpartumT1: sociodemographic characteristics, children’s development, MAS–RC, SMFQT2, T3: MAS–RC, SMFQT4-T6: DASS-21T7: EPDS	At T7 10% of fathers reported depression (EPDS ≥ 10)	83% of the pregnancies in which fathers reported postnatal depression were preceded by a preconception history of mental health problems. Perinatal depression was consistently higher in those with ahistory of persistent mental health problems across adolescence and young adulthood thanthose without
[173]	ELFE study (Étude Longitudinale Française depuis l’Enfance). 12,386 couples 2 mo postpartum	Cross-sectional. EPDS ≥ 12 for ♀ and ≥ 10 for ♂	10,278 (83%) were not depressed; in 166 (1.34%), both parents were depressed; in 1238 (10.0%), only the ♀ was depressed and in 703 (5.7%), only the ♂ was depressed.	The couples’ concordance for depression was a small proportion of the whole sample; a substantial proportion of expecting fathers shows depression
[174]	318 ♂; 233 first-time ♂and 85 second-time ♂; $\bar{x}$ age: 31.78	Cross-sectional. Fathers were screened for depressive symptoms and self-efficacy, during the first three days after childbirth, using the EPDS and GSES	70 fathers (22%) EPDS scores were more than 10, indicating depressive symptoms. 22.3% of the 233 first-time fathers and 21.2% of the 85 second-time fathers were experiencing depressive symptoms. The $\bar{x}$ age of fathers with depression (32.56 yr), was significantly higher than that of fathers without depression ($\bar{x}$ = 31.56 yr). Fathers without depression scored higher on the GSES than fathers with depression. Other factors associated with paternal depression were: satisfaction with living conditions, delivery mode, and partners’ comorbidity	About one-fifth of expecting fathers had at least moderate depression; fathers with depression were older than fathers without; living conditions and mother’s comorbidity are factors associated with paternal depression
[175]	181 fathers; $\bar{x}$ age: 30	Cross-sectional. EPDS and STAI in a single occasion during the third trimester of pregnancy	Increased risk of depression (EPDS score of 12 pointsand more) was observed in almost 4% of 181 fathers. 4.7% of fathers had ↑ trait anxiety. ↑state anxiety in almost 10% of men and in 19% of both expectant mothers and fathers. ↑ by 1 in trait anxiety → ↑ risk of depression in men 1.23 times. What is noteworthy, having trait anxiety at the level of 8 stens and more increased the risk of depression in men 38 times	Anxiety and depression are common among primiparous parents. ↑Anxiety levels → ↑ risk of depression in both mothers and fathers
[176]	1598 fathers at 32nd gestational week; 1343 fathers at 3 months;1211 fathers at 8 months; 776 fathers at 24 months postpartum. 1436 fathers $\bar{x}$ age: 32.5	Longitudinal. Depressive symptoms were measured prenatally and at the child’s age of 3, 8 and 24 months using the 10- item version of the CES-D. Prenatal predictors (sociodemographic, health, substance use, sleep, and stress related factors, family atmosphere) of depressive symptoms trajectories were derived from questionnaires.	Among fathers, mean levels of depression ↑ after the 3-mo postpartum assessment. The prevalence of self-reported depression (CES-D ≥ 10) among fathers, doubled from 5.1% (prenatal) to 10.2% (24 mo). 74.9% of fathers constantly report mild depressive symptoms (mean CES-D < 4). 22.6% with stable, moderate, or subthreshold depression ($\bar{x}$ CES-D >6 and <9); 2.65 reported constantly depression above clinical threshold (CES-D > 10)	Depressive symptoms trajectories from pregnancy up to 2 years postpartum seem stable, indicating the chronic nature of perinatal or postpartum depression. This study underlines the importance of inquiring about both the mothers and fathers’ depressive symptoms already during pregnancy. Given the chronicity of symptoms suggested by the trajectories, the treatment should begin as soon as the elevated levels are observed
[177]	127 fathers; 88.3% were primiparous; 67.9% were between 30–30 yr; 26.6% were between 18–29 yr; 5.5% were between 40–49 yr.	Longitudinal. Two assessments were performed: (1) 2nd trimester of pregnancy (2) 6 mo post-partum. In both assessment waves, participants completed Socio-demographic Questionnaire, EPDS, and PAPA	Mean EPDS scores: 2nd trimester of pregnancy: 4.59. 6 mo postpartum: 4.41. From 2nd trimester of pregnancy to six 6 mo postpartum, men with ↑ depression symptoms showed ↓ positive attitudes towards sex, a steeper ↓ in satisfaction with marital relationship and ↓ in positive attitudes towards pregnancy and the baby	Depression symptoms early in pregnancy may represent a risk factor to increase paternal adjustment problems and negative paternal attitudes during the transition to parenthood
[178]	192 UK fathers at third trimester. 115 USA fathers at third trimester	Longitudinal. Two assessments at: third trimester and 4 months postnatal, with 12-item GHQ, 20-item CES-D, and 6-item STAI. In the 3rd trimester of pregnancy and 4 mo post-delivery, fathers completed the Birth Memories and Recall Questionaire (birthMARQ), to examine in relation to anxiety and depression	Third trimester: 192 UK fathers mean scores on CES-D, GHQ, STAI: 7.83; 1.49; 11.16. 115 USA fathers mean scores on CES-D, GHQ, STAI: 9.13; 1.55, 11.87. 4 months postnatal: 178 UK fathers completed CES-D: 9.12, 182 UK fathers completed GHQ: 2.26;179 UK fathers completed STAI: 11.15. 103 USA fathers completed CES-D 11.12;99 USA father completed GHQ 2.62;103 USA fathers completed STAI: 12.01	The mediation effect of negative birth experiences in the association between mode of delivery and postnatal wellbeing highlights the need to take steps to reduce the trauma associated with giving birth by caesarean section
[179]	1023 fathers at 1-month postdelivery, $\bar{x}$ age: 32.9; 1330 fathers at 6 month postdelivery, $\bar{x}$ age: 32.8	Longitudinal. Paternal post-partum depression to assess with EPDS at 1 and 6-month postdelivery and association with covariates.	Prevalence of paternal postpartum depression was 11.2% (115) 1 mo postpartum. Associated factors: history of mental health disorders before pregnancy, psychological distress during pregnancy, low income, and infant disease under medical treatment. At 6 mo prevalence was 12.0% (N = 160). Associated factors: psychological distress during pregnancy, unemployment, and maternal depressive symptoms.	1- and 6-mo postpartum, psychological distress during pregnancy and socioeconomic difficulties are factors associated to postpartum paternal depression
[180]	Fathers sample size vary from 9846 at 18 weeks gestation ($\bar{x}$ age: 30.40) to 1951 at 21 years post-partum ($\bar{x}$ age: 53.34).	Longitudinal. Paternal post-partum depression to assess with EPDS at 10 timepoints (first at 18 weeks gestation, last at 21 years post-partum). 14915 have at least one EPDS assessment, with 4067 having all 10 assessments and 1476 having zero.	The greatest proportion with probable depression occurs at 21 years post-partum (9.17% of fathers with EPDS ≥ 13 and 20.30% of fathers with EPDS ≥ 10) while the lowest proportion is at 8 months post-partum (2.96% with EPDS ≥ 13 and 7.31% with EPDS ≥ 10).	A strength of the study is the size of the population, but the EPDS does not assess the duration and intensity of depressive symptoms and, in addition, approximately 10 years elapse between the measurement at timepoint 9 and timepoint 10, and parental depression may relate to the adolescent period of the children
[181]	138 first-time parents couples; mothers $\bar{x}$ age: 33.09; fathers $\bar{x}$ age: 35.19	Cross-sectional. PSS, subscale Dyadic Satisfaction (DS) of DAS; CES-D	No significant levels of depressive symptoms (women: $\bar{x}$ 11.30; men: $\bar{x}$ 8.58) Only six couples (4.37%) scored ≥ 16 on the CES-D. Mothers perceived more stress than their partners (women: $\bar{x}$ 11.88; men: $\bar{x}$ 10.94) and they obtained the same score in marital satisfaction as their counterpart did (women: $\bar{x}$ 41.27; men: $\bar{x}$ 41.27)	The model revealed that there was an intrapersonal indirect effect of fathers’ perceived stress on prenatal paternal depression through their marital satisfaction. Mothers’ perceived stress was associated with prenatal paternal depression through fathers’ dyadic satisfaction. Maternal indirect effects were all not significant, suggesting that their dyadic satisfaction and that of their partner did not mediate the relation between their perceived stress and that of their partner and their prenatal depression
[182]	350 Italian expectant fathers $\bar{x}$ age = 35.63 in the last trimester of pregnancy	Cross-sectional. Sociodemographic and individual information about previous history of psychiatric disorders and presence of stressful life events, addictions or other risky behaviours. CES-D, SCL-90-R, PSS, DAS.	CES-D: $\bar{x}$ 8.13, SCL-90-ANX: $\bar{x}$ 2.10, SCL-90-SOM: $\bar{x}$ 3.48, SCL-90-HOS: $\bar{x}$ 1.63, PSS $\bar{x}$ 10.97, DAS $\bar{x}$ 124.47, Addiction/risky behaviours: yes 19.9% no 80.01%; Stressful life events: none 63.4%, One 32.4%, more than two 4.3%; Previous psychiatric diagnosis: no 87.5%, yes 12.5%	32% of fathers were at-risk of developing a paternal affective disorder. Three different clusters were found:“psychologically healthy men” (68%) with low levels of symptoms on all the scales; “men at risk ofexternalized behavioral problems” (17.1%), characterized by one or more addictive or risky behaviorsand moderate levels of scales scores; and “men experiencing psychological distress” (14.9%), with thehighest scores on all the scales. A significant association emerged among the perceived stress, maritaladjustment, and cluster membership
[183]	Study group: 51 ♂ age 20–25: 2%, 25–30: 7.8%, 30–35: 23.5%, 35–40: 37.3%, 40–45: 21.6%, >45: 7.8%; control group: 33 ♂ age 20–25: 0%, 25–30: 9.1%, 30–35: 45.5%, 35–40: 30.3%, 40–45: 15.2%, >45: 0%; T1: 3rd postpartum; T2: 15–20 d postpartum.	Longitudinal. EPDS, STAI, PSS	EPDS and STAI were assessed during infant hospitalization and during mother hospitalization depending on the duration. EPDS DIH (duration infant hospitalization): ≤15 d: T1: 7.55 ± 5.46, T2: 5.45 ± 6.62; 15 d-1 mo.: T1: 6.44 ± 4.77, T2: 4.25 ± 5.55; >1 mo.: T1: 8.17 ± 4.68, T2: 5.71 ± 3.52. EPDS DMH (duration of mother hospitalization): <48 h: T1: 5.75 ± 4.26, T2: 3.88 ± 3.52; 3–5 d: T1: 6.92 ± 5.57, T2: 5.69 ± 6.66; 6–10 d: T1: 8.15 ± 4.45, T2: 4.46 ± 2.50. >10 d: T1: 8.24 ± 4.95, T2: 6.00 ± 3.57. STAI DIH: ≤15 d: T1: 22.91 ± 10.25, T2: 19.45 ± 12.79; 15 d-1 mo.: T1: 24.00 ± 11.25, T2: 16.88 ± 6.43; >1 mo.: T1: 25.04 ± 11.35, T2: 19.67 ± 9.52. STAI DMH: <48 h: T1: 20.13 ± 5.66, T2: 14.63 ± 4.40; 3–5 d: T1: 22.08 ± 12.45, T2: 22.15 ± 13.83; 6–10 d: T1: 23.31 ± 4.51, T2: 17.08 ± 6.61. >10 d: T1: 28.59 ± 12.42, T2: 19.35 ± 8.30	Parents of infants hospitalized in the NICU had significantly more symptoms of depression than parents in the control group for the first three days after birth. 3 weeks later, even if the EPDS scores for the study group were still slightly higher than those of the fathers of healthy infants, the difference between the groups was not significant
[184]	18 depressed fathers ($\bar{x}$ age = 33.11) from low-income households with low education backgrounds, with a child aged 0–3	Longitudinal. T0 (BL): demographic questionnaire, EPDS, HAM-D, PSI-SF, KAP, MSPSS, RSE, BDQ, EQ-5DT1(post-intervention): 3 months of LTP+ Dads, EPDS, HAM-D, PSI-SF, KAP, MSPSS, RSE, BDQ, EQ-5DT2 (follow-up):6 months post baseline, EPDS, HAM-D, PSI-SF, KAP, MSPSS, RSE, BDQ, EQ-5D	T1: EPDS: $\bar{x}$ 14.17, HAM-D: $\bar{x}$ 16.44, PSI-SF: $\bar{x}$ 91.17, KAP: sense of self $\bar{x}$ 21.06, physical development $\bar{x}$ 19.28, relationships $\bar{x}$ 21.06, understanding of the world $\bar{x}$ 16.83, communication $\bar{x}$ 16.83, MSPSS: $\bar{x}$ 34.28, RSE: $\bar{x}$ 18.33, BDQ: $\bar{x}$ 9.67, EQ-5D: $\bar{x}$ 40.28 T2: EPDS: $\bar{x}$ 7.00, HAM-D: $\bar{x}$ 11.33, PSI-SF: $\bar{x}$ 79.67, KAP: sense of self $\bar{x}$ 22.28, physical development $\bar{x}$ 19.17, relationships $\bar{x}$ 21.11, understanding of the world $\bar{x}$ 18.22, communication $\bar{x}$ 17.61, MSPSS: $\bar{x}$ 56.89, RSE: $\bar{x}$ 19.78, BDQ: $\bar{x}$ 4.78, EQ-5D: $\bar{x}$ 59.17T3: EPDS: $\bar{x}$ 6.06, HAM-D: $\bar{x}$ 9.89, PSI-SF: $\bar{x}$ 73.61, KAP: sense of self $\bar{x}$ 23.53, physical development $\bar{x}$ 20.76, relationships $\bar{x}$ 22.76, understanding of the world $\bar{x}$ 19.41, communication $\bar{x}$ = 18.18, MSPSS: $\bar{x}$ 55.00, RSE: $\bar{x}$ 20.28, BDQ: $\bar{x}$ 3.89, EQ-5D: $\bar{x}$ 59.44	LTP + Dads, which combines early child development information and group CBT, significantly improved paternal depression, parental stress, disability, perceived social support and self-esteem
[185]	131 participants 66 women $\bar{x}$ age: 29.54 65 men $\bar{x}$ age: 30.84	Longitudinal. T1: 2nd trimester T2: 3rd trimester T3: 3 months postpartum Sociodemographic questionnaire, STAI-S. EPDS. 24-Hour urinary free cortisol (24-HUFC)	T1: EPDS:♀$\bar{x}$ 6.59; ♂$\bar{x}$ 5.06 STAI-S: ♀$\bar{x}$ 36.38; ♂$\bar{x}$ 32.80 24-HUFC: ♀$\bar{x}$ 79.88; ♂$\bar{x}$ 76.23 T2: EPDS: ♀$\bar{x}$ 6.97; ♂$\bar{x}$ 4.17 STAI-S: ♀$\bar{x}$ 37.52; ♂$\bar{x}$ 33.93 24-HUFC: ♀$\bar{x}$ 104.62; ♂$\bar{x}$ 78.94 T3: EPDS: ♀$\bar{x}$ 6.07; ♂$\bar{x}$ 3.67 STAI-S: ♀$\bar{x}$ 35.61; ♂$\bar{x}$ 32.31 24-HUFC: ♀$\bar{x}$ 78.44; ♂$\bar{x}$ 90.98	In men, high anxiety or depressive symptoms are associated with less accentuated cortisol variations from the 2nd pregnancy trimester to 3- months postpartum, compared to men with low anxiety or depressive symptoms, who show a more accentuated cortisol variations
[186]	612 fathers 363 primiparous fathers (59%) 247 multiparous fathers (40%)	Cross-sectional. Socio-demographic factors, social support, professional support, depressive symptoms (EPDS)	21.2% of the fathers had at least mild depressive symptoms (EPDS ≥ 10), and 15.8% had severe depressive symptoms (EPDS ≥ 12). Multiparous fathers received significantly less professional and social support and were less frequently invited to child health visits than first-time fathers	Fathers receiving professional support from prenatal midwife, labor/birth midwife/nurse team, and child health nurse, as well as social support from their partner, and with higher income: ↓ depression.
[187]	1021 fathers 928 fathers of term babies, $\bar{x}$ age: 31.01 72 fathers of preterm babies, $\bar{x}$ age: 32.10	Longitudinal. T1: 3rd trimester of pregnancy (antenatal) T2: six wk after birth (postnatal) HADS. SWLS.	T1: no significant differences in HADS total, anxiety or depression subscale and SWLS scores. T2: fathers of preterm babies were significantly more likely to meet the case criteria for anxiety compared to fathers of term babies (25% vs. 12%). They also reported significantly lower SWLS scores compared to fathers of term infants (27.31 I. 27.88)	Fathers of preterm babies have ↑rates of anxiety and ↓QoL scores at six weeks postnatal compared to fathers of term babies
[188]	Partners with RPL 412 ♀ with RPL 281 ♂ with RPL ♂ $\bar{x}$ age: 36.4 yr ♀ $\bar{x}$ age: 34.6 yr	Cross-sectional. MDI. PSS. Scores of both sexes compared with scores from relevant cohorts of ♀ and ♂ from the general population	MDI: ♀$\bar{x}$ 9; ♂$\bar{x}$ 6 PSS: ♀$\bar{x}$ 15.7; ♂$\bar{x}$ 11.4 RPL men reported scores of depressive symptoms similar to comparison group (1.8% vs. 2.0%). RPL men reported fewer stress level than comparison group (10.7% vs. 15.8%)	Male partners in RPL couples did not have increased prevalence of stress and depression compared with other men, higher prevalence of moderate to severe depression and stress emerged among women who had experienced RPL
[8]	818 couples 1 yr after the birth of a child Fathers $\bar{x}$ age: 34.3 yr Mothers $\bar{x}$ age: 31.3 yr	Cross-sectional. 1 yr after the birth of a child PSS: ♀$\bar{x}$ 14; ♂$\bar{x}$ 14 EPDS: ♀$\bar{x}$ 4; ♂$\bar{x}$ 3 HADS: ♀$\bar{x}$ 4; ♂$\bar{x}$ 4 ACEs: ♀x89%; ♂$\bar{x}$ 71%	59% of both mothers and partners reported exposure to at least one of the ten ACE categories. 11% ♀ and 9% ♂ reported exposure to >4 ACE categories (p = 0.12). In multiple logistic regression analyses, there were associations between the ACE exposure load and unfavourable outcomes like bad health, anxiety, depression and perceived high stress	Mothers exposed to ACEs tend to have partners also exposed to ACEs. Exposure to ACEs was associated with bad health and unfavourable life conditions within the couples, especially among couples where both members reported exposure to multiple ACEs. These results should stimulate clinical incentives to find, to support and to treat mothers and partners as well as couples afflicted by ACEs
[189]	2546 individuals T1 ($\bar{x}$ 30.5 wk pregnant): 828 fathers, 1146 mothers T2 ($\bar{x}$ 8.9 wk post-partum): 698 fathers, 1028 mothers Fathers $\bar{x}$ age: 32.4 yr Mothers $\bar{x}$ age: 30.1 yr	Longitudinal. T1: sociodemographic factors, work-factors, EPDS, SCL-90 T2: pregnancy-related factors, birth-related factors, IES-R; Cross-sectional	T1: EPDS:♀$\bar{x}$ 5.7; ♂$\bar{x}$ 4.0 SCL-90:♀$\bar{x}$ 2.6; ♂$\bar{x}$ 1.8 2.9% of mothers and 0.3% of fathers suffered from PTSD before childbirth T2: IES -R: ♀ $\bar{x}$ 11.8; ♂ $\bar{x}$ 8.4 2.3% of mothers and 0.7% of fathers showed clinically important PTSD symptoms after childbirth	Depressive and anxiety symptoms, pregnancy complications, and poorer subjective birth experience predicted PTSD symptoms in both mothers and fathers. Lower job satisfaction, higher job burden, being first-time father, lower education, and mothers’ lower support during are predictors for PTSD symptoms in fathers
[190]	54♂, $\bar{x}$ age 35.06 yr ± 5.18, all non-depressed	Cross-sectional. CES-D, PSQI	CES-D total scores 8.19 ± 6.51; at a cutoff of ≥16, no man was at risk of depression. Higher depressive scores in fathers with poorer sleep at 6 months postpartum	Although depressive severity was not clinically significant, its levels correlated with poor sleep quality
[191]	1187♂, age ≤ 29, N = 345 (29.06%), 30–34 N = 530 (44.65%), >34 N = 312 (26.28%)	Cross-sectional. EPDS	Prevalence of postpartum depression in fathers was 13.82% (EPDS > 9)	These figures were reduced during the COVID-19 pandemic with respect to the period before
[192]	175♂ first-time fathers, $\bar{x}$ age 29.64 ± 5.13 (range 18–45 yr)	Cross-sectional. Assessed 3–6 months post-delivery with EPDS to measure depression and PIAS to measure attachment	EPDS > 13 14.2% of sample; correlation between EPDS and PIAS patience and tolerance *r* = –0.622 (*p* < 0.001); PIAS pleasure in interaction *r* = –0.330 (*p* < 0.001); PIAS affection and pride *r* = –0.527 (*p* < 0.001)	Attachment and depression are inversely correlated in new fathers 3–6 months post-delivery
[193]	769 pregnant ♀ and their partners;, $\bar{x}$ age = 29.49 yr ± 3.89; 769♂; $\bar{x}$ age = 31.16 yr ± 4.45	Cross-sectional. EPDS	139 (18.08%, 95% CI: 15.35–20.80%) ♀ reported minor depression (EPDS total score ≥ 10), and 60 (7.80%, 95% CI: 5.90–9.70%) of them reported severe depression (EPDS total score ≥ 13); 61 (7.93%, 95% CI: 6.02–9.85%) partners reported minor depression, and 23 (2.99%, 95% CI: 1.78–4.20%) severe depression. In all, 30 couples (3.90%, 95% CI: 2.53–5.27%) reported minor depression, and 9 couples (1.17%, 95% CI: 0.41–1.93%) reported severe depression	The network structures between women and their partners differ. The concordance for depression in the couples is not very high
[194]	T1: bl 30 couples. ♂$\bar{x}$ = 27.7 yr ± 6.0; ♀$\bar{x}$ = 26.5 yr ± 5.5.; 24♂ 27♀ T2: 3 mo postpartum; 17♂ 23♀ T3: 6 mo postpartum	Longitudinal. BDI-II, GAD-7, PSS-10, SSE-Q, NIH Toolbox Instrumental Support and Emotional Support Survey	♂ BDI-II score: T1: $\bar{x}$ = 6.5 ± 6.6; T2: $\bar{x}$ = 4.5 ± 4.6; T3: $\bar{x}$ = 3.6 ± 5.1. GAD-7 score: T1: $\bar{x}$ = 4.1 ± 4.5; T2: $\bar{x}$ = 3.2 ± 3.4; T3: $\bar{x}$ = 2.8 ± 3.2. PSS-10 score: T1: $\bar{x}$ = 14.9 ± 7.6; T2: $\bar{x}$ = 12.6 ± 7.1; T3: $\bar{x}$ = 10.9 ± 9.1. SSE-Q score: T1: $\bar{x}$ = 60.9 ± 14.3; T2: $\bar{x}$ = 57.8 ± 12.6; T3: $\bar{x}$ = 59.3 ± 13.7. ♀ BDI-II score: T1: $\bar{x}$ = 9.7 ± 8.1; T2: $\bar{x}$ = 8.4 ± 9.5; T3: $\bar{x}$ = 8.3 ± 8.4. GAD-7 score: T1: $\bar{x}$ = 6.8 ± 5.5; T2: $\bar{x}$ = 6.2 ± 5.1; T3: $\bar{x}$ = 6.3 ± 4.8. PSS-10 score: T1: $\bar{x}$ = 20.4 ± 9.1; T2: $\bar{x}$ = 16.9 ± 7.4; T3: $\bar{x}$ = 16.8 ± 6.8. SSE-Q score: T1: $\bar{x}$ = 48.9 ± 11.6; T2: $\bar{x}$ = 51.4 ± 14.5; T3: $\bar{x}$ = 52.2 ± 12.2	Consistent with previous research, ♀ exhibited ↑ anxiety and depression than ♂. Significant ↓ in perceived stress among both ♂ and ♀ with moderate effect sizes. Small effect sizes associated with the extent of ↓depression among ♂. Results of social support were mixed
[7]	608 participants $\bar{x}$ = 30.4 yr ± 1.4	Longitudinal. Cohort study DASS-21	First study: longitudinal analysis of MAPP data identified 5 characteristic profiles of ♂’s patterns of depressive symptom severity and presentation of anger. Profiles indicating pronounced anger and depressive symptoms were associated with fathers’ lack of perceived social support, and problems with coparenting and bonding with infants. Second study: MAPP data were combined with 3 other Australian cohorts in a meta-analysis of associations between fathers’ self-reported sleep problems up to 3 years postpartum and symptoms of depression, anxiety and stress. Adjusted meta-analytic associations between paternal sleep and mental health risk ranged from 0.25 to 0.37	MAPP represents a unique study with recruitment of men approaching the peak age for entering fatherhood in order to specifically understand preconception risks and protective factors associated with a normatively timed transition to fatherhood. Despite calls for greater preconception engagement with prospective fathers, psychological and social factors that prepare men for fatherhood are vastly understudied compared with the equivalent in women
[195]	Prospective cohort study; 230 ♀, 201 ♂	Longitudinal. EPDS ≥ 13	At initial EPDS assessment shortly after NICU (Neonatal Intensive Care Unit) admission, 33% of mothers (N = 57) and 17% of fathers (N = 21) scored positive on the EPDS, with average scores of 8 for mothers and 6 for fathers (range: 0–21)	♀ experience both a decrease in average depression symptom scores and probability of screening positive for depression. ♂ did have a small and significant ↓ in EPDS scores from initial assessment to 1 mo after discharge, the probability of a positive depression screen remained the same across time
[196]	427 ♀ $\bar{x}$ = 30.80 yr ± 4.56, 170 ♂$\bar{x}$ = 33.19 yr ± 5.63	Cross-sectional. EPDS, DASS-21	♀ Depression symptoms: $\bar{x}$ = 7.04 ± 4.72; anxiety: $\bar{x}$ = 3.47 ± 5.51; stress: $\bar{x}$ = 9.18 ± 7.96. ♂ Depression symptoms: $\bar{x}$ = 5.78 ± 4.52; anxiety: $\bar{x}$ = 3.51 ± 5.95; stress: $\bar{x}$ = 7.65 ± 7.83	↑Mental health problems (depression and anxiety) associated with ↓ levels of parental responsiveness and poor parent-infant bonding
[197]	8 postpartum couples ♀$\bar{x}$ = 24.88 yr ± 5.54, ♂$\bar{x}$ = 29.63 yr ± 4.43	Cross-sectional. EPDS ≥ 13 on post partum ♀, BDI on ♂	BDI score: $\bar{x}$ = 7.88 ± 10.79. EPDS score: $\bar{x}$ = 11.75 ± 7.06	With ↑ levels of depression among ♀, many ♂ manifested depression themselves. Family functioning scores in the clinical range. ♂ reported multiple problems when their partner was experiencing depression symptoms, including feeling exhausted, unprepared, uninformed regarding ♀ postpartum illness, and having trouble balancing work and family needs
[198]	189 ♂$\bar{x}$ = 36.12 yr ± 2.39	Cross-sectional. Neuroticism subscale of EPQ, PSS, EPDS	PSS score: $\bar{x}$ = 16.91 ± 7.23. Neuroticism subscale of EPQ: $\bar{x}$ = 5.73 ± 3.26. EPDS: $\bar{x}$ = 11.79 ± 4.01	Perceived stress, neuroticism, and psychological inflexibility were significantly associated with depressive symptoms. This study found that the prevalence of depressive symptoms of new Chinese ♂ was 23.81%, which is ↑ than the 13.6% prevalence observed in a previous meta-analysis of Chinese ♂
[199]	198 ♀ $\bar{x}$ = 30.5 yr ± 4.94 (range 19–42 yr), 118 ♂ $\bar{x}$ = 31.9 yr ± 5.42 (range 21–47 yr)	Cross-sectional. EPDS, HADS, PSS, MSPSS.	♀ EPDS: $\bar{x}$ = 6.91 ± 4.25; HADS: $\bar{x}$ = 5.12 ± 3.49; PSS: $\bar{x}$ = 15.63 ± 6.60; MSPSS: $\bar{x}$ = 5.39 ± 0.74. ♂ EPDS: $\bar{x}$ = 5.52 ± 3.84; HADS: $\bar{x}$ = 4.67 ± 3.48; PSS: $\bar{x}$ = 15.00 ± 6.86; MSPSS: $\bar{x}$ = 5.17 ± 0.88	Pandemic impact is not homogenous; 69% of all participants found no changes to their intimate lives. For ♀ and ♂ whose relationships were affected by the pandemic, results showed heterogenous changes. Most ♂ and ♀ participants (58.6%) reported pandemic-induced negative changes, with only 28.3% noting positive changes in their relationships; 13.1% of new parents reported both positive and negative effects
[200]	591 couples ♂$\bar{x}$ = 32.9 yr ± 5.2	Cross-sectional. EPDS, GHQ-12, MSPSS, PSS.	The prevalence of ♂ and ♀ PPD was 15.7% (93) and 31.8% (188), respectively.	↑ PPD prevalence in ♂ after childbirth and significant role of ♀ PPD and ♂’s psychological and demographic factors in developing ♂ PPD. Significant ↑ in PPD in first father ♂
[201]	151♂. Clinical group (CLG) $\bar{x}$ = 40.83 yr ± 7.04 (28–59); Control Group (COG) $\bar{x}$ = 36.69 yr ± 8.20 (24–63)	Cross-sectional. OBQ-49, DASS-2, Fertility Quality of Life Questionnaire.	CLG DASS-2 score: depression $\bar{x}$ = 2.03 ± 2.90; anxiety $\bar{x}$ = 1.61 ± 2.20; stress $\bar{x}$ = 5.01 ± 3.88. COG DASS-2 score: depression $\bar{x}$ = 4.30 ± 4.22; anxiety $\bar{x}$ = 2.72 ± 2.85; stress $\bar{x}$ = 7.44 ± 4.99	♂ undergoing a MAR pathway had ↓ depressive-anxious and stress symptoms than the control group from the general population without infertility. The finding that men undergoing MAR reported lower depressive-anxious symptoms than control ♂ was somewhat unexpected. An explanation for this surprising result might be that the prevalence rates of anxious and depressive symptoms significantly ↑ also in the general population during the pandemic
[202]	A cohort study was completed between 2015–2019. 2442 ♂ $\bar{x}$ = 33.90 yr ± 4.88; T1: bl, T2: 3 mo postpartum, T3: 6 mo postpartum, T4: 9 mo. postpartum, T5: 12 mo. postpartum, T6: 18 mo. postpartum, T7: 24 mo. postpartum	Longitudinal. EPDS ≥ 12, STAI ≥ 44	A total of 569 ♂ (22.4%) had comorbid depression/anxiety at some point in the first year postpartum (2.2% at T1, 8.44% at T2, 8.91% at T3, 8.0% at T4, and 8.1% at T5) and 323 ♂ (13.2%) had comorbidity at some point in their second year postpartum (8.1% at T6 and 8.6% at T7).	For those with severe comorbidity, the prevalence started from 0.47% at baseline rising to 3.0% at the 12 mo. postpartum and remaining consistent to 3.3% at 24 mo. postpartum. The prevalence of paternal dep ression symptoms (EPDS > 9) increased from birth to 6 mo., from 4.0% at baseline within the first 3 wks postpartum to 11.5% at 3 mo. postpartum and 11.7% at 6 mo. postpartum. Rates decreased slightly to 10.8% at 9 mo., 10.3% at 12 mo. postpartum and stabilized thereafter to 24 mo. The prevalence of anxiety symptoms (STAI > 38) followed a similar trend starting at 8.8% at bl increasing significantly to 22.2% at 3 mo. and 21.9% at 6 months postpartum and stabilizing to 20.4% at 24 mo. postpartum
[203]	T1: 295 participants. A total of 79 participants (26.7%) dropped out during the study, of whom 65.8% were partners = 52. At T2 and T3 respectively 55 participants (♀ = 22, partners = 33) and 24 participants (♀ = 5, partners = 19) were lost to follow-up	Longitudinal. BDI, STAI PSQI at T1: pre-treatment fertility, T2: post-treatment ferility, T3: 9–12 we after pregnancy test	Mean psychological distress scores for the total sample at baseline were $\bar{x}$ = 10.0 ± 8.6 for depressive symptoms, $\bar{x}$ = 37.3 ± 10.9 for state anxiety, and $\bar{x}$ = 16.5 ± 9.1 for infertility-related distress. PSQI global scores were statistically significantly associated with state anxiety and depressive symptoms	Poor sleep quality is a prevalent problem among couples undergoing fertility treatment and is associated with psychological distress and possibly with pregnancy outcomes. Success rates after fertility treatment remain moderate, and poor sleep quality, a potentially modifiable factor, could be relevant to screen for and treat among couples undergoing fertility treatment
[204]	543,555 ♂	Longitudinal. Descriptive prospective study design; data from the Danish National registers. Perinatal psychiatric episodes assessed as incidence of first-time and prevalence of recorded in- or outpatient admissions for any mental disorder and redeemed prescriptions for psychotropic medication in ♂ at T1: 9 mo before birth, T2: until 12 mo after birth	Prevalence proportions for fathers psychiatric in- and outpatient episodes showed an increasing trend over the perinatal period and were marginally higher PP compared to pregnancy. No difference between the periods for incidence of prescriptions for psychotropic medication. Psychiatric disorders in expecting and new fatherhood were mainly treated in primary care with cumulative incidence of prescriptions for psychotropic medication of 14.56 per 1000 births during the first year of fatherhood	Becoming a father did not appear to trigger a substantially increased risk of severe psychiatric disorders, as it has been observed for new mothers
[205]	529 ♀ $\bar{x}$ = 32.5 yr ± 4.1 92 ♂ $\bar{x}$ = 34.1 ± 5.3	Cross-sectional. PHQ-9; GAD-7; PSS	PHQ-9 $\bar{x}$ = 3.0 ± 3.3; GAD-7 $\bar{x}$ = 2.5 ± 3.1; PSS $\bar{x}$ = 21.4 ± 6.3	There is a high prevalence of high-risk health behaviours in ♀ ♂ actively trying to conceive or planning to achieve pregnancy soon. Health promotion should be a key component of preconception health interventions for both ♀ ♂ as part of a life course approach to optimizing population health
[206]	FICare Group (family integrated care model): 89 ♂ $\bar{x}$ = 35.1 yr ± 4.8; SNC group (standard neonatal care): 93♂ $\bar{x}$ = 36.4 yr ± 5.5	Cross-sectional. PSS	PSS scores: FICare group $\bar{x}$ = 40.8 ± 20.3; SNC group $\bar{x}$ = 49.4 ± 18.9. ♂ in FICare experienced less stress and had higher participation scores compared with those in SNC. Participation mediated the beneficial association of the FICare model with ♂ depressive symptoms and bonding with their newborns	The FICare model is associated with ↓ paternal stress at discharge and enables ♂ to be present and participate more than SNC, thus improving paternal mental health
[207]	418 ♂ at T1:1 we Post-partum; 398 ♂ at T2: 3 mo Post-partum	Cross-sectional. EPDS, STAI	EPDS scores: T1: $\bar{x}$ = 3.11 ± 3.16; T2: $\bar{x}$ = 3.90 ± 3.73. STAI scores: T1: $\bar{x}$ = 34.17 ± 8.65; T2: $\bar{x}$ = 34.49 ± 8.85	Further studies should focus on possible pathways of the association between paternal ACE (adverse childhood experience) and mental health problems, which may help develop an intervention to prevent postpartum depression and anxiety in fathers with ACE
[208]	352 couples	Longitudinal. EPDS cut-off ≥ 12, T1: prenatal, T2: post-partum	EPDS scores ♂: T1: $\bar{x}$ = 35 ± 9.9, T2: $\bar{x}$ = 76 ± 21.5	Our study provided that the mechanism of prenatalmarital satisfaction and ♀prenatal depression that best predicted ♂postpartum depression. These findings suggest that the prenatal period is the ideal time for PPD. Considering the quality of marital relationships and couples’ perinatal mental health are necessary to promote fathers’ mental health. Addressing ♂♀ PPD and considering marital satisfaction cooccurrence is essential in expecting families’ research and practice
[209]	100 ♂ $\bar{x}$ = 31.34 yr ± 4.38	Longitudinal. Correlations between the Baby Care Scale- Antenatal BCS-AN at T1(Prenatal) and the Baby Care Scale- postnatal BCS-PN at T2 (Postnatal) and measures of anxiety (STAI-S) and depressive symptoms (EPDS)	Regarding the BCS-AN and BCS-PN criterion validity, significant medium-sized correlations were obtained between the BCS-AN and BCS-PN and measures of anxiety and depressive symptoms and measures of father-infant emotional involvement. Significant associations between mental health problems of ♂ and frequency of care provided by the ♂ to the infant	This study suggested that the BCS-AN and the BCS-PN are reliable multidimensional self-reported measures to assess the involvement of father in infant care during pregnancy and the postpartum period
[210]	81 ♂ $\bar{x}$ = 37.0 yr	Longitudinal. MADRS cut-off ≥ 7, EPDS cut-off ≥ 10. T0 = pregnancy, T1 = 3 mo postpartum, T2 = 6 mo postpartum, T3 = 12 mo postpartum	EPDS evidence ♂ depressive symptoms at T0: 3.7%; T1: 5.6%; T2: 3.6%; T3: 12%. MADRS evidence ♂ depressive symptoms at T0: 7.4%; T1: 16.4%; T2: 13.1%; T3: 15.7%. Using either the EPDS or MADRS, 25.6% ♂ screened positive for depression at least once during the study period. 69.2% ♂ who screened positive for depression experienced the increased depressive symptoms during pregnancy and the frst 3 mo postpartum for the frst time, 15.4% between 3 and 6 mo, and 15.4% between 6 and 12 mo postpartum.	The aim of this pilot study was to investigate the prevalence of ♂♀ perinatal depression and additionally identify psychosocial and biological risk factors. Despite a relatively small sample size, we found that ♂ are also at risk of developing peri- and postnatal depression, consistent with previous studies.
[211]	454 ♂ $\bar{x}$ = 32.21 yr ± 4.70	Cross-sectional. EPDS cut-off score ≥ 13	EPDS scores♂: $\bar{x}$ = 6.45 ± 4.25. PPD ♂7.5%. The predictors of paternal depression were paternal parenting satisfaction and selfefficacy, maternal depression and whether the pregnancy was planned	♂♀ depression were positively correlated and were predictive factors for one another. Healthcare professionals should screen both ♂♀for depression in the early PP and provide targeted support during time in hospital following birth. The focus of future interventions should be on both parents rather than just mothers
[212]	177♂	Longitudinal. EPDS cut-off ♂ ≥ 13. T1: 3 mo, T2: 9 mo, T3: 12 mo postpartum	EPDS results revealed a general decrease in Perinatal Depression across the year. Considering birth weight, ELBW (Extremely low birth weight) parents showed higher PND levels at T1 and a higher reduction of symptoms over time than VLBW (Very low birth weight) and FT(full term) ones. Given also parental role, ELBW ♀ showed higher PND levels at T1 and a higher decrease of symptoms over time than VLBW and FT ♂♀	Findings suggest that premature birth in relation to its severity may lead to different affective reactions in ♂♀; particularly ♀ in case of more serious preterm condition, are at higher risk for PND in the first trimester, however showing improvement over time. Interventions should be promoted, andtailored, according to the risk connected to severity of prematurity.

Abbreviations: ACEs: Adverse Childhood Experiences; ADS-L: Allgemeine Depression Skala; AQ: Aggression Questionnaire; ART: Assisted Reproduction Treatment; AUDIT: Alcohol Use Disorders Identification Test; BAI: Beck Anxiety Inventory; BCAI: Baby Care Activities Inventory; BDI: Beck Depression Inventory; BDQ: Brief Disability Questionnaire; BL, baseline; CES-D: Center for Epidemiologic Studies Depression Scale; CI: Confidence Interval; CIDI: Composite international diagnostic interview; CIS-R: Clinical Interview Schedule Revised; CMS: Child Maltreatment Scale; CRI: Coping Response Inventory; DAS: Dyadic Adjustment Scale; DAS-SF: Dyadic Adjustment Scale-Short Form; DASS-21: 21-item Depression, Anxiety, Stress Scales; DI: Depression Inventory; DIS: Diagnostic Interview Schedule; DI-SF: Depression Inventory-Short Form; DSQ: Defense Style Questionnaire; EPDS: Edinburgh Postnatal Depression Scale; EPDS-P: Edinburgh Postnatal Depression Scale-Partner; EPQ: Eysenck Personality Questionnaire; EQ-5D: Euro-Qol-5 Dimensions; FAIOC: Fathering Activities Inventory with Own Child; FAIOF: Fathering Activities Inventory with Own Father; FFFS: Feetham Family Function Survey; FIAI: Father–Infant Attachment Inventory; FSOC-S: Family Sense of Coherence Scale Short Form; FFS: Family Support Scale; GAD-7: Generalized Anxiety Disorder Scale; GHI: General Health Index; GHQ: General Health Questionnaire; GMDS: Gotland Male Depression Scale; HADS: Hospital Anxiety and Depression Scale; HPQ: Health Perceptions Questionnaire; Ham-D or HDRS: Hamilton Depression Rating Scale; IBM: Intimate Bond Measure; ICSI: Intracytoplasmic Sperm Injection; ICQ: Infant Characteristic Questionnaire; IDD: Inventory to Diagnose Depression; IES: Impact Of Event Scale; IES-R: Impact of Event Scale-Revised; IPSM: Interpersonal Sensitivity Measure; IRS: Interpersonal Reactivity Scale; ISSB: Inventory of Socially Supportive Behaviors; IVF: In Vitro Fertilisation; KAP: Learning through Play Knowledge, Attitude and Practices; KSP: The Karolinska Scales of Personality; LES: life event scale; LIFE: Longitudinal Interval Follow-up Evaluation; LCI: Lifestyle Changes Inventory; LTP + Dads: Learning through Play+ Dads; MADRS: Montgomery-Åsberg Depression Rating Scale; MAR: Medically Assisted Reproduction; MAT: Marital Adjustment Test; mo.: months; MDI: Major Depression Index; MINI: Mini International Neuropsychiatric Interview; MIRI: Maternal Infant Responsiveness Instrument; MOS-FMFM: Medical Outcomes Study Family and Marital Fuctioning Measures; MOS-SSS: Medical Outcomes Study Social Support Survey M(F)-PHI: mothers’ (and fathers’) Postnatal Health Instruments; MSPSS: Multidimensional Scale of Perceived Social Support; NA: Negative Affection; NCATS: Nursing Child Assessment Teaching Scale; PA: Positive Affection; PANAS: Positive and Negative Affect Schedule; PBI: Parental Bonding Instrument; PBQ: Postpartum Bonding Questionnaire; PCS: Perceived Control Scale; PFA: Paternal Foetal Attachment Scale; PIAS: paternal–infant attachment scale; PLDS: Perception of Labour and Delivery Scale; POMS: Profile of Mood States; PND: Postnatal Depression; m-PNTQ: Modified postnatal negative thoughts questionnaire; PPD: Post-Partum Depression; PRAQ: Pregnancy Related Anxiety Questionnaire; Prime-MD: Primary Care Evaluation of Mental Disorders; PSI: Parenting Stress Index; PSI-SF: Parenting Stress Index-Short Form; PSOC: Parental Sense of Competence scale; PSQI, Pittsburgh Sleep Quality Index; PSS: Perceived Stress Scale: PTSD-Q: Post-traumatic Stress Disorder Questionnaire; QoL: quality of life; RLCQ: Recent Life Change Questionnaire; RPL: recurrent pregnancy loss; RSE: Rosenberg’s Self-Esteem Scale; SADS: Schedule for Affective Disorders and Schizophrenia; SCID-I: Structured Clinical Interview for DSM-IV axis I disorders; SCID-NP: Structured Clinical Interview for DSM-III-R, Non-Patient Version; SCL-90R: Symptom Checklist 90-R; SMAT: Locke-Wallace Short Marital Adjustment Test; SOC: Sense of Coherence; SOMS: Sense of Mastery Scale; SPSQ Swedish Parenthood Stress Questionnaire; SPHERE: Somatic and Psychological Health Report; SRRS: Social Readjustment Rating Scale; SSE-Q: Social Support Effectiveness Questionnaire; SSNI: Social Support Network Inventory; STAI: State-Trait Anxiety Inventory; STAXI: State/trait anger inventory; SSQ: Social Support Questionnaire; SSS: Social Support Scale; SWLS: Satisfaction with Life Scales; Tes: testosterone; VAS: Visual Analogue Scale; VPSQ Vulnerable Personality Style Questionnaire; WEMWBS: Warwick and Edinburgh Mental Well-Being Scale; WFCS: Work–Family Conflict Scale; wk: weeks; WMHCIDI: World Mental Health Composite International Diagnostic Interview; WPBL-R: What Being the Parent of a New Baby is Like- Revised; $\bar{x}$, mean; yr: year(s) ♂: man; ♀: woman. d: day(s).

Of the eligible studies, 110 (53.92%) were longitudinal and 94 (46.08%) were cross-sectional. Most studies adopted the EPDS for diagnosing depression in fathers (N = 157; 76.96%), 20 used the CES-D (9.80%), 17 used the BDI (8.33%), 10 used the HADS (4.90%); 8 used the depression dimension of the SCL-90 (3.92%), only 2 used the HAM-D (0.98%) or the Depression Inventory (N = 2; 0.98%), and only 1 used the MADRS (0.49%). The eligible studies involved a total of 849,913 fathers (Table 1).

The number of studies dealing with the prevalence of depression in fathers, independently from whether they were comparing men and women, was 108 [9,15,16,18,19,20,22,23,24,28,30,34,36,38,40,42,43,44,45,46,48,51,53,55,57,58,59,60,61,62,65,66,68,69,70,71,72,73,74,76,77,79,80,82,84,86,87,88,89,93,94,97,100,102,104,105,108,109,112,115,119,120,126,127,129,130,131,133,135,138,139,140,142,144,146,147,148,152,154,155,156,157,159,165,168,169,170,171,172,173,174,175,176,179,180,181,186,188,190,191,192,193,195,200,202,210,211]. These studies involved 49,239 male participants, either new fathers or expecting fathers. In these studies, prevalence ranged from 0% to 33.79%, with a weighted mean prevalence of 4.97%. Generally, studies with smaller samples were likely to yield extreme values, while large cohort studies found figures near to the weighted mean. Studies dealing with women/men comparisons usually found higher proportions of depressive symptomatology in women compared to men and only a small amount of intracouple correspondence between mothers’ and fathers’ depression.

## 4. Discussion

In this systematic review, we sought to identify the prevalence of depression in expecting fathers and to see whether the occurrence of depression in the pregnant or delivering woman matches the development of depression in the father. We found a figure of around 5% for depressive symptoms in fathers of newborn children or in expecting fathers. Despite the low occurrence of depression in men who are partners of child-bearing women, the entity of paternal depression has the right to be framed within the current nosographic systems. We also found that depression may co-occur in the same couple, but in a very low, though not negligible, amount—around 1–2% [173]. It is more probable that depression in expecting mothers and fathers follows its own course, while in some instances, it has been shown that the presence of depression in one member may predict levels of depression in the other.

Despite the fact that in the DSM-5 [213], the issue of male or female depression is not fully clarified, some studies point to there being gender-related symptom differences for depression [214,215]. It is possible that such symptom differences are amplified by the condition of expecting a child, with the risk factors and the characteristics of depression differing between women and men [152]. Seeing one’s body changing in shape, whether one’s own or one’s partner, may have a disconcerting effect in both, similar to what one has already experienced during his/her transition from adolescence to adulthood. This could induce someone to feel awkward about oneself or about their partner. In turn, this may ensue in a declining couple adjustment, which is independently associated with depressive symptoms 6 months postnatally in men during their transition to parenthood [154]. The finding of a differential response to antidepressant treatment between the two genders offers hope for the personalization of depression with respect to diagnosis and treatment [216,217].

There are data for both coordinated depressive symptom manifestation within a couple and for the disjoint occurrence of depression in couples. The former is suggested by the fact that having depression in a partner increases the odds of developing depressive symptoms in the other, and this holds true for both women and men [115]. During the postpartum period, the mean severity of depressive symptoms declined linearly with similar slopes in both men and women [112]. While prenatal depression in fathers predicts the worsening of depressive symptoms in the new mother across the first six postpartum months, which speaks in favour of the former hypothesis, this is not true for the opposite [218], which supports the latter [112]. No association between depressive symptoms was found between men and women [114,219]. Longitudinal patterns of psychopathology differ between men and women; for men, the early phase of pregnancy is the most difficult period in the transition to parenthood, while prenatal EPDS scores predicted postnatal scores in both men and women independently [129]. Parents whose parenting efficacy was more negative than expected displayed higher depression levels one month post-delivery; four months post-delivery, this effect disappeared in women, but not in men. This suggests that the two genders experience their transition to parenthood differently [134]. Furthermore, Nishimura and Ohashi found no association between maternal and paternal depression [53]. The results of these studies speak against the hypothesis that in couples, depression co-occurs in both members or that it has a similarity in its course. Yet another Japanese study found a majority of prenatal and postnatal parents (83%) not to be depressed, while a very small proportion (1.34%) of both parents were depressed, with 10.0% of new mothers and 5.7% of new fathers having depressive symptoms [173]. Hence, only in a very small proportion of couples did depression coexist in both members.

### Limitations

The literature included in this systematic review used very different methodologies, including the instruments used, the time of observation, outcomes (some included psychosocial factors and others not, different parities), and the design (cross-sectional and longitudinal studies were prominently represented; the former do not allow for probing causality, while the latter employed different assessment timepoints). This rendered the data unable to be meta-analysed. Furthermore, the extensive use of the EPDS in men, an instrument that has proved valid for perinatal and especially postnatal depression in women, subtracts from the validity of the data. The male variant of the questionnaire has found little application heretofore (just two studies among those eligible, i.e., Moran and O’Hara, 2006; Fisher et al., 2012) [41,69], and its reliability compared to the original women’s version seems questionable [41]. The EPDS provides three different cutoffs according to severity and probability, thus rendering the very concept of depression fragmentary. Different studies used different cutoffs, often chosen arbitrarily—for example, different cutoffs for fathers and mothers (9 for fathers and 10 for mothers [130] and 6 for fathers and 8 for mothers [195]; this would have affected the results. Finally, we used only two databases/registries. The inclusion of more databases could have increased the final output; however, it would not have dramatically changed the conclusions.

## 5. Conclusions

Depression in couples expecting a baby or having had a newborn recently occurs disjointly, although depression in one member may affect depression in the other. Paternal depression is an entity with its own clinical dignity, despite its low occurrence (around 5%). The studies dedicated to this issue use poorly validated specific instruments, different methodologies, different assessment timepoints and follow-ups, unstandardized cutoffs, and different patient interview methods (i.e., vis à vis interviews, telephone calls, internet-based surveys, etc.). There is a need for the standardization of the methods, including outcomes, assessment tools, and follow-up intervals, and a focus on clinical and social measures that may constitute predictors of depression in men (and women) undergoing their transition into parenthood.

## Figures and Tables

**Figure 1 jpm-12-01598-f001:**
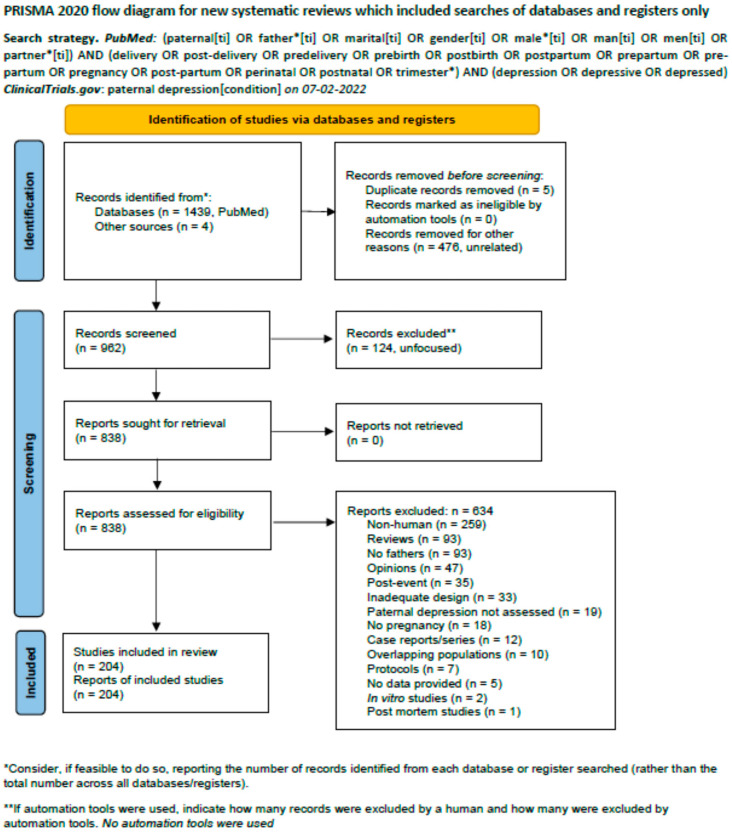
PRISMA flow diagram of the search strategy, showing the selection process with reasons for exclusion [11].

**Figure 2 jpm-12-01598-f002:**
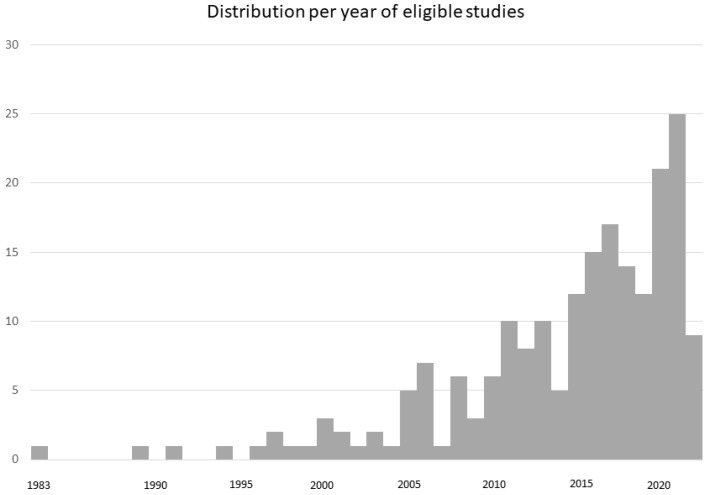
The per year distribution of eligible studies. There is a steady increase starting from 2000 (please note that 2022 includes only January and February until its 3rd day).

## Data Availability

Not applicable.

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
