# Peer review of "Depressive Symptoms in Expecting Fathers: Is Paternal Perinatal Depression a Valid Concept? A Systematic Review of Evidence"

_jpm, 2022, doi:10.3390/jpm12101598_

Round 1
Reviewer 1 Report
This is a wonderful concise review, methods, results, and discussion are comprehensible and of appropriate elaborateness.
Minor suggestions:
I do not think that the following observation can help to understand depressive symptoms in expecting fathers:
"Of course, this disparity is much lamented by people who focus their research on fathers’ perinatal depression, since it entails lower resources available for dedicated programs to deal with it (MacDonald et al., 2021; Andersson et al., 2021)."
Please omit.
Please check spelling (e.g. The results of these studies speaks against..).
Author Response
This is a wonderful concise review, methods, results, and discussion are comprehensible and of appropriate elaborateness.
Thank you for reviewing our paper and for finding it interesting. Please find our changes to the text in red characters.
Minor suggestions:
I do not think that the following observation can help to understand depressive symptoms in expecting fathers:
"Of course, this disparity is much lamented by people who focus their research on fathers’ perinatal depression, since it entails lower resources available for dedicated programs to deal with it (MacDonald et al., 2021; Andersson et al., 2021)."
Please omit.
We completely rephrased the observation that was meant not for better understanding depressive symptoms in expecting fathers, but just to remark the fact that the focus in literature is on maternal depression, hence the dearth of financial support to programmes directed at improving paternal depression .
Please check spelling (e.g. The results of these studies speaks against..).
You are right, it was a typo. We corrected. Thank you for carefully reading our paper and for suggestions that led to its improvement.
Reviewer 2 Report
This study is to quantify the extent of depression in expectant fathers and to determine if it is consistent with depression in their pregnant partners. The manuscript is well written and compelling.
Why did you limit your search engine to PubMed and ClinicalTrials.gov?
Would the inclusion of other search engines change the results of this study(e.g., Embase)?
Author Response
This study is to quantify the extent of depression in expectant fathers and to determine if it is consistent with depression in their pregnant partners. The manuscript is well written and compelling.
Thank you for appreciating our manuscript and for your positive attitude. Please find our changes to the text in red characters in the revised version.
Why did you limit your search engine to PubMed and ClinicalTrials.gov?
It is our experience that using databases such as CINAHL, Scopus or Google Scholar would have produced many unrelated papers, while relevant ones would have been all contained in the PubMed search. Cochrane searches would have been very restrictive and not representative of literature. Embase and PsycLit/PsycINFO/PsycARTICLES (searching it on September 22, 2022 produced 991 articles, far less than those retrieved by PubMed on February 2022 and much less pertinent) databases are affected by producing many intradatabase duplicates and few relevant papers, which would not have considerably added to PubMed. We believe the National Library’s PubMed to provide all needed material and a limited amount of garbage.
Would the inclusion of other search engines change the results of this study (e.g., Embase)?
If we were to use extensive search strategies, Embase and Scopus would have provided 72,811 results, a little bit too much even for 20 persons to get through; by using instead a restrictive and focused strategy, we would obtain 572 articles, a little bit too little and much less than PubMed. Reuter’s/Clarivate’s Web of Science is a bit better, but does not include articles not significantly overlapping with those of PubMed. By using PubMed one gets almost all what is needed. However, we added this point in Limitations, although we do not believe it was a real limitation. Thank you for carefully reading our manuscript and for suggesting tips that could improve it.